# PRINCIPLED FEDERATED DOMAIN ADAPTATION: GRADIENT PROJECTION AND AUTO-WEIGHTING

**Enyi Jiang**[*]
enyij2@illinois.edu
UIUC

**Yibo Jacky Zhang**[*]
yiboz@stanford.edu
Stanford

**Sanmi Koyejo**
sanmi@cs.stanford.edu
Stanford

## ABSTRACT

Federated Domain Adaptation (FDA) describes the federated learning (FL) setting where source clients and a server work collaboratively to improve the performance of a target client where limited data is available. The domain shift between the source and target domains, coupled with limited data of the target client, makes FDA a challenging problem, e.g., common techniques such as federated averaging and fine-tuning fail due to domain shift and data scarcity. To theoretically understand the problem, we introduce new metrics that characterize the FDA setting and a theoretical framework with novel theorems for analyzing the performance of server aggregation rules. Further, we propose a novel lightweight aggregation rule, Federated Gradient Projection (`FedGP`), which significantly improves the target performance with domain shift and data scarcity. Moreover, our theory suggests an *auto-weighting scheme* that finds the optimal combinations of the source and target gradients. This scheme improves both `FedGP` and a simpler heuristic aggregation rule. Extensive experiments verify the theoretical insights and illustrate the effectiveness of the proposed methods in practice.

## 1 INTRODUCTION

Federated learning (FL) is a distributed machine learning paradigm that aggregates clients' models on the server while maintaining data privacy (McMahan et al., 2017). FL is particularly interesting in real-world applications where data heterogeneity and insufficiency are common issues, such as healthcare settings. For instance, a small local hospital may struggle to train a generalizable model independently due to insufficient data, and the domain divergence from other hospitals further complicates the application of FL. A promising framework for addressing this problem is Federated Domain Adaptation (FDA), where source clients collaborate with the server to enhance the model performance of a target client (Peng et al., 2020). FDA presents a considerable hurdle due to two primary factors: (i) the *domain shift* existing between source and target domains, and (ii) the *scarcity of data* in the target domain.

Recent works have studied these challenges of domain shift and limited data in federated settings. Some of these works aim to minimize the impacts of distribution shifts between clients (Wang et al., 2019; Karimireddy et al., 2020; Xie et al., 2020b), e.g., via personalized federated learning (Deng et al., 2020; Li et al., 2021; Collins et al., 2021; Marfoq et al., 2022). However, these studies commonly presume that all clients possess ample data, an assumption that may not hold for small hospitals in a cross-silo FL setting. Data scarcity challenges can be a crucial bottleneck in real-world scenarios, e.g., small hospitals lack *data* – whether labeled or unlabeled. In the special (and arguably less common) case where the target client has access to abundant unlabeled data, Unsupervised Federated Domain Adaptation (UFDA) (Peng et al., 2020; Feng et al., 2021; Wu & Gong, 2021) may be useful. Despite existing work, there remains an under-explored gap in the literature addressing both challenges, namely *domain shift* and *data scarcity*, coexist.

To fill the gap, this work directly approaches the two principal challenges associated with FDA, focusing on carefully designing server aggregation rules, i.e., mechanisms used by the server to combine updates across source and target clients within each global optimization loop. We focus on

---

[*]Equal contribution

aggregation rules as they are easy to implement – requiring only the server to change its operations (e.g., variations of federated averaging (McMahan et al., 2017)), and thus have become the primary target of innovation in the federated learning literature. In brief, our work is motivated by the question:

*How does one define a "good" FDA aggregation rule?*

To our best understanding, there are no theoretical foundations that systematically examine the behaviors of various federated aggregation rules within the context of the FDA. Therefore, we introduce a theoretical framework that establishes two metrics to characterize the FDA settings and employ them to analyze the performance of FDA aggregation rules. The proposed metrics characterize (i) the divergence between source and target domains and (ii) the level of training data scarcity in the target domain. Leveraging the proposed theoretical framework, we propose and analyze two aggregation approaches. The first is a simple heuristic `FedDA`, a simple convex combination of source and target gradients. Perhaps surprisingly, we discover that even noisy gradients, computed using the limited data of the target client, can still deliver a valuable signal. The second is a novel filtering-based gradient projection method, `FedGP`. This method is designed to extract and aggregate beneficial components of the source gradients with the assistance of the target gradient, as depicted in Figure 1. `FedGP` calculates a convex combination of the target gradient and its positive projection along the direction of source gradients. Intriguingly, using a generalization analysis on the target domain, our theoretical framework unravels why `FedGP` may outperform `FedDA`– specifically, we find that performing the projection operation before the convex combination is crucial.

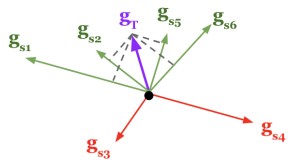

Figure 1: `FedGP` filters out the negative source gradients (colored in red) and convexly combines $g_T$ and its projections to to direction of the remaining source gradients (green ones).

Importantly, our theoretical framework suggests the optimal weights for combining source and target gradients, leading to auto-weighted versions of both `FedGP` and `FedDA`. In particular, we find that the under-performing `FedDA` is significantly improved by using auto-weighting – enough to be competitive with `FedGP`, demonstrating the value of our theory. Across extensive datasets, we demonstrate that `FedGP`, as well as the auto-weighted `FedDA` and `FedGP`, outperforms personalized FL and UFDA baselines. Our code is at https://github.com/jackyzyb/AutoFedGP.

**Summary of Contributions.** Our contributions are both theoretical and practical, addressing the FDA problem through federated aggregation in a principled way.

- We introduce a theoretical framework understanding and analyzing the performance of FDA aggregation rules, inspired by two challenges existing in FDA. Our theories provide a principled response to the question: *How do we define a "good" FDA aggregation rule?*
- We propose `FedGP` as an effective solution to the FDA challenges of substantial domain shifts and limited target data.
- Our theory determines the optimal weight parameter for aggregation rules, `FedDA` and `FedGP`. This *auto-weighting* scheme leads to further performance improvements.
- Extensive experiments illustrate that our theory is predictive of practice. The proposed methods outperform personalized FL and UFDA baselines on real-world ColoredMNIST, VLCS, TerraIncognita, and DomainNet datasets.

## 2 THE PROBLEM OF FEDERATED DOMAIN ADAPTATION

We begin with a general definition of the problem of Federated Domain Adaptation and, subsequently a review of related literature in the field.

**Notation.** Let $\mathcal{D}$ be a data domain[1] on a ground set $\mathcal{Z}$. In our supervised setting, a data point $z \in \mathcal{Z}$ is the tuple of input and output data[2] We denote the loss function as $\ell : \Theta \times \mathcal{Z} \to \mathbb{R}_+$ where the parameter space is $\Theta = \mathbb{R}^m$; an $m$-dimensional Euclidean space. The population loss is $\ell_{\mathcal{D}}(\theta) := \mathbb{E}_{z \sim \mathcal{D}} \ell(\theta, z)$, where $\mathbb{E}_{z \sim \mathcal{D}}$ is the expectation w.r.t. $\mathcal{D}$. Let $\widehat{\mathcal{D}}$ be a finite sample dataset

---

[1]In this paper, the terms distribution and domain are used interchangeably.

[2]For example, let $x$ be the inputs and $y$ be the targets, then $z = (x, y)$.

drawn from $\mathcal{D}$, then $\ell_{\widehat{\mathcal{D}}}(\theta) := \frac{1}{|\widehat{\mathcal{D}}|} \sum_{z \in \widehat{\mathcal{D}}} \ell(\theta, z)$, where $|\widehat{\mathcal{D}}| = n$ is the size of the dataset. We use $[N] := \{1, 2, \ldots, N\}$. By default, $\langle \cdot, \cdot \rangle$, and $\| \cdot \|$ denote the Euclidean inner product and Euclidean norm, respectively.

In FDA, there are $N$ source clients with their respective source domains $\{\mathcal{D}_{S_i}\}_{i \in [N]}$ and a target client with the target domain $\mathcal{D}_T$. For $\forall i \in [N]$, $\widehat{\mathcal{D}}_{S_i}$ denotes the $i^{th}$ source client dataset, and $\widehat{\mathcal{D}}_T$ denotes the target client dataset. We focus on the setting where $|\widehat{\mathcal{D}}_T|$ is relatively small. In standard federated learning, all clients collaborate to learn a global model orchestrated by the server, i.e, clients cannot communicate directly, and all information is shared from/to the server. In contrast, FDA uses the same system architecture to improve a single client's performance.

**Definition 2.1** (Aggregation for Federated Domain Adaptation (FDA)). *The FDA problem is a federated learning problem where all clients collaborate to improve the global model for a target domain. The global model is trained by iteratively updating the global model parameter*

$$\theta \leftarrow \theta - \mu \cdot \mathtt{Aggr}(\{\nabla \ell_{\widehat{\mathcal{D}}_{S_i}}(\theta)\}_{i \in [N]}, \nabla \ell_{\widehat{\mathcal{D}}_T}(\theta)),$$

*where $\nabla \ell$ is the gradient, $\mu$ is the step size. We seek an aggregation strategy $\mathtt{Aggr}(\cdot)$ such that after training, the global model parameter $\theta$ minimizes the target domain population loss function $\ell_{\mathcal{D}_T}(\theta)$. Note that we allow the aggregation function $\mathtt{Aggr}(\cdot)$ to depend on the iteration index.*

There are a number of challenges that make FDA difficult to solve. First, the amount of labeled data in the target domain is typically limited, which makes it difficult to learn a generalizable model. Second, the source and target domains have different data distributions, which can lead to a mismatch between the features learned by the source and target models. Moreover, the model must be trained in a privacy-preserving manner where local data cannot be shared.

## 2.1 RELATED WORK

**Data heterogeneity, personalization and label deficiency in FL.** Distribution shifts between clients remain a crucial challenge in FL. Current work often focuses on improving the aggregation rules: Karimireddy et al. (2020) use control variates and Xie et al. (2020b) cluster the client weights to correct the drifts among clients. More recently, there are works (Deng et al., 2020; Li et al., 2021; Collins et al., 2021; Marfoq et al., 2022) concentrating on personalized federated learning by finding a better mixture of local/global models and exploring shared representation. Further, recent works have addressed the label deficiency problem with self-supervision or semi-supervision for personalized models (Jeong et al., 2020; He et al., 2021; Yang et al., 2021). To our knowledge, all existing work assumes sufficient data for all clients - nevertheless, the performance of a client with data deficiency and large shifts may become unsatisfying (Table 1). Compared to related work on personalized FL, our method is more robust to data scarcity on the target client.

**Unsupervised federated domain adaptation.** There is a considerable amount of recent work on unsupervised federated domain adaptation (UFDA), with recent highlights in utilizing adversarial networks (Saito et al., 2018; Zhao et al., 2018), knowledge distillation (Nguyen et al., 2021), and source-free methods (Liang et al., 2020). Peng et al. (2020); Li et al. (2020) is the first to extend MSDA into an FL setting; they apply adversarial adaptation techniques to align the representations of nodes. More recently, in KD3A (Feng et al., 2021) and COPA (Wu & Gong, 2021), the server with unlabeled target samples aggregates the local models by learning the importance of each source domain via knowledge distillation and collaborative optimization. Their work assumes abundant data without labels in the target domain, while small hospitals usually do not have enough data. Also, training with unlabeled data every round is computationally expensive. Compared to their work, we study a more important challenge where the data (not just the labels) are scarce. We show our approaches achieve superior performance using substantially less target data on various benchmarks.

**Using additional gradient information in FL.** Model updates in each communication round may provide valuable insights into client convergence directions. This idea has been explored for robustness in FL, particularly with untrusted clients. For example, Zeno++ (Xie et al., 2020a) and FLTrust (Cao et al., 2021) leverage the additional gradient computed from a small clean training dataset on the server to compute the scores of candidate gradients for detecting the malicious adversaries. Differently, our work focuses on a different task of improving the performance of the target domain with auto-weighted aggregation rules that utilize the gradient signals from all clients.

## 3 A THEORETICAL FRAMEWORK FOR ANALYZING AGGREGATION RULES FOR FDA

This section introduces a general framework and a theoretical analysis of aggregation rules for federated domain adaptation.

**Additional Notation and Setting.** We use additional notation to motivate a functional view of FDA. Let $g_{\mathcal{D}} : \Theta \to \Theta$ with $g_{\mathcal{D}}(\theta) := \nabla \ell_{\mathcal{D}}(\theta)$. Given a distribution $\pi$ on the parameter space $\Theta$, we define an inner product $\langle g_{\mathcal{D}}, g_{\mathcal{D}'} \rangle_\pi = \mathbb{E}_{\theta \sim \pi}[\langle g_{\mathcal{D}}(\theta), g_{\mathcal{D}'}(\theta) \rangle]$. We interchangeably denote $\pi$ as both the distribution and the probability measure. The inner product induces the $L^\pi$-norm on $g_{\mathcal{D}}$ as $\|g_{\mathcal{D}}\|_\pi := \sqrt{\mathbb{E}_{\theta \sim \pi} \|g_{\mathcal{D}}(\theta)\|^2}$. With the $L^\pi$-norm, we define the $L^\pi$ space as $\{g : \Theta \to \Theta \mid \|g\|_\pi < \infty\}$. Given an aggregation rule $\texttt{Aggr}(\cdot)$, we denote $\widehat{g}_{\texttt{Aggr}}(\theta) = \texttt{Aggr}(\{g_{\mathcal{D}_{S_i}}(\theta)\}_{i=1}^N, g_{\widehat{\mathcal{D}}_T}(\theta))$. Note that we do not care about the generalization on the source domains, and therefore, for the theoretical analysis, we can view $\mathcal{D}_S = \widehat{\mathcal{D}}_S$ without loss of generality. Throughout our theoretical analysis, we make the following standard assumption about $\widehat{\mathcal{D}}_T$, and we use the hat symbol, $\widehat{\cdot}$, to emphasize that a random variable is associated with the sampled target domain dataset $\widehat{\mathcal{D}}_T$.

**Assumption 3.1.** *We assume the target domain's local dataset $\widehat{\mathcal{D}}_T = \{z_i\}_{i \in [n]}$ consists of $n$ i.i.d. samples from its underlying target domain distribution $\mathcal{D}_T$. Note that this implies $\mathbb{E}_{\widehat{\mathcal{D}}_T}[g_{\widehat{\mathcal{D}}_T}] = g_{\mathcal{D}_T}$.*

Observing Definition 2.1, we can see intuitively that a good aggregation rule should have $\widehat{g}_{\texttt{Aggr}}$ be "close" to the ground-truth target domain gradient $g_{\mathcal{D}_T}$. From a functional view, we need to measure the distance between the two functions. We choose the $L^\pi$-norm, formally stated in the following.

**Definition 3.2** (Delta Error of an aggregation rule $\texttt{Aggr}(\cdot)$). *We define the following squared error term to measure the closeness between $\widehat{g}_{\texttt{Aggr}}$ and $g_{\mathcal{D}_T}$, i.e.,*

$$\Delta^2_{\texttt{Aggr}} := \mathbb{E}_{\widehat{\mathcal{D}}_T} \|g_{\mathcal{D}_T} - \widehat{g}_{\texttt{Aggr}}\|_\pi^2.$$

*The distribution $\pi$ characterizes where to measure the gradient difference in the parameter space.*

The Delta error $\Delta^2_{\texttt{Aggr}}$ is crucial in theoretical analysis and algorithm design due to its two main benefits. First, it indicates the performance of an aggregation rule. Second, it reflects fundamental domain properties that are irrelevant to the aggregation rules applied. Thus, the Delta error disentangles these two elements, allowing in-depth analysis and algorithm design.

Concretely, for the first benefit: one expects an aggregation rule with a small Delta error to converge better as measured by the population target domain loss function gradient $\nabla \ell_{\mathcal{D}_T}$.

**Theorem 3.3** (Convergence and Generalization). *For any probability measure $\pi$ over the parameter space, and an aggregation rule $\texttt{Aggr}(\cdot)$ with step size $\mu > 0$. Given target domain sampled dataset $\widehat{\mathcal{D}}_T$, update the parameter for $T$ steps by $\theta^{t+1} := \theta^t - \mu \widehat{g}_{\texttt{Aggr}}(\theta^t)$. Assume the gradient $\nabla \ell(\theta, z)$ and $\widehat{g}_{\texttt{Aggr}}(\theta)$ are $\frac{\gamma}{2}$-Lipschitz in $\theta$ such that $\theta^t \to \widehat{\theta}_{\texttt{Aggr}}$. Then, given step size $\mu \leq \frac{1}{\gamma}$ and a small enough $\epsilon > 0$, with probability at least $1 - \delta$ we have*

$$\|\nabla \ell_{\mathcal{D}_T}(\theta^T)\|^2 \leq \frac{1}{\delta^2} \left( \sqrt{C_\epsilon \cdot \Delta^2_{\texttt{Aggr}}} + \mathcal{O}(\epsilon) \right)^2 + \mathcal{O}\left(\frac{1}{T}\right) + \mathcal{O}(\epsilon),$$

*where $C_\epsilon = \mathbb{E}_{\widehat{\mathcal{D}}_T}[1/\pi(B_\epsilon(\widehat{\theta}_{\texttt{Aggr}}))]^2$ and $B_\epsilon(\widehat{\theta}_{\texttt{Aggr}}) \subset \mathbb{R}^m$ is the ball with radius $\epsilon$ centered at $\widehat{\theta}_{\texttt{Aggr}}$. The $C_\epsilon$ measures how well the probability measure $\pi$ covers where the optimization goes, i.e., $\widehat{\theta}_{\texttt{Aggr}}$.*

**Interpretation.** The left-hand side reveals the convergence quality of the optimization with respect to the *true* target domain loss. As we can see, a smaller Delta error indicates better convergence and generalization. In addition, we provide an analysis of a single gradient step in Theorem A.1, showing similar properties of the Delta error. In the above theorem, $\pi$ is arbitrary, allowing for its appropriate choice to minimize the $C_\epsilon$. Ideally, $\pi$ would accurately cover where the model parameters are after optimization. We take this insight in the design of an auto-weighting algorithm, to be discussed later.

The behavior of an aggregation rule should vary with the degree and nature of source-target domain shift and the data sample quality in the target domain. This suggests the necessity of their formal characterizations for further in-depth analysis. Given a source domain $\mathcal{D}_S$, we can measure its distance to the target domain $\mathcal{D}_T$ as the $L^\pi$-norm distance between $g_{\mathcal{D}_S}$ and the target domain model ground-truth gradient $g_{\mathcal{D}_T}$, hence the following definition.

**Definition 3.4** ($L^\pi$ Source-Target Domain Distance). *Given a source domain $\mathcal{D}_S$, its distance to the target domain $\mathcal{D}_T$ is defined as*

$$d_\pi(\mathcal{D}_S, \mathcal{D}_T) := \|g_{\mathcal{D}_T} - g_{\mathcal{D}_S}\|_\pi.$$

This proposed metric $d_\pi$ has some properties inherited from the norm, including: *(i. symmetry)* $d_\pi(\mathcal{D}_S, \mathcal{D}_T) = d_\pi(\mathcal{D}_T, \mathcal{D}_S)$; *(ii, triangle inequality)* For any data distribution $\mathcal{D}$ we have $d_\pi(\mathcal{D}_S, \mathcal{D}_T) \leq d_\pi(\mathcal{D}_S, \mathcal{D}) + d_\pi(\mathcal{D}_T, \mathcal{D})$; *(iii. zero property)* For any $\mathcal{D}$ we have $d_\pi(\mathcal{D}, \mathcal{D}) = 0$.

To formalize the target domain sample quality, we again measure the distance between $\widehat{\mathcal{D}}_T$ and $\mathcal{D}_T$. Thus, its mean squared error characterizes how the sample size affects the target domain variance.

**Definition 3.5** ($L^\pi$ Target Domain Variance). *Given the target domain $\mathcal{D}_T$ and dataset $\widehat{\mathcal{D}}_T = \{z_i\}_{i\in[n]}$ where $z_i \sim \mathcal{D}_T$ is sampled i.i.d., the target domain variance is defined as*

$$\sigma_\pi^2(\widehat{\mathcal{D}}_T) := \mathbb{E}_{\widehat{\mathcal{D}}_T}\|g_{\mathcal{D}_T} - g_{\widehat{\mathcal{D}}_T}\|_\pi^2 = \frac{1}{n}\mathbb{E}_{z\sim\mathcal{D}_T}\|g_{\mathcal{D}_T} - \nabla\ell(\cdot, z)\|_\pi^2 =: \frac{1}{n}\sigma_\pi^2(z),$$

*where $\sigma_\pi^2(z)$ is the variance of a single sampled gradient function $\nabla\ell(\cdot, z)$.*

Taken together, our exposition shows the second benefit of the Delta error: it decomposes into a mix of the target-source domain shift $d_\pi(\mathcal{D}_S, \mathcal{D}_T)$ and the target domain variance $\sigma_\pi^2(\widehat{\mathcal{D}}_T)$ for at least a wide range of aggregation rules (including our `FedDA` and `FedGP`).

**Theorem 3.6** ($\Delta_{Aggr}^2$ Decomposition Theorem). *Consider any aggregation rule* `Aggr`$(\cdot)$ *in the form of $\widehat{g}_{Aggr} = \frac{1}{N}\sum_{i\in[N]} F_{Aggr}[g_{\widehat{\mathcal{D}}_T}, g_{\mathcal{D}_{S_i}}]$, i.e., the aggregation rule is defined by a mapping $F_{Aggr} : L^\pi \times L^\pi \to L^\pi$. If $F_{Aggr}$ is affine w.r.t. its first argument (i.e., the target gradient function), and $\forall g \in L^\pi : F_{Aggr}[g, g] = g$, and the linear mapping associated with $F_{Aggr}$ has its eigenvalue bounded in $[\lambda_{min}, \lambda_{max}]$, then for any source and target distributions $\{\mathcal{D}_{S_i}\}_{i\in[N]}, \mathcal{D}_T, \widehat{\mathcal{D}}_T$ we have $\Delta_{Aggr}^2 \leq \frac{1}{N}\sum_{i\in[N]} \Delta_{Aggr, \mathcal{D}_{S_i}}^2$, where*

$$\Delta_{Aggr, \mathcal{D}_{S_i}}^2 \leq \max\{\lambda_{max}^2, \lambda_{min}^2\} \cdot \frac{\sigma_\pi^2(z)}{n} + \max\{(1 - \lambda_{max})^2, (1 - \lambda_{min})^2\} \cdot d_\pi(\mathcal{D}_{S_i}, \mathcal{D}_T)^2.$$

**Interpretation.** The implications of this theorem are significant: first, it predicts how effective an aggregation rule would be, which we use to compare `FedGP` vs. `FedDA`. Second, given an estimate of the domain distance $d_\pi(\mathcal{D}_{S_i}, \mathcal{D}_T)$ and the target domain variance $\sigma_\pi^2(\widehat{\mathcal{D}}_T)$, we can optimally select hyper-parameters for the aggregation operation, a process we name the auto-weighting scheme.

With the relevant quantities defined, we can describe an alternative definition of FDA, useful for our analysis, which answers the pivotal question of *how do we define a "good" aggregation rule*.

**Definition 3.7** (An Error-Analysis Definition of FDA Aggregation). *Given the target domain variance $\sigma_\pi^2(\widehat{\mathcal{D}}_T)$ and source-target domain distances $\forall i \in [N] : d_\pi(\mathcal{D}_{S_i}, \mathcal{D}_T)$, the problem of FDA is to find a good strategy* `Aggr`$(\cdot)$ *such that its Delta error $\Delta_{Aggr}^2$ is minimized.*

These definitions give a powerful framework for analyzing and designing aggregation rules:

- given an aggregation rule, we can derive its Delta error and see how it would perform given an FDA setting (as characterized by the target domain variance and the source-target distances);
- given an FDA setting, we can design aggregation rules to minimize the Delta error.

## 4 METHODS: GRADIENT PROJECTION AND THE AUTO-WEIGHTING SCHEME

To start, we may try two simple methods, i.e., only using the target gradient and only using a source gradient (e.g., the $i^{th}$ source domain). The Delta error of these baseline aggregation rules is straightforward. By definition, we have that

$$\Delta_{\widehat{\mathcal{D}}_T \text{ only}}^2 = \sigma_\pi^2(\widehat{\mathcal{D}}_T), \quad \text{and} \quad \Delta_{\mathcal{D}_{S_i} \text{ only}}^2 = d_\pi^2(\mathcal{D}_{S_i}, \mathcal{D}_T).$$

This immediate result demonstrates the usefulness of the proposed framework: if $\widehat{g}_{\texttt{Aggr}}$ only uses the target gradient then the error is the target domain variance; if $\widehat{g}_{\texttt{Aggr}}$ only uses a source gradient then

the error is the corresponding source-target domain bias. Therefore, a good aggregation method must strike a balance between the bias and variance, i.e., a bias-variance trade-off, and this is precisely what we will design our auto-weighting mechanism to do. Next, we propose two aggregation methods and then show how their auto-weighting can be derived.

## 4.1 THE AGGREGATION RULES: FedDA AND FedGP

A straightforward way to combine the source and target gradients is to convexly combine them, as defined in the following.

**Definition 4.1** (FedDA). *For each source domains $i \in [N]$, let $\beta_i \in [0,1]$ be the weight that balances between the $i^{th}$ source domain and the target domain. The FedDA aggregation operation is*

$$FedDA(\{g_{\mathcal{D}_{S_i}}(\theta)\}_{i=1}^N, g_{\widehat{\mathcal{D}}_T}(\theta)) = \frac{1}{N} \sum_{i=1}^N \left( (1-\beta_i)g_{\widehat{\mathcal{D}}_T}(\theta) + \beta_i g_{\mathcal{D}_{S_i}}(\theta) \right).$$

Let us examine the Delta error of FedDA.

**Theorem 4.2.** *Consider FedDA. Given the target domain $\widehat{\mathcal{D}}_T$ and $N$ source domains $\mathcal{D}_{S_1}, \ldots, \mathcal{D}_{S_N}$, we have $\Delta^2_{FedDA} \leq \frac{1}{N} \sum_{i=1}^N \Delta^2_{FedDA,S_i}$, where*

$$\Delta^2_{FedDA,S_i} = (1-\beta_i)^2 \sigma^2_\pi(\widehat{\mathcal{D}}_T) + \beta_i^2 d^2_\pi(\mathcal{D}_{S_i}, \mathcal{D}_T). \tag{1}$$

Therefore, we can see the benefits of combining the source and target domains. For example, with $\beta_i = \frac{1}{2}$, we have $\Delta^2_{FedDA,S_i} = \frac{1}{4}\sigma^2_\pi(\widehat{\mathcal{D}}_T) + \frac{1}{4}d^2_\pi(\mathcal{D}_{S_i}, \mathcal{D}_T)$. We note that $\Delta^2_{FedDA,S_i}$ attains the upper bound in Theorem 3.6 with $\lambda_{max} = \lambda_{min} = 1 - \beta_i$. This hints that there may be other aggregation rules that can do better, as we shown in the following.

Intuitively, due to domain shift, signals from source domains may not always be relevant. Inspired by the filtering technique in Byzantine robustness of FL (Xie et al., 2020a), we propose Federated Gradient Projection (FedGP). This method refines and combines beneficial components of the source gradients, aided by the target gradient, by gradient projection and filtering out unfavorable ones.

**Definition 4.3** (FedGP). *For each source domains $i \in [N]$, let $\beta_i \in [0,1]$ be the weight that balances between $i^{th}$ source domain and the target domain. The FedGP aggregation operation is*

$$FedGP(\{g_{\mathcal{D}_{S_i}}(\theta)\}_{i=1}^N, g_{\widehat{\mathcal{D}}_T}(\theta)) = \frac{1}{N} \sum_{i=1}^N \left( (1-\beta_i)g_{\widehat{\mathcal{D}}_T}(\theta) + \beta_i Proj_+(g_{\widehat{\mathcal{D}}_T}(\theta)|g_{\mathcal{D}_{S_i}}(\theta)) \right).$$

*where $Proj_+(g_{\widehat{\mathcal{D}}_T}(\theta)|g_{\mathcal{D}_{S_i}}(\theta)) = \max\{\langle g_{\widehat{\mathcal{D}}_T}(\theta), g_{\mathcal{D}_{S_i}}(\theta)\rangle, 0\}g_{\mathcal{D}_{S_i}}(\theta)/\|g_{\mathcal{D}_{S_i}}(\theta)\|^2$ is the operation that projects $g_{\widehat{\mathcal{D}}_T}(\theta)$ to the positive direction of $g_{\mathcal{D}_{S_i}}(\theta)$.*

We first derive the Delta error of FedGP, and compare it to that of FedDA.

**Theorem 4.4** (Informal Version). *Consider FedGP. Given the target domain $\widehat{\mathcal{D}}_T$ and $N$ source domains $\mathcal{D}_{S_1}, \ldots, \mathcal{D}_{S_N}$, we have $\Delta^2_{FedGP} \leq \frac{1}{N} \sum_{i=1}^N \Delta^2_{FedGP,S_i}$, where*

$$\Delta^2_{FedGP,S_i} \approx \left( (1-\beta_i)^2 + \frac{2\beta_i - \beta_i^2}{m} \right) \sigma^2_\pi(\widehat{\mathcal{D}}_T) + \beta_i^2 \bar{\tau}^2 d^2_\pi(\mathcal{D}_{S_i}, \mathcal{D}_T), \tag{2}$$

*In the above equation, $m$ is the model dimension and $\bar{\tau}^2 = \mathbb{E}_\pi[\tau(\theta)^2] \in [0,1]$ where $\tau(\theta)$ is the $\sin(\cdot)$ value of the angle between $g_{\mathcal{D}_S}(\theta)$ and $g_{\mathcal{D}_T}(\theta) - g_{\mathcal{D}_S}(\theta)$.*

We note the above theorem is the approximated version of Theorem A.5 where the derivation is non-trivial. The approximations are mostly done in analog to a mean-field analysis, which are detailed in Appendix A.4.

**Interpretation.** Comparing the Delta error of FedGP (equation 2) and that of FedDA (equation 1), we can see that FedGP is more robust to large source-target domain shift $d_\pi(\mathcal{D}_{S_i}, \mathcal{D}_T)$ given $\bar{\tau} < 1$. This aligns with our motivation of FedGP which filters out biased signals from the source domain. Moreover, our theory reveals a surprising benefit of FedGP as follows. Note that

$$\Delta^2_{FedDA,S_i} - \Delta^2_{FedGP,S_i} \approx \beta_i^2(1 - \bar{\tau}^2)d^2_\pi(\mathcal{D}_{S_i}, \mathcal{D}_T) - \frac{2\beta_i - \beta_i^2}{m}\sigma^2_\pi(\widehat{\mathcal{D}}_T).$$

In practice, the model dimension $m \gg 1$ while the $\bar{\tau}^2 < 1$, thus we can expect FedGP to be mostly better than FedDA with the same weight $\beta_i$. With that said, we move on the auto-weighting scheme.

## 4.2 THE AUTO-WEIGHTING FEDGP AND FEDDA

Naturally, the above analysis implies a good choice of weighting parameters for either of the methods. For each source domains $S_i$, we can solve for the optimal $\beta_i$ that minimize the corresponding Delta errors, i.e., $\Delta^2_{\text{FedDA},S_i}$ (equation 1) for FedDA and $\Delta^2_{\text{FedGP},S_i}$ (equation 2) for FedGP. Note that for $\Delta^2_{\text{FedGP},S_i}$ we can safely view $\frac{2\beta_i - \beta_i^2}{m} \approx 0$ given the high dimensionality of our models. Since either of the Delta errors is quadratic in $\beta_i$, they enjoy closed-form solutions:

$$\beta_i^{\text{FedDA}} = \frac{\sigma_\pi^2(\widehat{\mathcal{D}}_T)}{d_\pi^2(\mathcal{D}_{S_i}, \mathcal{D}_T) + \sigma_\pi^2(\widehat{\mathcal{D}}_T)}, \qquad \beta_i^{\text{FedGP}} = \frac{\sigma_\pi^2(\widehat{\mathcal{D}}_T)}{\bar{\tau}^2 d_\pi^2(\mathcal{D}_{S_i}, \mathcal{D}_T) + \sigma_\pi^2(\widehat{\mathcal{D}}_T)}. \tag{3}$$

The exact values of $\sigma_\pi^2(\widehat{\mathcal{D}}_T), d_\pi^2(\mathcal{D}_{S_i}, \mathcal{D}_T), \bar{\tau}^2$ are unknown, since they would require knowing the ground-truth target domain gradient $g_{\mathcal{D}_T}$. Fortunately, using only the available training data, we can efficiently obtain unbiased estimators for those values, and accordingly obtain estimators for the best $\beta_i$. The construction of the estimators is non-trivial and is detailed in Appendix A.5.

The proposed methods are summarized in Algorithm 1, and detailed in Appendix C. During one round of computation, the target domain client does $B$ local model updates with $B$ batches of data. In practice, we use these intermediate local updates $\{g_{\widehat{\mathcal{D}}_T}^j\}_{j=1}^B$ to estimate the optimal $\beta_i$, where $g_{\widehat{\mathcal{D}}_T}^j$ stands for the local model update using the $j^{th}$ batch of data. In other words, we choose $\pi$ to be the empirical distribution of the model parameters encountered along the optimization path, aligning with Theorem 3.3's suggestion for an ideal $\pi$.

We observe that FedGP, quite remarkably, is robust to the choice of $\beta$: simply choosing $\beta = 0.5$ is good enough for most of the cases as observed in our experiments. On the other hand, although FedDA is sensitive to the choice of $\beta$, the auto-weighted procedure significantly improves the performance for FedDA, demonstrating the usefulness of our theoretical framework.

---

**Algorithm 1** FDA: Gradient Projection and the Auto-Weighting Scheme

---

**Input:** $N$ source domains $\mathcal{D}_S = \{\mathcal{D}_{S_i}\}_{i=1}^N$, target domain $\mathcal{D}_T$; $N$ source clients $\{\mathcal{C}_{S_i}\}_{i=1}^N$, target client $\mathcal{C}_T$, server $\mathcal{S}$; number of rounds $R$; aggregation rule Aggr; whether to use auto_weight.
Initialize global model $h_{global}^{(0)}$. Default $\{\beta_i\}_{i=1}^N \leftarrow \{0.5\}_{i=1}^N$.
**for** $r = 1, 2, ..., R$ **do**
    **for** source domain client $\mathcal{C}_{S_i}$ in $\{\mathcal{C}_{S_i}\}_{i=1}^N$ **do**
        Initialize local model $h_{S_i}^{(r)} \leftarrow h_{global}^{(r-1)}$, optimize $h_{S_i}^{(r)}$ on $\mathcal{D}_{S_i}$, send $h_{S_i}^{(r)}$ to server $\mathcal{S}$.
    **end for**
    Target domain client $\mathcal{C}_T$ initialize $h_T^{(r)} \leftarrow h_{global}^{(r-1)}$, optimizes $h_T^{(r)}$ on $\mathcal{D}_T$, send $h_T^{(r)}$ to server $\mathcal{S}$.
    Server $\mathcal{S}$ computes model updates $g_T \leftarrow h_T^{(r)} - h_{global}^{(r-1)}$ and $g_{S_i} \leftarrow h_{S_i}^{(r)} - h_{global}^{(r-1)}$ for $i \in [N]$.
    **if** auto_weight **then**
        $\mathcal{C}_T$ sends intermediate local model updates $\{g_{\widehat{\mathcal{D}}_T}^j\}_{j=1}^B$ to $\mathcal{S}$.
        $\mathcal{S}$ estimates of $\{d_\pi(\mathcal{D}_{S_i}, \mathcal{D}_T)\}_{i=1}^N, \bar{\tau}^2$ and $\sigma_\pi(\widehat{\mathcal{D}}_T)$ using $\{g_{S_i}\}_{i=1}^N$ and $\{g_{\widehat{\mathcal{D}}_T}^j\}_{j=1}^B$.
        $\mathcal{S}$ updates $\{\beta_i\}_{i=1}^N$ according to (3).
    **end if**
    $\mathcal{S}$ updates the global model as $h_{global}^{(r)} \leftarrow h_{global}^{(r-1)} + \text{Aggr}(\{g_{S_i}\}_{i=1}^N, g_T, \{\beta_i\}_{i=1}^N)$.
**end for**

---

## 5 EXPERIMENTS

In this section, we present and discuss the results of real dataset experiments with controlled domain shifts (Section 5.1) and real-world domain shifts (Section 5.2). Ablation studies on target data scarcity and visualizations are available in Appendix C.5 & C.8. Synthetic data experiments verifying our theoretical insights are presented in Appendix B. In Appendix C.3, we show our methods surpass UFDA and Domain Generalization (DG) methods on PACS (Li et al., 2017), Office-Home (Venkateswara et al., 2017), and DomainNet (Peng et al., 2019). Implementation details and extended experiments can be found in the appendix.

## 5.1 SEMI-SYNTHETIC DATASET EXPERIMENTS WITH VARING SHIFTS

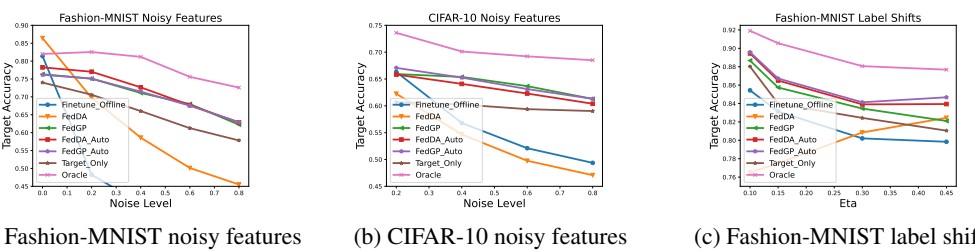

(a) Fashion-MNIST noisy features     (b) CIFAR-10 noisy features     (c) Fashion-MNIST label shifts

Figure 2: The impact of changing domain shifts with noisy features or label shifts.

**Datasets, models, and methods.** We create controlled distribution shifts by adding different levels of feature noise and label shifts to Fashion-MNIST (Xiao et al., 2017) and CIFAR-10 (Krizhevsky et al., 2009) datasets, adapting from the Non-IID benchmark (Li et al., 2022) with the following two settings: 1) **Noisy features:** We add Gaussian noise levels of $std = (0.2, 0.4, 0.6, 0.8)$ to input images of the target client of two datasets, to create various degrees of shifts between source and target domains. 2) **Label shifts**: We split the Fashion-MNIST into two sets with 3 and 7 classes, respectively, denoted as $D_1$ and $D_2$. $D_S = \eta$ portion from $D_1$ and $(1 - \eta)$ portion from $D_2$, $D_T$ = $(1 - \eta)$ portion from $D_1$ and $\eta$ portion from $D_2$ with $\eta = [0.45, 0.30, 0.15, 0.10, 0.05, 0.00]$. In addition, we use a CNN model architecture. We set the communication round $R = 50$ and the local update epoch to 1, with 10 clients (1 target, 9 source clients) in the system. We compare the following methods: Source Only (only averaging the source gradients), `Finetune_Offline` (fine-tuning locally after source-only training), `FedDA` ($\beta = 0.5$), `FedGP`($\beta = 0.5$), and their auto-weighted versions `FedDA_Auto` and `FedGP_Auto`, the Oracle (supervised training with all target data ($\mathcal{D}_T$)) and Target Only (only use target gradient ($\widehat{\mathcal{D}}_T$)). More details can be found in Appendix C.7.

**Auto-weighted methods and `FedGP` keep a better trade-off between bias and variance.** As shown in Figure 2, when the source-target shifts grow bigger, `FedDA`, `Finetune_Offline`, and Source Only degrade more severely compared with auto-weighted methods, `FedGP` and Target Only. We find that auto-weighted methods and `FedGP` outperform other baselines in most cases, being less sensitive to changing shifts. In addition, auto-weighted `FedDA` manages to achieve a significant improvement compared with the fixed weight `FedDA`, with a competitive performance compared with `FedGP_Auto`, while `FedGP_Auto` generally has the best accuracy compared with other methods, which coincides with the theoretical findings. Full experiment results can be found in Appendix C.7.

## 5.2 REAL DATASET EXPERIMENTS WITH REAL-WORLD SHIFTS

**Datasets, models, baselines, and implementations.** We use the Domainbed (Gulrajani & Lopez-Paz, 2020a) benchmark with multiple domains, with realistic shifts between source and target clients. We conduct experiments on three datasets: ColoredMNIST (Arjovsky et al., 2019), VLCS (Fang et al., 2013), TerraIncognita (Beery et al., 2018) datasets. We randomly sampled $0.1\%$ samples of ColoredMNIST, and $5\%$ samples of VLCS and TerraIncognita for their respective target domains. The task is classifying the target domain. We use a CNN model for ColoredMNIST, ResNet-18 He et al. (2016) for VLCS and TerraIncognita. For baselines in particular, in addition to the methods in Section 5.1, we compare (1) *personalization* baselines: `FedAvg`, `Ditto` (Li et al., 2021), `FedRep` (Collins et al., 2021), `APFL` (Deng et al., 2020), and `KNN-per` (Marfoq et al., 2022); (2) UFDA methods: `KD3A` (Feng et al., 2021) (current SOTA): note that our proposed methods use *few* percentages of target data, while UFDA here uses $100\%$ unlabeled target data; (3) DG method: we report the best DG performance in DomainBed (Gulrajani & Lopez-Paz, 2020b). For each dataset, we test the target accuracy of each domain using the left-out domain as the target and the rest as source domains. More details and full results are in Appendix C.2.

**Our methods consistently deliver superior performance.** Table 1 reveals that our auto-weighted methods outperform others in all cases, and some of their accuracies approach/outperform the corresponding upper bound (Oracle). The auto-weighted scheme improves `FedDA` significantly.

Interestingly, we observe that FedGP, even with default fixed betas ($\beta = 0.5$), achieves competitive results. Our methods surpass personalized FL, UFDA, and DG baselines by significant margins.

| Domains | ColoredMNIST (0.1%) | | | | VLCS (5%) | | | | Terre (5%) |
| | +90% | +80% | -90% | Avg | C | L | V | S | Avg | Avg |
|---|---|---|---|---|---|---|---|---|---|---|
| Source Only | 56.8(0.8) | 62.4(1.8) | 27.8(0.8) | 49.0 | 90.5(5.3) | 60.7(1.8) | 70.2(2.0) | 69.1(2.0) | 72.6 | 37.5 |
| FedDA | 60.5(2.5) | 65.1(1.3) | 33.0(3.2) | 52.9 | 97.7(0.5) | 68.2(1.4) | 75.3(1.5) | 76.7(0.9) | 79.5 | 64.7 |
| FedGP | 83.7(9.9) | 74.4(4.4) | 89.8(0.5) | 82.4 | 99.4(0.3) | 71.1(1.2) | 73.6(3.1) | 78.7(1.3) | 80.7 | 71.2 |
| FedDA_Auto | 85.3(5.7) | 73.1(7.3) | 88.9(1.1) | 82.7 | 99.8(0.2) | 73.1(1.3) | **78.4(1.5)** | 83.7(2.3) | **83.8** | **74.6** |
| FedGP_Auto | **86.2(4.5)** | **76.5(7.3)** | **89.6(0.4)** | **84.1** | **99.9(0.2)** | **73.2(1.8)** | 78.6(1.5) | 83.47(2.5) | **83.8** | 74.4 |
| Target Only | 85.6(4.8) | 73.5(3.0) | 87.1(3.4) | 82.1 | 97.8(1.5) | 68.9(1.9) | 72.3(1.7) | 76.0(1.9) | 78.7 | 67.2 |
| FedAvg | 63.2 | 72.0 | 10.9 | 48.7 | 96.3 | 68.0 | 69.8 | 68.7 | 75.7 | 30.0 |
| Ditto | 62.2 | 71.3 | 19.3 | 50.9 | 95.9 | 67.5 | 70.5 | 66.1 | 75.0 | 28.8 |
| FedRep | 65.4 | 36.4 | 31.7 | 44.5 | 91.1 | 60.4 | 70.3 | 70.1 | 73.0 | 20.9 |
| APFL | 43.8 | 61.6 | 30.2 | 45.2 | 68.6 | 61.0 | 65.4 | 49.9 | 61.2 | 52.7 |
| KNN-per | 67.5 | 67.6 | 12.4 | 49.1 | 97.8 | 65.9 | 75.7 | 74.4 | 78.4 | 43.1 |
| KD3A (**100%** data) | 65.2 | 73.1 | 9.7 | 49.3 | 99.6 | 63.3 | 78.1 | 80.5 | 79.0 | 39.0 |
| Best DG | 49.9 | 62.1 | 10.0 | 40.7 | 96.9 | 65.7 | 73.3 | 78.7 | 78.7 | 48.7 |
| Oracle | 90.0(0.4) | 80.3(0.4) | 90.0(0.5) | 86.8 | 100.0(0.0) | 72.7(2.5) | 78.7(1.4) | 82.7(1.1) | 83.5 | 93.1 |

Table 1: **Target domain test accuracy** (%) on ColoredMNIST, VLCS, and DomainNet.

## 5.3 ABLATION STUDY AND DISCUSSION

**The effect of source-target balancing weight $\beta$.** We run FedDA and FedGP with varying $\beta$ on Fashion-MNIST, CIFAR10, and Colored-MNIST, as shown in Figure 3. In most cases, we observe FedGP outperforming FedDA, and FedDA being *more sensitive* to the varying $\beta$ values, suggesting that FedDA_Auto can choose optimal enough $\beta$. Complete results are in Appendix C.6.

**Effectiveness of projection and filtering.** We show the effectiveness of gradient projection and filtering in Fashion-MNIST and CIFAR-10 noisy feature experiments in Table 4. Compared with FedDA, which does not perform projection and filtering, projection achieves a large (15%) performance gain, especially when the shifts are larger. Further, we generally get a $1\% - 2\%$ gain via filtering.

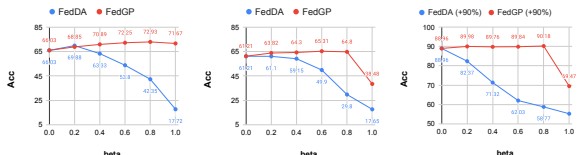

(a) Fashion-MNIST (0.4 noise level)  (b) CIFAR-10 (0.4 noise level)  (c) ColoredMNIST (target: +90%)

Figure 3: The effect of $\beta$ on FedDA and FedGP.

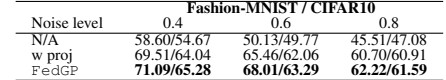

| | Fashion-MNIST / CIFAR10 | | |
| Noise level | 0.4 | 0.6 | 0.8 |
|---|---|---|---|
| N/A | 58.60/54.67 | 50.13/49.77 | 45.51/47.08 |
| w proj | 69.51/64.04 | 65.46/62.06 | 60.70/60.91 |
| FedGP | **71.09/65.28** | **68.01/63.29** | **62.22/61.59** |

Figure 4: Ablation study on projection and filtering.

**Discussion.** We analyze the computational and communication cost for our proposed methods in Appendix C.10 and C.4, where we show how the proposed aggregation rules, especially the auto-weighted operation, can be implemented efficiently. Moreover, we observe in our experiments that Finetune_Offline is sensitive to its pre-trained model (obtained via FedAvg), highlighting the necessity of a deeper study of the relation between personalization and adaptation. Lastly, although different from UFDA and semi-supervised domain adaptation (SSDA) settings, which use unlabeled samples (Saito et al., 2019; Kim & Kim, 2020), we conduct experiments comparing them in Appendix C.3 on DomainNet and C.12 for SSDA. Our auto-weighted methods have better or comparable performance across domains, especially for large shift cases.

## 6 CONCLUSION

We provide a theoretical framework that first formally defines the metrics to connect FDA settings with aggregation rules. We propose FedGP, a filtering-based aggregation rule via gradient projection, and develop the auto-weighted scheme that dynamically finds the best weights - both significantly improve the target performances and outperform various baselines. In the future, we plan to extend the current framework to perform FDA simultaneously on several source/target clients, explore the relationship between personalization and adaptation, as well as devise stronger aggregation rules.

ACKNOWLEDGMENTS

This work is partially supported by NSF III 2046795, IIS 1909577, CCF 1934986, NIH 1R01MH116226-01A, NIFA award 2020-67021-32799, the Alfred P. Sloan Foundation, and Google Inc.

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

# Appendix Contents

## A    Supplementary Theoretical Results

In this section, we provide theoretical results that are omitted in the main paper due to space limitations. Specifically, in the first four subsections, we provide proofs of our theorems. Further, in subsection A.5 we present an omitted discussion for our auto-weighting method, including how the estimators are constructed.

### A.1    Proof of Theorem 3.3

We first prove the following theorem where we study what happens when we do one step of optimization. While requiring minimal assumptions, this theorem shares the same intuition as Theorem 3.3 regarding the importance of the Delta error.

**Theorem A.1.** *Consider model parameter $\theta \sim \pi$ and an aggregation rule* $\mathtt{Aggr}(\cdot)$ *with step size* $\mu > 0$. *Define the updated parameter as*

$$\theta^+ := \theta - \mu \widehat{g}_{Aggr}(\theta).$$

*Assuming the gradient $\nabla\ell(\theta, z)$ is $\gamma$-Lipschitz in $\theta$ for any $z$, and let the step size $\mu \leq \frac{1}{\gamma}$, we have*

$$\mathbb{E}_{\widehat{\mathcal{D}}_T, \theta}[\ell_{\mathcal{D}_T}(\theta^+) - \ell_{\mathcal{D}_T}(\theta)] \leq -\tfrac{\mu}{2}(\|g_{\mathcal{D}_T}\|_\pi^2 - \Delta_{Aggr}^2).$$

*Proof.* Given any distribution data $\mathcal{D}$, we first prove that $\nabla\ell_{\mathcal{D}}$ is also $\gamma$-Lipschitz as below. For $\forall \theta_1, \theta_2 \in \Theta$:

$$
\begin{aligned}
\|\nabla\ell_{\mathcal{D}}(\theta_1) - \nabla\ell_{\mathcal{D}}(\theta_2)\| &= \|\mathbb{E}_{z \sim \mathcal{D}}[\nabla\ell(\theta_1, z) - \nabla\ell(\theta_2, z)]\| \\
&\leq \mathbb{E}_{z \sim \mathcal{D}}\|\nabla\ell(\theta_1, z) - \nabla\ell(\theta_2, z)\| && \text{(Jensen's inequality)} \\
&\leq \mathbb{E}_{z \sim \mathcal{D}}\gamma\|\theta_1 - \theta_2\| && (\nabla\ell(\cdot, z) \text{ is } \gamma\text{-Lipschitz}) \\
&= \gamma\|\theta_1 - \theta_2\|.
\end{aligned}
$$

Therefore, we know that $\ell_{\mathcal{D}_T}$ is $\gamma$-smooth. Conditioned on a $\theta$ and a $\widehat{\mathcal{D}}_T$, and apply the definition of smoothness we have

$$
\begin{aligned}
\ell_{\mathcal{D}_T}(\theta^+) - \ell_{\mathcal{D}_T}(\theta) &\leq \langle \nabla \ell_{\mathcal{D}_T}(\theta), \theta^+ - \theta \rangle + \frac{\gamma}{2}\|\theta^+ - \theta\|^2 \\
&= -\langle \nabla \ell_{\mathcal{D}_T}(\theta), \mu\widehat{g}_{\mathrm{Aggr}}(\theta)\rangle + \frac{\gamma}{2}\|\mu\widehat{g}_{\mathrm{Aggr}}(\theta)\|^2 \\
&= -\mu\langle \nabla \ell_{\mathcal{D}_T}(\theta), \widehat{g}_{\mathrm{Aggr}}(\theta) - \nabla \ell_{\mathcal{D}_T}(\theta) + \nabla \ell_{\mathcal{D}_T}(\theta)\rangle \\
&\quad + \frac{\gamma\mu^2}{2}\|\widehat{g}_{\mathrm{Aggr}}(\theta) - \nabla \ell_{\mathcal{D}_T}(\theta) + \nabla \ell_{\mathcal{D}_T}(\theta)\|^2 \\
&= -\mu(\langle \nabla \ell_{\mathcal{D}_T}(\theta), \widehat{g}_{\mathrm{Aggr}}(\theta) - \nabla \ell_{\mathcal{D}_T}(\theta)\rangle + \|\nabla \ell_{\mathcal{D}_T}(\theta)\|^2) \\
&\quad + \frac{\gamma\mu^2}{2}(\|\widehat{g}_{\mathrm{Aggr}}(\theta) - \nabla \ell_{\mathcal{D}_T}(\theta)\|^2 + \|\nabla \ell_{\mathcal{D}_T}(\theta)\|^2 + 2\langle \widehat{g}_{\mathrm{Aggr}}(\theta) - \nabla \ell_{\mathcal{D}_T}(\theta), \nabla \ell_{\mathcal{D}_T}(\theta)\rangle) \\
&= (\mu - \gamma\mu^2)(\langle \nabla \ell_{\mathcal{D}_T}(\theta), \nabla \ell_{\mathcal{D}_T}(\theta) - \widehat{g}_{\mathrm{Aggr}}(\theta)\rangle) \\
&\quad + (\frac{\gamma\mu^2}{2} - \mu)\|\nabla \ell_{\mathcal{D}_T}(\theta)\|^2 + \frac{\gamma\mu^2}{2}\|\widehat{g}_{\mathrm{Aggr}}(\theta) - \nabla \ell_{\mathcal{D}_T}(\theta)\|^2 \\
&\leq (\mu - \gamma\mu^2) \cdot \|\nabla \ell_{\mathcal{D}_T}(\theta)\| \cdot \|\widehat{g}_{\mathrm{Aggr}}(\theta) - \nabla \ell_{\mathcal{D}_T}(\theta)\| \qquad \text{(Cauchy–Schwarz inequality)} \\
&\quad + (\frac{\gamma\mu^2}{2} - \mu)\|\nabla \ell_{\mathcal{D}_T}(\theta)\|^2 + \frac{\gamma\mu^2}{2}\|\widehat{g}_{\mathrm{Aggr}}(\theta) - \nabla \ell_{\mathcal{D}_T}(\theta)\|^2. \\
&\leq \frac{\mu - \gamma\mu^2}{2}\left(\|\nabla \ell_{\mathcal{D}_T}(\theta)\|^2 + \|\widehat{g}_{\mathrm{Aggr}}(\theta) - \nabla \ell_{\mathcal{D}_T}(\theta)\|^2\right) \qquad \text{(AM-GM inequality)} \\
&\quad + (\frac{\gamma\mu^2}{2} - \mu)\|\nabla \ell_{\mathcal{D}_T}(\theta)\|^2 + \frac{\gamma\mu^2}{2}\|\widehat{g}_{\mathrm{Aggr}}(\theta) - \nabla \ell_{\mathcal{D}_T}(\theta)\|^2. \\
&= -\frac{\mu}{2}\left(\|\nabla \ell_{\mathcal{D}_T}(\theta)\|^2 - \|\widehat{g}_{\mathrm{Aggr}}(\theta) - \nabla \ell_{\mathcal{D}_T}(\theta)\|^2\right). \qquad (4)
\end{aligned}
$$

Additionally, note that the above two inequalities stand because: the step size $\mu \leq \frac{1}{\gamma}$ and thus $\mu - \gamma\mu^2 = \mu^2(\frac{1}{\mu} - \gamma) \geq 0$. Taking the expectation $\mathbb{E}_{\widehat{\mathcal{D}}_T, \theta}$ on both sides gives

$$
\mathbb{E}_{\widehat{\mathcal{D}}_T, \theta}[\ell_{\mathcal{D}_T}(\theta^+) - \ell_{\mathcal{D}_T}(\theta)] \leq -\frac{\mu}{2}\left(\mathbb{E}_{\widehat{\mathcal{D}}_T, \theta}[\|\nabla \ell_{\mathcal{D}_T}(\theta)\|^2] - \mathbb{E}_{\widehat{\mathcal{D}}_T, \theta}[\|\widehat{g}_{\mathrm{Aggr}}(\theta) - \nabla \ell_{\mathcal{D}_T}(\theta)\|^2]\right)
$$

Note that we denote $g_{\mathcal{D}_T} = \nabla \ell_{\mathcal{D}_T}$. Thus, with the $L^\pi$ norm notation we have

$$
\mathbb{E}_{\widehat{\mathcal{D}}_T, \theta}[\ell_{\mathcal{D}_T}(\theta^+) - \ell_{\mathcal{D}_T}(\theta)] \leq -\frac{\mu}{2}\left(\|g_{\mathcal{D}_T}\|_\pi^2 - \mathbb{E}_{\widehat{\mathcal{D}}_T}\|\widehat{g}_{\mathrm{Aggr}} - g_{\mathcal{D}_T}\|_\pi^2\right).
$$

Finally, by Definition 3.2 we can see $\Delta_{\mathrm{Aggr}}^2 = \mathbb{E}_{\widehat{\mathcal{D}}_T}\|\widehat{g}_{\mathrm{Aggr}} - g_{\mathcal{D}_T}\|_\pi^2$, which concludes the proof. $\quad\square$

**Theorem A.2** (Theorem 3.3 Restated). *For any probability measure $\pi$ over the parameter space, and an aggregation rule $\mathtt{Aggr}(\cdot)$ with step size $\mu > 0$. Given target dataset $\widehat{\mathcal{D}}_T$, update the parameter for $T$ steps as*

$$
\theta^{t+1} := \theta^t - \mu\widehat{g}_{Aggr}(\theta^t).
$$

*Assume the gradient $\nabla \ell(\theta, z)$ and $\widehat{g}_{Aggr}(\theta)$ is $\frac{\gamma}{2}$-Lipschitz in $\theta$ such that $\theta^t \to \widehat{\theta}_{Aggr}$. Then, given step size $\mu \leq \frac{1}{\gamma}$ and a small enough $\epsilon > 0$, with probability at least $1 - \delta$ we have*

$$
\|\nabla \ell_{\mathcal{D}_T}(\theta^T)\|^2 \leq \frac{1}{\delta^2}\left(\sqrt{C_\epsilon \cdot \Delta_{Aggr}^2} + \mathcal{O}(\epsilon)\right)^2 + \mathcal{O}\left(\frac{1}{T}\right) + \mathcal{O}(\epsilon),
$$

*where $C_\epsilon = \mathbb{E}_{\widehat{\mathcal{D}}_T}\left[1/\pi(B_\epsilon(\widehat{\theta}_{Aggr}))\right]^2$ and $B_\epsilon(\widehat{\theta}_{Aggr}) \subset \mathbb{R}^m$ is the ball with radius $\epsilon$ centered at $\widehat{\theta}_{Aggr}$. The $C_\epsilon$ measures how well the probability measure $\pi$ covers where the optimization goes, i.e., $\widehat{\theta}_{Aggr}$.*

*Proof.* We prove this theorem by starting from a seemingly mysterious place. However, its meaning is assured to be clear as we proceed.

Denote random function $\widehat{f} : \mathbb{R}^m \to \mathbb{R}_+$ as

$$\widehat{f}(\theta) = \|\widehat{g}_{\text{Aggr}}(\theta) - \nabla\ell_{\mathcal{D}_T}(\theta)\|, \qquad (5)$$

where the randomness comes from $\widehat{\mathcal{D}}_T$. Note that $\widehat{f}$ is $\gamma$-Lipschitz by assumption. Now we consider $B_\epsilon(\widehat{\theta}_{\text{Aggr}}) \subset \mathbb{R}^m$, i.e., the ball with radius $\epsilon$ centered at $\widehat{\theta}_{\text{Aggr}}$. Then, by $\gamma$-Lipschitzness we have

$$
\begin{aligned}
\mathbb{E}_{\theta\sim\pi}\widehat{f}(\theta) &= \int \widehat{f}(\theta)\,\mathrm{d}\pi(\theta) \\
&\geq \int_{B_\epsilon(\widehat{\theta}_{\text{Aggr}})} (\widehat{f}(\widehat{\theta}_{\text{Aggr}}) - \gamma\epsilon)\,\mathrm{d}\pi(\theta) \\
&= (\widehat{f}(\widehat{\theta}_{\text{Aggr}}) - \gamma\epsilon)\pi(B_\epsilon(\widehat{\theta}_{\text{Aggr}})).
\end{aligned}
$$

Therefore,

$$\widehat{f}(\widehat{\theta}_{\text{Aggr}}) \leq \frac{1}{\pi(B_\epsilon(\widehat{\theta}_{\text{Aggr}}))} \cdot \mathbb{E}_{\theta\sim\pi}\widehat{f}(\theta) + \mathcal{O}(\epsilon).$$

Taking expectation w.r.t. $\widehat{\mathcal{D}}_T$ on both sides, we have

$$
\begin{aligned}
\mathbb{E}_{\widehat{\mathcal{D}}_T}\widehat{f}(\widehat{\theta}_{\text{Aggr}}) &\leq \mathbb{E}_{\widehat{\mathcal{D}}_T}\left[\frac{1}{\pi(B_\epsilon(\widehat{\theta}_{\text{Aggr}}))} \cdot \mathbb{E}_{\theta\sim\pi}\widehat{f}(\theta)\right] + \mathcal{O}(\epsilon) \\
&\leq \sqrt{\mathbb{E}_{\widehat{\mathcal{D}}_T}\left[\frac{1}{\pi(B_\epsilon(\widehat{\theta}_{\text{Aggr}}))}\right]^2 \cdot \mathbb{E}_{\widehat{\mathcal{D}}_T}\left[\mathbb{E}_{\theta\sim\pi}\widehat{f}(\theta)\right]^2} + \mathcal{O}(\epsilon) && \text{(Cauchy-Schwarz)} \\
&= \sqrt{C_\epsilon \cdot \mathbb{E}_{\widehat{\mathcal{D}}_T}\left[\mathbb{E}_{\theta\sim\pi}\widehat{f}(\theta)\right]^2} + \mathcal{O}(\epsilon) && \text{(by definition of } C_\epsilon) \\
&\leq \sqrt{C_\epsilon \cdot \mathbb{E}_{\widehat{\mathcal{D}}_T}\mathbb{E}_{\theta\sim\pi}\left[\widehat{f}(\theta)\right]^2} + \mathcal{O}(\epsilon) && \text{(Jensen's inequality)} \\
&= \sqrt{C_\epsilon \cdot \Delta_{\text{Aggr}}^2} + \mathcal{O}(\epsilon)
\end{aligned}
$$

Therefore, by Markov's inequality, with probability at least $1 - \delta$ we have a sampled dataset $\widehat{\mathcal{D}}_T$ such that

$$\widehat{f}(\widehat{\theta}_{\text{Aggr}}) \leq \frac{1}{\delta}\mathbb{E}_{\widehat{\mathcal{D}}_T}\widehat{f}(\widehat{\theta}_{\text{Aggr}}) \leq \frac{1}{\delta}\sqrt{C_\epsilon \cdot \Delta_{\text{Aggr}}^2} + \mathcal{O}(\epsilon/\delta) \qquad (6)$$

Conditioned on such event, we proceed on to the optimization part.

Note that Theorem A.1 characterizes how the optimization works for one gradient update. Therefore, for any time step $t = 0, \ldots, T - 1$, we can apply (4) which only requires the Lipschitz assumption:

$$\ell_{\mathcal{D}_T}(\theta^{t+1}) - \ell_{\mathcal{D}_T}(\theta^t) \leq -\frac{\mu}{2}\left(\|\nabla\ell_{\mathcal{D}_T}(\theta^t)\|^2 - \|\widehat{g}_{\text{Aggr}}(\theta^t) - \nabla\ell_{\mathcal{D}_T}(\theta^t)\|^2\right).$$

On both sides, summing over $t = 0, \ldots, T - 1$ gives

$$\ell_{\mathcal{D}_T}(\theta^T) - \ell_{\mathcal{D}_T}(\theta^0) \leq -\frac{\mu}{2}\left(\sum_{t=0}^{T-1}\|\nabla\ell_{\mathcal{D}_T}(\theta^t)\|^2 - \sum_{t=0}^{T-1}\|\widehat{g}_{\text{Aggr}}(\theta^t) - \nabla\ell_{\mathcal{D}_T}(\theta^t)\|^2\right).$$

Dividing both sides by $T$, and with regular algebraic manipulation we derive

$$\frac{1}{T}\sum_{t=0}^{T-1}\|\nabla\ell_{\mathcal{D}_T}(\theta^t)\|^2 \leq \frac{2}{\mu T}(\ell_{\mathcal{D}_T}(\theta^0) - \ell_{\mathcal{D}_T}(\theta^T)) + \frac{1}{T}\sum_{t=0}^{T-1}\|\widehat{g}_{\text{Aggr}}(\theta^t) - \nabla\ell_{\mathcal{D}_T}(\theta^t)\|^2.$$

Note that we assume the loss function $\ell : \Theta \times \mathcal{Z} \to \mathbb{R}_+$ is non-negative (described at the beginning of Section 2). Thus, we have

$$\frac{1}{T}\sum_{t=0}^{T-1}\|\nabla\ell_{\mathcal{D}_T}(\theta^t)\|^2 \leq \frac{2\ell_{\mathcal{D}_T}(\theta^0)}{\mu T} + \frac{1}{T}\sum_{t=0}^{T-1}\|\widehat{g}_{\text{Aggr}}(\theta^t) - \nabla\ell_{\mathcal{D}_T}(\theta^t)\|^2. \qquad (7)$$

Note that we assume given $\widehat{\mathcal{D}}_T$ we have $\theta^t \to \widehat{\theta}_{\text{Aggr}}$. Therefore, for any $\epsilon > 0$ there exist $T_\epsilon$ such that

$$\forall t > T_\epsilon : \|\theta^t - \widehat{\theta}_{\text{Aggr}}\| < \epsilon. \tag{8}$$

This implies that $\forall t > T_\epsilon$:

$$\mu\|\widehat{g}_{\text{Aggr}}(\theta^t)\| = \|\theta^{t+1} - \widehat{\theta}_{\text{Aggr}} + \widehat{\theta}_{\text{Aggr}} - \theta^t\| \le \|\theta^{t+1} - \widehat{\theta}_{\text{Aggr}}\| + \|\widehat{\theta}_{\text{Aggr}} - \theta^t\| < 2\epsilon. \tag{9}$$

Moreover, (8) also implies $\forall t_1, t_2 > T_\epsilon$:

$$\|\nabla\ell_{\mathcal{D}_T}(\theta^{t_1}) - \nabla\ell_{\mathcal{D}_T}(\theta^{t_2})\| \le \gamma\|\theta^{t_1} - \theta^{t_2}\| \qquad (\gamma\text{-Lipschitzness})$$
$$< 2\epsilon. \tag{10}$$

The above inequality means that $\{\nabla\ell_{\mathcal{D}_T}(\theta^t)\}_t$ is a Cauchy sequence.

Now, let's get back to (7). For $\forall T > T_\epsilon$ we have

$$\frac{1}{T}\sum_{t=0}^{T-1}\|\nabla\ell_{\mathcal{D}_T}(\theta^t)\|^2 \le \frac{2\ell_{\mathcal{D}_T}(\theta^0)}{\mu T} + \frac{1}{T}\sum_{t=0}^{T_\epsilon-1}\|\widehat{g}_{\text{Aggr}}(\theta^t) - \nabla\ell_{\mathcal{D}_T}(\theta^t)\|^2 + \frac{1}{T}\sum_{t=T_\epsilon}^{T-1}\|\widehat{g}_{\text{Aggr}}(\theta^t) - \nabla\ell_{\mathcal{D}_T}(\theta^t)\|^2$$

$$= \mathcal{O}\left(\frac{1}{T}\right) + \frac{1}{T}\sum_{t=T_\epsilon}^{T-1}\|\widehat{g}_{\text{Aggr}}(\theta^t) - \nabla\ell_{\mathcal{D}_T}(\theta^t)\|^2$$

$$= \mathcal{O}\left(\frac{1}{T}\right) + \frac{1}{T}\sum_{t=T_\epsilon}^{T-1}\|\widehat{g}_{\text{Aggr}}(\theta^t) - \widehat{g}_{\text{Aggr}}(\widehat{\theta}_{\text{Aggr}}) + \widehat{g}_{\text{Aggr}}(\widehat{\theta}_{\text{Aggr}}) - \nabla\ell_{\mathcal{D}_T}(\theta^t)\|^2$$

$$\le \mathcal{O}\left(\frac{1}{T}\right) + \frac{1}{T}\sum_{t=T_\epsilon}^{T-1}\left(\|\widehat{g}_{\text{Aggr}}(\theta^t) - \widehat{g}_{\text{Aggr}}(\widehat{\theta}_{\text{Aggr}})\| + \|\widehat{g}_{\text{Aggr}}(\widehat{\theta}_{\text{Aggr}}) - \nabla\ell_{\mathcal{D}_T}(\theta^t)\|\right)^2$$
$$\text{(triangle inequality)}$$

$$= \mathcal{O}\left(\frac{1}{T}\right) + \frac{1}{T}\sum_{t=T_\epsilon}^{T-1}\left(\mathcal{O}(\epsilon) + \|\widehat{g}_{\text{Aggr}}(\widehat{\theta}_{\text{Aggr}}) - \nabla\ell_{\mathcal{D}_T}(\theta^t)\|\right)^2 \qquad \text{(by (9))}$$

$$= \mathcal{O}\left(\frac{1}{T}\right) + \mathcal{O}(\epsilon) + \frac{1}{T}\sum_{t=T_\epsilon}^{T-1}\left(\|\widehat{g}_{\text{Aggr}}(\widehat{\theta}_{\text{Aggr}}) - \nabla\ell_{\mathcal{D}_T}(\widehat{\theta}_{\text{Aggr}}) + \nabla\ell_{\mathcal{D}_T}(\widehat{\theta}_{\text{Aggr}}) - \nabla\ell_{\mathcal{D}_T}(\theta^t)\|\right)^2$$

$$\le \mathcal{O}\left(\frac{1}{T}\right) + \mathcal{O}(\epsilon) + \frac{1}{T}\sum_{t=T_\epsilon}^{T-1}\left(\|\widehat{g}_{\text{Aggr}}(\widehat{\theta}_{\text{Aggr}}) - \nabla\ell_{\mathcal{D}_T}(\widehat{\theta}_{\text{Aggr}})\| + \mathcal{O}(\epsilon)\right)^2$$
$$\text{(by (10))}$$

$$= \mathcal{O}\left(\frac{1}{T}\right) + \mathcal{O}(\epsilon) + \|\widehat{g}_{\text{Aggr}}(\widehat{\theta}_{\text{Aggr}}) - \nabla\ell_{\mathcal{D}_T}(\widehat{\theta}_{\text{Aggr}})\|^2 \tag{11}$$

Then, we can continue with what we have done at the beginning of the proof of this theorem:

$$(11) = \mathcal{O}\left(\frac{1}{T}\right) + \mathcal{O}(\epsilon) + f(\widehat{\theta}_{\text{Aggr}})^2 \qquad \text{(by (5))}$$

$$\le \mathcal{O}\left(\frac{1}{T}\right) + \mathcal{O}(\epsilon) + \left(\frac{1}{\delta}\sqrt{C_\epsilon \cdot \Delta_{\text{Aggr}}^2} + \mathcal{O}(\epsilon/\delta)\right)^2 \qquad \text{(by (6))}$$

Therefore, combining the above we finally have: for $\forall T > T_\epsilon$ with probability at least $1 - \delta$,

$$\frac{1}{T}\sum_{t=0}^{T-1}\|\nabla\ell_{\mathcal{D}_T}(\theta^t)\|^2 \le \mathcal{O}\left(\frac{1}{T}\right) + \mathcal{O}(\epsilon) + \frac{1}{\delta^2}\left(\sqrt{C_\epsilon \cdot \Delta_{\text{Aggr}}^2} + \mathcal{O}(\epsilon)\right)^2 \tag{12}$$

To complete the proof, let us investigate the left hand side.

$$
\frac{1}{T} \sum_{t=0}^{T-1} \|\nabla \ell_{\mathcal{D}_T}(\theta^t)\|^2 = \frac{1}{T} \sum_{t=0}^{T_\epsilon-1} \|\nabla \ell_{\mathcal{D}_T}(\theta^t)\|^2 + \frac{1}{T} \sum_{t=T_\epsilon}^{T-1} \|\nabla \ell_{\mathcal{D}_T}(\theta^t)\|^2
$$

$$
= \mathcal{O}\left(\frac{1}{T}\right) + \frac{1}{T} \sum_{t=T_\epsilon}^{T-1} \|\nabla \ell_{\mathcal{D}_T}(\theta^t)\|^2
$$

$$
\geq \mathcal{O}\left(\frac{1}{T}\right) + \frac{1}{T} \sum_{t=T_\epsilon}^{T-1} \left( \|\nabla \ell_{\mathcal{D}_T}(\theta^t) - \nabla \ell_{\mathcal{D}_T}(\theta^T)\| - \|\nabla \ell_{\mathcal{D}_T}(\theta^T)\| \right)^2
$$

$$
\text{(triangle inequality)}
$$

$$
= \mathcal{O}\left(\frac{1}{T}\right) + \frac{1}{T} \sum_{t=T_\epsilon}^{T-1} \left( \mathcal{O}(\epsilon) + \|\nabla \ell_{\mathcal{D}_T}(\theta^T)\| \right)^2 \qquad \text{(by (10))}
$$

$$
= \mathcal{O}\left(\frac{1}{T}\right) + \mathcal{O}(\epsilon) + \|\nabla \ell_{\mathcal{D}_T}(\theta^T)\|^2. \tag{13}
$$

Combining (12) and (13), we finally have

$$
\|\nabla \ell_{\mathcal{D}_T}(\theta^T)\|^2 \leq \mathcal{O}\left(\frac{1}{T}\right) + \mathcal{O}(\epsilon) + \frac{1}{\delta^2} \left( \sqrt{C_\epsilon \cdot \Delta_{\text{Aggr}}^2} + \mathcal{O}(\epsilon) \right)^2,
$$

which completes the proof.

$\square$

## A.2 Proof of Theorem 3.6

**Theorem A.3** (Theorem 3.6 Restated). *Consider any aggregation rule $\mathrm{Aggr}(\cdot)$ in the form of $\widehat{g}_{Aggr} = \frac{1}{N} \sum_{i \in [N]} F_{Aggr}[g_{\widehat{\mathcal{D}}_T}, g_{\mathcal{D}_{S_i}}]$, i.e., the aggregation rule is defined by a mapping $F_{Aggr} : L^\pi \times L^\pi \to L^\pi$. If $F_{Aggr}$ is affine w.r.t. to its first argument (i.e., the target gradient function), and $\forall g \in L^\pi : F_{Aggr}[g, g] = g$, and the linear mapping associated with $F_{Aggr}$ has its eigenvalue bounded in $[\lambda_{min}, \lambda_{max}]$, then for any source and target distributions $\{\mathcal{D}_{S_i}\}_{i \in [N]}, \mathcal{D}_T, \widehat{\mathcal{D}}_T$ we have $\Delta_{Aggr}^2 \leq \frac{1}{N} \sum_{i \in [N]} \Delta_{Aggr, \mathcal{D}_{S_i}}^2$, where*

$$
\Delta_{Aggr, \mathcal{D}_{S_i}}^2 \leq \max\{\lambda_{max}^2, \lambda_{min}^2\} \cdot \frac{\sigma_\pi^2(z)}{n} + \max\{(1-\lambda_{max})^2, (1-\lambda_{min})^2\} \cdot d_\pi(\mathcal{D}_{S_i}, \mathcal{D}_T)^2
$$

*Proof.* Let's begin from the definition of the Delta error.

$$
\Delta_{Aggr}^2 = \mathbb{E}_{\widehat{\mathcal{D}}_T} \|g_{\mathcal{D}_T} - \widehat{g}_{\text{Aggr}}\|_\pi^2
$$

$$
= \mathbb{E}_{\widehat{\mathcal{D}}_T} \left\| g_{\mathcal{D}_T} - \frac{1}{N} \sum_{i \in [N]} F_{\text{Aggr}}[g_{\widehat{\mathcal{D}}_T}, g_{\mathcal{D}_{S_i}}] \right\|_\pi^2
$$

$$
= \mathbb{E}_{\widehat{\mathcal{D}}_T} \left\| \frac{1}{N} \sum_{i \in [N]} \left( g_{\mathcal{D}_T} - F_{\text{Aggr}}[g_{\widehat{\mathcal{D}}_T}, g_{\mathcal{D}_{S_i}}] \right) \right\|_\pi^2
$$

$$
\leq \frac{1}{N} \sum_{i \in [N]} \mathbb{E}_{\widehat{\mathcal{D}}_T} \left\| g_{\mathcal{D}_T} - F_{\text{Aggr}}[g_{\widehat{\mathcal{D}}_T}, g_{\mathcal{D}_{S_i}}] \right\|_\pi^2 \qquad \text{(Jensen's Inequality)}
$$

$$
= \frac{1}{N} \sum_{i \in [N]} \Delta_{Aggr, \mathcal{D}_{S_i}}^2, \tag{14}
$$

where we denote

$$
\Delta_{Aggr, \mathcal{D}_{S_i}}^2 := \mathbb{E}_{\widehat{\mathcal{D}}_T} \left\| g_{\mathcal{D}_T} - F_{\text{Aggr}}[g_{\widehat{\mathcal{D}}_T}, g_{\mathcal{D}_{S_i}}] \right\|_\pi^2,
$$

We continue on upper bounding this term. Noting that $F_{\text{Aggr}}[\cdot, \cdot]$ is affine in its first argument, we can denote

$$\bar{g} := \mathbb{E}_{\widehat{\mathcal{D}}_T} F_{\text{Aggr}}[g_{\widehat{\mathcal{D}}_T}, g_{\mathcal{D}_{S_i}}] = F_{\text{Aggr}}[\mathbb{E}_{\widehat{\mathcal{D}}_T}[g_{\widehat{\mathcal{D}}_T}], g_{\mathcal{D}_{S_i}}] = F_{\text{Aggr}}[g_{\mathcal{D}_T}, g_{\mathcal{D}_{S_i}}]. \qquad (15)$$

Therefore,

$$
\begin{aligned}
\Delta^2_{Aggr, \mathcal{D}_{S_i}} &= \mathbb{E}_{\widehat{\mathcal{D}}_T} \left\| g_{\mathcal{D}_T} - \bar{g} + \bar{g} - F_{\text{Aggr}}[g_{\widehat{\mathcal{D}}_T}, g_{\mathcal{D}_{S_i}}] \right\|^2_\pi \\
&= \|g_{\mathcal{D}_T} - \bar{g}\|^2_\pi + \mathbb{E}_{\widehat{\mathcal{D}}_T} \left\| \bar{g} - F_{\text{Aggr}}[g_{\widehat{\mathcal{D}}_T}, g_{\mathcal{D}_{S_i}}] \right\|^2_\pi + 2\mathbb{E}_{\widehat{\mathcal{D}}_T} \langle g_{\mathcal{D}_T} - \bar{g}, \bar{g} - F_{\text{Aggr}}[g_{\widehat{\mathcal{D}}_T}, g_{\mathcal{D}_{S_i}}] \rangle_\pi \\
&= \|g_{\mathcal{D}_T} - \bar{g}\|^2_\pi + \mathbb{E}_{\widehat{\mathcal{D}}_T} \left\| \bar{g} - F_{\text{Aggr}}[g_{\widehat{\mathcal{D}}_T}, g_{\mathcal{D}_{S_i}}] \right\|^2_\pi + 2 \langle g_{\mathcal{D}_T} - \bar{g}, \bar{g} - \mathbb{E}_{\widehat{\mathcal{D}}_T} F_{\text{Aggr}}[g_{\widehat{\mathcal{D}}_T}, g_{\mathcal{D}_{S_i}}] \rangle_\pi \\
&= \|g_{\mathcal{D}_T} - \bar{g}\|^2_\pi + \mathbb{E}_{\widehat{\mathcal{D}}_T} \left\| \bar{g} - F_{\text{Aggr}}[g_{\widehat{\mathcal{D}}_T}, g_{\mathcal{D}_{S_i}}] \right\|^2_\pi + 2 \langle g_{\mathcal{D}_T} - \bar{g}, \bar{g} - \bar{g} \rangle_\pi \qquad \text{(by (15))} \\
&= \|g_{\mathcal{D}_T} - \bar{g}\|^2_\pi + \mathbb{E}_{\widehat{\mathcal{D}}_T} \left\| \bar{g} - F_{\text{Aggr}}[g_{\widehat{\mathcal{D}}_T}, g_{\mathcal{D}_{S_i}}] \right\|^2_\pi. \qquad (16)
\end{aligned}
$$

The above derivation is based on properties of inner product. Next, we deal with the two terms in (16). Denote $G : L^\pi \to L^\pi$ as the linear mapping associated with the affine mapping $F_{\text{Aggr}}[\cdot, g_{\mathcal{D}_{S_i}}]$. By definition, $\forall g_1, g_2 \in L^\pi$:

$$F_{\text{Aggr}}[g_1, g_{\mathcal{D}_{S_i}}] - F_{\text{Aggr}}[g_2, g_{\mathcal{D}_{S_i}}] = G[g_1] - G[g_2] = G[g_1 - g_2]. \qquad (17)$$

For the first term in (16), we can do a similar trick as the following.

$$
\begin{aligned}
\|g_{\mathcal{D}_T} - \bar{g}\|^2_\pi &= \left\| g_{\mathcal{D}_T} - g_{\mathcal{D}_{S_i}} + g_{\mathcal{D}_{S_i}} - \bar{g} \right\|^2_\pi \\
&= \left\| g_{\mathcal{D}_T} - g_{\mathcal{D}_{S_i}} + g_{\mathcal{D}_{S_i}} - F_{\text{Aggr}}[g_{\mathcal{D}_T}, g_{\mathcal{D}_{S_i}}] \right\|^2_\pi \qquad \text{(by (15))} \\
&= \left\| g_{\mathcal{D}_T} - g_{\mathcal{D}_{S_i}} + F_{\text{Aggr}}[g_{\mathcal{D}_{S_i}}, g_{\mathcal{D}_{S_i}}] - F_{\text{Aggr}}[g_{\mathcal{D}_T}, g_{\mathcal{D}_{S_i}}] \right\|^2_\pi \qquad (18) \\
&= \left\| g_{\mathcal{D}_T} - g_{\mathcal{D}_{S_i}} + G[g_{\mathcal{D}_{S_i}} - g_{\mathcal{D}_T}] \right\|^2_\pi \qquad \text{(by (17))} \\
&= \left\| (I - G)[g_{\mathcal{D}_{S_i}} - g_{\mathcal{D}_T}] \right\|^2_\pi \qquad \text{(\textit{I} stands for identity map)} \\
&\leq \max\{(1 - \lambda_{max})^2, (1 - \lambda_{min})^2\} \|g_{\mathcal{D}_{S_i}} - g_{\mathcal{D}_T}\|^2_\pi \\
&= \max\{(1 - \lambda_{max})^2, (1 - \lambda_{min})^2\} d_\pi(\mathcal{D}_{S_i}, \mathcal{D}_T)^2.
\end{aligned}
$$

where (18) is by the identity assumption of $F_{\text{Aggr}}$. This assumption is valid in then sense that: if all of the inputs to an federated aggregation rule are the same thing, the aggregation rule should output the same thing.

This upper bounds the first term in (16), and lets move on to the second term in (16) as the following.

$$
\begin{aligned}
\mathbb{E}_{\widehat{\mathcal{D}}_T} \left\| \bar{g} - F_{\text{Aggr}}[g_{\widehat{\mathcal{D}}_T}, g_{\mathcal{D}_{S_i}}] \right\|^2_\pi &= \mathbb{E}_{\widehat{\mathcal{D}}_T} \left\| F_{\text{Aggr}}[g_{\mathcal{D}_T}, g_{\mathcal{D}_{S_i}}] - F_{\text{Aggr}}[g_{\widehat{\mathcal{D}}_T}, g_{\mathcal{D}_{S_i}}] \right\|^2_\pi \qquad \text{(by (15))} \\
&= \mathbb{E}_{\widehat{\mathcal{D}}_T} \left\| G[g_{\mathcal{D}_T} - g_{\widehat{\mathcal{D}}_T}] \right\|^2_\pi \qquad \text{(by (17))} \\
&\leq \max\{\lambda^2_{max}, \lambda^2_{min}\} \mathbb{E}_{\widehat{\mathcal{D}}_T} \left\| g_{\mathcal{D}_T} - g_{\widehat{\mathcal{D}}_T} \right\|^2_\pi \\
&= \max\{\lambda^2_{max}, \lambda^2_{min}\} \sigma^2_\pi(\widehat{\mathcal{D}}_T). \qquad \text{(by Definition 3.5)}
\end{aligned}
$$

Combining the above, we finally have

$$\Delta^2_{Aggr, \mathcal{D}_{S_i}} = (16) \leq \max\{\lambda^2_{max}, \lambda^2_{min}\} \sigma^2_\pi(\widehat{\mathcal{D}}_T) + \max\{(1 - \lambda_{max})^2, (1 - \lambda_{min})^2\} d_\pi(\mathcal{D}_{S_i}, \mathcal{D}_T)^2.$$

Putting (14) and the above inequality together, and noting that $\sigma^2_\pi(\widehat{\mathcal{D}}_T) = \frac{\sigma^2_\pi(z)}{n}$, we can see that the proof is complete. $\qquad \square$

### A.3 PROOF OF THEOREM 4.2

Next, we prove the following theorem.

**Theorem A.4** (Theorem 4.2 Restated). *Consider* FedDA. *Given the target domain $\widehat{\mathcal{D}}_T$ and $N$ source domains $\mathcal{D}_{S_1}, \ldots, \mathcal{D}_{S_N}$.*

$$\Delta^2_{FedDA} \leq \frac{1}{N} \sum_{i=1}^{N} \Delta^2_{FedDA, S_i}, \quad \text{where} \ \Delta^2_{FedDA, S_i} = (1-\beta_i)^2 \sigma^2_\pi(\widehat{\mathcal{D}}_T) + \beta_i^2 d^2_\pi(\mathcal{D}_{S_i}, \mathcal{D}_T)$$

*is the Delta error when only considering $\mathcal{D}_{S_i}$ as the source domain.*

*Proof.* Recall

$$\widehat{g}_{\text{FedDA}} = \sum_{i=1}^{N} \frac{1}{N} \left( (1-\beta_i) g_{\widehat{\mathcal{D}}_T} + \beta_i g_{\mathcal{D}_{S_i}} \right).$$

Thus,

$$\Delta^2_{\text{FedDA}} = \mathbb{E}_{\widehat{\mathcal{D}}_T} \| g_{\mathcal{D}_T} - \widehat{g}_{\text{FedDA}} \|^2_\pi = \mathbb{E}_{\widehat{\mathcal{D}}_T} \| g_{\mathcal{D}_T} - \sum_{i=1}^{N} \frac{1}{N} \left( (1-\beta_i) g_{\widehat{\mathcal{D}}_T} + \beta_i g_{\mathcal{D}_{S_i}} \right) \|^2_\pi$$

$$= \mathbb{E}_{\widehat{\mathcal{D}}_T} \| \sum_{i=1}^{N} \frac{1}{N} \left( g_{\mathcal{D}_T} - (1-\beta_i) g_{\widehat{\mathcal{D}}_T} + \beta_i g_{\mathcal{D}_{S_i}} \right) \|^2_\pi$$

$$\leq \sum_{i=1}^{N} \frac{1}{N} \mathbb{E}_{\widehat{\mathcal{D}}_T} \| g_{\mathcal{D}_T} - (1-\beta_i) g_{\widehat{\mathcal{D}}_T} + \beta_i g_{\mathcal{D}_{S_i}} \|^2_\pi \tag{19}$$

where the inequality is derived by Jensen's inequality. Next, we prove for each $i \in [N]$:

$$\mathbb{E}_{\widehat{\mathcal{D}}_T} \| g_{\mathcal{D}_T} - (1-\beta_i) g_{\widehat{\mathcal{D}}_T} + \beta_i g_{\mathcal{D}_{S_i}} \|^2_\pi = \mathbb{E}_{\widehat{\mathcal{D}}_T} \| (1-\beta_i)(g_{\mathcal{D}_T} - g_{\widehat{\mathcal{D}}_T}) + \beta_i(g_{\mathcal{D}_T} - g_{\mathcal{D}_{S_i}}) \|^2_\pi$$

$$= (1-\beta_i)^2 \mathbb{E}_{\widehat{\mathcal{D}}_T} \| g_{\mathcal{D}_T} - g_{\widehat{\mathcal{D}}_T} \|^2_\pi + \beta_i^2 \| g_{\mathcal{D}_T} - g_{\mathcal{D}_{S_i}} \|^2_\pi$$

$$\qquad + 2(1-\beta_i)\beta_i \mathbb{E}_{\widehat{\mathcal{D}}_T} \langle g_{\mathcal{D}_T} - g_{\widehat{\mathcal{D}}_T}, g_{\mathcal{D}_T} - g_{\mathcal{D}_{S_i}} \rangle_\pi$$

$$= (1-\beta_i)^2 \sigma^2_\pi(\widehat{\mathcal{D}}_T) + \beta_i^2 d^2_\pi(\mathcal{D}_S, \mathcal{D}_T) + 2(1-\beta_i)\beta_i \langle \mathbb{E}_{\widehat{\mathcal{D}}_T}[g_{\mathcal{D}_T} - g_{\widehat{\mathcal{D}}_T}], g_{\mathcal{D}_T} - g_{\mathcal{D}_{S_i}} \rangle_\pi$$

$$= (1-\beta_i)^2 \sigma^2_\pi(\widehat{\mathcal{D}}_T) + \beta_i^2 d^2_\pi(\mathcal{D}_{S_i}, \mathcal{D}_T)$$

$$= \Delta^2_{\text{FedDA}, S_i}.$$

Plugging the above equation into (19) gives the theorem. $\qquad \square$

### A.4 PROOF OF THEOREM 4.4

**Theorem A.5** (Theorem 4.4 Rigorously). *Consider* FedGP. *Given the target domain $\widehat{\mathcal{D}}_T$ and $N$ source domains $\mathcal{D}_{S_1}, \ldots, \mathcal{D}_{S_N}$.*

$$\Delta^2_{FedGP} \leq \frac{1}{N} \sum_{i=1}^{N} \Delta^2_{FedGP, S_i},$$

*where*

$$\Delta^2_{FedGP, S_i} = (1-\beta_i)^2 \sigma^2_\pi(\widehat{\mathcal{D}}_T) + \beta_i^2 \mathbb{E}_{\theta, \widehat{\mathcal{D}}_T} [\widehat{\delta}_i(\theta) \tau_i^2(\theta) \| g_{\mathcal{D}_T}(\theta) - g_{\mathcal{D}_{S_i}}(\theta) \|^2]$$

$$+ (2\beta_i - \beta_i^2) \mathbb{E}_{\theta, \widehat{\mathcal{D}}_T} \widehat{\delta}_i(\theta) \langle g_{\widehat{\mathcal{D}}_T}(\theta) - g_{\mathcal{D}_T}(\theta), u_{\mathcal{D}_{S_i}}(\theta) \rangle^2$$

$$+ 2\beta_i(1-\beta_i) \mathbb{E}_{\theta, \widehat{\mathcal{D}}_T} [\widehat{\delta}_i(\theta) \langle g_{\widehat{\mathcal{D}}_T}(\theta), g_{\mathcal{D}_{S_i}}(\theta) \rangle \cdot \langle g_{\widehat{\mathcal{D}}_T}(\theta) - g_{\mathcal{D}_T}(\theta), u_{\mathcal{D}_{S_i}}(\theta) \rangle]$$

$$+ \beta_i^2 \mathbb{E}_{\theta, \widehat{\mathcal{D}}_T} [(1 - \widehat{\delta}_i(\theta)) \| g_{\mathcal{D}_T}(\theta) \|^2].$$

*In the above equation, $\widehat{\delta}_i(\theta) := \mathbf{1}(\langle g_{\widehat{\mathcal{D}}_T}(\theta), g_{\mathcal{D}_{S_i}}(\theta) \rangle > 0)$ is the indicator function and it is 1 if the condition is satisfied. $\tau_i(\theta) := \frac{\| g_{\mathcal{D}_T}(\theta) \sin \rho_i(\theta) \|}{\| g_{\mathcal{D}_{S_i}}(\theta) - g_{\mathcal{D}_T}(\theta) \|}$ where $\rho_i(\theta)$ is the angle between $g_{\mathcal{D}_{S_i}}(\theta)$ and $g_{\mathcal{D}_T}(\theta)$. Moreover, $u_{\mathcal{D}_{S_i}}(\theta) := g_{\mathcal{D}_{S_i}}(\theta) / \| g_{\mathcal{D}_{S_i}}(\theta) \|$.*

*Proof.* Recall that

$$\widehat{g}_{\texttt{FedGP}}(\theta) = \sum_{i=1}^{N} \frac{1}{N} \left( (1-\beta_i) g_{\widehat{\mathcal{D}}_T}(\theta) + \beta_i \texttt{Proj}_+ (g_{\widehat{\mathcal{D}}_T}(\theta) | g_{\mathcal{D}_{S_i}}(\theta)) \right) = \sum_{i=1}^{N} \frac{1}{N} \widehat{g}_{\texttt{FedGP}, \mathcal{D}_{S_i}}(\theta),$$

where we denote

$$\widehat{g}_{\texttt{FedGP}, \mathcal{D}_{S_i}}(\theta) := \left( (1-\beta_i) g_{\widehat{\mathcal{D}}_T}(\theta) + \beta_i \texttt{Proj}_+ (g_{\widehat{\mathcal{D}}_T}(\theta) | g_{\mathcal{D}_{S_i}}(\theta)) \right).$$

Then, we can derive:

$$\Delta^2_{\texttt{FedGP}} = \mathbb{E}_{\widehat{\mathcal{D}}_T} \| g_{\mathcal{D}_T} - \widehat{g}_{\texttt{FedGP}} \|_\pi^2 = \mathbb{E}_{\widehat{\mathcal{D}}_T, \theta} \| g_{\mathcal{D}_T}(\theta) - \widehat{g}_{\texttt{FedGP}}(\theta) \|^2$$

$$= \mathbb{E}_{\widehat{\mathcal{D}}_T, \theta} \| g_{\mathcal{D}_T}(\theta) - \sum_{i=1}^{N} \frac{1}{N} \widehat{g}_{\texttt{FedGP}, \mathcal{D}_{S_i}}(\theta) \|^2$$

$$= \mathbb{E}_{\widehat{\mathcal{D}}_T, \theta} \| \sum_{i=1}^{N} \frac{1}{N} (g_{\mathcal{D}_T}(\theta) - \widehat{g}_{\texttt{FedGP}, \mathcal{D}_{S_i}}(\theta)) \|^2$$

$$\leq \sum_{i=1}^{N} \frac{1}{N} \mathbb{E}_{\widehat{\mathcal{D}}_T, \theta} \| g_{\mathcal{D}_T}(\theta) - \widehat{g}_{\texttt{FedGP}, \mathcal{D}_{S_i}}(\theta) \|^2 \tag{20}$$

where the inequality is derived by Jensen's inequality.

Next, we show $\mathbb{E}_{\widehat{\mathcal{D}}_T, \theta} \| g_{\mathcal{D}_T}(\theta) - \widehat{g}_{\texttt{FedGP}, \mathcal{D}_{S_i}}(\theta) \|^2 = \Delta^2_{\texttt{FedGP}, S_i}$. First, we simplify the notation. Recall the definition of $\widehat{g}_{\texttt{FedGP}, \mathcal{D}_{S_i}}$ is that

$$\widehat{g}_{\texttt{FedGP}, \mathcal{D}_{S_i}}(\theta) = (1-\beta_i) g_{\widehat{\mathcal{D}}_T}(\theta) + \beta_i \max\{\langle g_{\widehat{\mathcal{D}}_T}(\theta), g_{\mathcal{D}_{S_i}} \rangle, 0\} g_{\mathcal{D}_{S_i}(\theta)} / \| g_{\mathcal{D}_{S_i}}(\theta) \|^2$$

$$= (1-\beta_i) g_{\widehat{\mathcal{D}}_T}(\theta) + \beta_i \widehat{\delta}_i(\theta) \langle g_{\widehat{\mathcal{D}}_T}(\theta), u_{\mathcal{D}_{S_i}}(\theta) \rangle u_{\mathcal{D}_{S_i}}(\theta).$$

Let us fix $\theta$ and then further simplify the notation by denoting

$$\hat{v} := g_{\widehat{\mathcal{D}}_T}(\theta)$$
$$v := g_{\mathcal{D}_T}(\theta)$$
$$u := u_{\mathcal{D}_{S_i}}(\theta)$$
$$\hat{\delta} := \hat{\delta}_i(\theta)$$

Therefore, we have $\widehat{g}_{\texttt{FedGP}, \mathcal{D}_{S_i}}(\theta) = (1-\beta_i)\hat{v} + \beta_i \hat{\delta} \langle \hat{v}, u \rangle u$.

Therefore, with the simplified notation,

$$\mathbb{E}_{\widehat{\mathcal{D}}_T} \| g_{\mathcal{D}_T}(\theta) - \widehat{g}_{\texttt{FedGP}, \mathcal{D}_{S_i}}(\theta) \|^2 = \mathbb{E}_{\widehat{\mathcal{D}}_T} \| (1-\beta_i)\hat{v} + \beta_i \hat{\delta} \langle \hat{v}, u \rangle u - v \|^2$$

$$= \mathbb{E}_{\widehat{\mathcal{D}}_T} \| \beta_i (\hat{\delta} \langle \hat{v}, u \rangle u - v) + (1-\beta_i)(\hat{v} - v) \|^2$$

$$= \mathbb{E}_{\widehat{\mathcal{D}}_T} \| \beta_i \hat{\delta} \langle \hat{v} - v, u \rangle u + \beta_i (\hat{\delta} \langle v, u \rangle u - v) + (1-\beta_i)(\hat{v} - v) \|^2$$

$$= \beta_i^2 \mathbb{E}_{\widehat{\mathcal{D}}_T} [\hat{\delta} \langle \hat{v} - v, u \rangle^2] + \beta_i^2 \mathbb{E}_{\widehat{\mathcal{D}}_T} [\| \hat{\delta} \langle v, u \rangle u - v \|^2] + (1-\beta_i)^2 \mathbb{E}_{\widehat{\mathcal{D}}_T} [\| \hat{v} - v \|^2]$$

$$+ 2\beta_i(1-\beta_i) \mathbb{E}_{\widehat{\mathcal{D}}_T} [\hat{\delta} \langle \hat{v} - v, u \rangle^2] + 2\beta_i(1-\beta_i) \mathbb{E}_{\widehat{\mathcal{D}}_T} [\hat{\delta} \langle v, u \rangle \langle u, \hat{v} - v \rangle]$$

$$- 2\beta_i^2 \mathbb{E}[\hat{\delta}(1-\hat{\delta}) \langle \hat{v} - v, u \rangle \langle v, u \rangle] \qquad \text{(expanding the squared norm)}$$

$$= (2\beta_i - \beta_i^2) \mathbb{E}_{\widehat{\mathcal{D}}_T} [\hat{\delta} \langle \hat{v} - v, u \rangle^2] + \beta_i^2 \underbrace{\mathbb{E}_{\widehat{\mathcal{D}}_T} [\| \hat{\delta} \langle v, u \rangle u - v \|^2]}_{A} + (1-\beta_i)^2 \mathbb{E}_{\widehat{\mathcal{D}}_T} [\| \hat{v} - v \|^2]$$

$$+ 2\beta_i(1-\beta_i) \mathbb{E}_{\widehat{\mathcal{D}}_T} [\hat{\delta} \langle v, u \rangle \langle u, \hat{v} - v \rangle] - 2\beta_i^2 \underbrace{\mathbb{E}[\hat{\delta}(1-\hat{\delta}) \langle \hat{v} - v, u \rangle \langle v, u \rangle]}_{B},$$

where the last equality is by merging the similar terms. Next, we deal with the terms $A$ and $B$.

Let us start from the term $B$. Noting that $\hat{\delta}$ is either $0$ or $1$, we can see that $\hat{\delta}(1-\hat{\delta}) = 0$. Therefore, $B = 0$.

As for the term $A$, noting that $\hat{\delta}^2 = \hat{\delta}$, expanding the squared term we have:

$$
\begin{aligned}
A &= \mathbb{E}_{\widehat{\mathcal{D}}_T}[\|\hat{\delta}\langle v, u\rangle u - v\|^2] = \mathbb{E}_{\widehat{\mathcal{D}}_T}[\|\hat{\delta}(\langle v, u\rangle u - v) - (1 - \hat{\delta})v\|^2] \\
&= \mathbb{E}_{\widehat{\mathcal{D}}_T}[\hat{\delta}\|\langle v, u\rangle u - v\|^2] + \mathbb{E}_{\widehat{\mathcal{D}}_T}[(1 - \hat{\delta})\|v\|^2] - 2\mathbb{E}_{\widehat{\mathcal{D}}_T}[\hat{\delta}(1 - \hat{\delta})\langle\langle v, u\rangle u - v, v\rangle] \\
&= \mathbb{E}_{\widehat{\mathcal{D}}_T}[\hat{\delta}\|\langle v, u\rangle u - v\|^2] + \mathbb{E}_{\widehat{\mathcal{D}}_T}[(1 - \hat{\delta})\|v\|^2].
\end{aligned}
$$

Therefore, combining the above, we have

$$
\begin{aligned}
\mathbb{E}_{\widehat{\mathcal{D}}_T}\|g_{\mathcal{D}_T}(\theta) - \widehat{g}_{\text{FedGP}, \mathcal{D}_{S_i}}(\theta)\|^2 = {}& (1 - \beta_i)^2\mathbb{E}_{\widehat{\mathcal{D}}_T}[\|\hat{v} - v\|^2] + \beta_i^2\mathbb{E}_{\widehat{\mathcal{D}}_T}[\hat{\delta}\|\langle v, u\rangle u - v\|^2] \\
& + (2\beta_i - \beta_i^2)\mathbb{E}_{\widehat{\mathcal{D}}_T}[\hat{\delta}\langle\hat{v} - v, u\rangle^2] \\
& + 2\beta_i(1 - \beta_i)\mathbb{E}_{\widehat{\mathcal{D}}_T}[\hat{\delta}\langle v, u\rangle\langle u, \hat{v} - v\rangle] + \beta_i^2\mathbb{E}_{\widehat{\mathcal{D}}_T}[(1 - \hat{\delta})\|v\|^2].
\end{aligned}
$$

Let us give an alternative form for $\|\langle v, u\rangle u - v\|^2$, as we aim to connect this term to $\|u - v\|$ which would become the domain-shift. Note that $\|\langle v, u\rangle u - v\|$ is the distance between $v$ and its projection to $u$. We can see that $\|\langle v, u\rangle u - v\| = \|u - v\|\frac{\|\langle v, u\rangle u - v\|}{\|u - v\|} = \|u - v\|\tau_i(\theta)$. Therefore, plugging this in, we have

$$
\begin{aligned}
\mathbb{E}_{\widehat{\mathcal{D}}_T}\|g_{\mathcal{D}_T}(\theta) - \widehat{g}_{\text{FedGP}, \mathcal{D}_{S_i}}(\theta)\|^2 = {}& (1 - \beta_i)^2\mathbb{E}_{\widehat{\mathcal{D}}_T}[\|\hat{v} - v\|^2] + \beta_i^2\mathbb{E}_{\widehat{\mathcal{D}}_T}[\hat{\delta}\|u - v\|^2\tau_i^2(\theta)] \\
& + (2\beta_i - \beta_i^2)\mathbb{E}_{\widehat{\mathcal{D}}_T}[\hat{\delta}\langle\hat{v} - v, u\rangle^2] \\
& + 2\beta_i(1 - \beta_i)\mathbb{E}_{\widehat{\mathcal{D}}_T}[\hat{\delta}\langle v, u\rangle\langle u, \hat{v} - v\rangle] + \beta_i^2\mathbb{E}_{\widehat{\mathcal{D}}_T}[(1 - \hat{\delta})\|v\|^2]
\end{aligned}
$$

Writing the abbreviations $u, v, \hat{v}, \hat{\delta}$ into their original forms and taking expectation over $\theta$ on the both side we can derive:

$$
\begin{aligned}
\mathbb{E}_{\widehat{\mathcal{D}}_T, \theta}\|g_{\mathcal{D}_T}(\theta) - \widehat{g}_{\text{FedGP}, \mathcal{D}_{S_i}}(\theta)\|^2 = {}& (1 - \beta_i)^2\sigma_\pi^2(\widehat{\mathcal{D}}_T) + \beta_i^2\mathbb{E}_{\theta, \widehat{\mathcal{D}}_T}[\widehat{\delta}_i(\theta)\tau_i^2(\theta)\|g_{\mathcal{D}_T}(\theta) - g_{\mathcal{D}_{S_i}}(\theta)\|^2] \\
& + (2\beta_i - \beta_i^2)\mathbb{E}_{\theta, \widehat{\mathcal{D}}_T}\widehat{\delta}_i(\theta)\langle g_{\widehat{\mathcal{D}}_T}(\theta) - g_{\mathcal{D}_T}(\theta), u_{\mathcal{D}_{S_i}}(\theta)\rangle^2 \\
& + 2\beta_i(1 - \beta_i)\mathbb{E}_{\theta, \widehat{\mathcal{D}}_T}[\widehat{\delta}_i(\theta)\langle g_{\mathcal{D}_T}(\theta), g_{\mathcal{D}_{S_i}}(\theta)\rangle \cdot \langle g_{\widehat{\mathcal{D}}_T}(\theta) - g_{\mathcal{D}_T}(\theta), u_{\mathcal{D}_{S_i}}(\theta)\rangle] \\
& + \beta_i^2\mathbb{E}_{\theta, \widehat{\mathcal{D}}_T}[(1 - \widehat{\delta}_i(\theta))\|g_{\mathcal{D}_T}(\theta)\|^2] \\
= {}& \Delta_{\text{FedGP}, S_i}^2.
\end{aligned}
$$

Combining the above equation with (20) concludes the proof.

$\square$

**Approximations.** As we can see, the Delta error of `FedGP` is rather complicated at its precise form. However, reasonable approximation can be done to extract the useful components from it which would help us to derive its auto-weighted version. In the following, we show how we derive the approximated Delta error for `FedGP`, leading to what we present in (2).

First, we consider an approximation which is analogous to a mean-field approximation, i.e., ignoring the cross-terms in the expectation of a product. Fixing $\widehat{\delta}_i(\theta) = \bar{\delta}$ and $\tau_i(\theta) = \bar{\tau}$, i.e., their expectations. This approximation is equivalent to assuming $\widehat{\delta}(\theta), \tau(\theta)$ can be viewed as independent random variables. This results in the following.

$$
\begin{aligned}
\Delta_{\text{FedGP}, \mathcal{D}_{S_i}}^2 \approx {}& (1 - \beta_i)^2\sigma_\pi^2(\widehat{\mathcal{D}}_T) + \beta_i^2\bar{\delta}\bar{\tau}^2 d_\pi(\mathcal{D}_{S_i}, \mathcal{D}_T)^2 \\
& + (2\beta_i - \beta_i^2)\bar{\delta}\mathbb{E}_{\theta, \widehat{\mathcal{D}}_T}\langle g_{\widehat{\mathcal{D}}_T}(\theta) - g_{\mathcal{D}_T}(\theta), u_{\mathcal{D}_{S_i}}(\theta)\rangle^2 + \beta_i^2(1 - \bar{\delta})\|g_{\mathcal{D}_T}(\theta)\|_\pi^2.
\end{aligned}
$$

The term $\mathbb{E}_{\widehat{\mathcal{D}}_T}\langle g_{\widehat{\mathcal{D}}_T}(\theta) - g_{\mathcal{D}_T}(\theta), u_{\mathcal{D}_{S_i}}(\theta)\rangle^2$ is the variance of $g_{\widehat{\mathcal{D}}_T}(\theta)$ when projected to a direction $u_{\mathcal{D}_{S_i}}(\theta)$.

We consider a further approximation based on the following intuition: consider a zero-mean random vector $\hat{v} \in \mathbb{R}^m$ with i.i.d. entries $\hat{v}_j$ for $j \in [m]$. After projecting the random vector to a fixed unit vector $u \in \mathbb{R}^m$, the projected variance is $\mathbb{E}\langle\hat{v}, u\rangle^2 = \mathbb{E}(\sum_{j=1}^m \hat{v}_j u_j)^2 = \sum_{j=1}^m \mathbb{E}[\hat{v}_j^2]u_j^2 = {}$

$\sum_{j=1}^{m} \frac{\sigma^2(\hat{v})}{m} u_j^2 = \sigma^2(\hat{v})/m$, i.e., the variance becomes much smaller. Therefore, knowing that the parameter space $\Theta$ is $m$-dimensional, combined with the approximation that $g_{\widehat{\mathcal{D}}_T}(\theta) - g_{\mathcal{D}_T}(\theta)$ is element-wise i.i.d., we derive a simpler (approximate) result.

$$\Delta^2_{\text{FedGP}, \mathcal{D}_{S_i}} \approx \left( (1 - \beta_i)^2 + \frac{(2\beta_i - \beta_i^2)\bar{\delta}}{m} \right) \sigma_\pi^2(\widehat{\mathcal{D}}_T) + \beta_i^2 \bar{\delta} \bar{\tau}^2 d_\pi(\mathcal{D}_{S_i}, \mathcal{D}_T)^2 + \beta_i^2 (1 - \bar{\delta}) \|g_{\mathcal{D}_T}\|_\pi^2.$$

In fact, this approximated form can already be used for deriving the auto-weighted version for FedGP, as it is quadratic in $\beta_i$ and all of the terms can be estimated. However, we find that in practice $\bar{\delta} \approx 1$, and thus simply setting $\bar{\delta} = 1$ makes little impact on $\Delta^2_{\text{FedGP}, \mathcal{D}_{S_i}}$. Therefore, for simplicity, we choose $\bar{\delta} = 1$ as an approximation, which results in

$$\Delta^2_{\text{FedGP}, \mathcal{D}_{S_i}} \approx \left( (1 - \beta_i)^2 + \frac{(2\beta_i - \beta_i^2)}{m} \right) \sigma_\pi^2(\widehat{\mathcal{D}}_T) + \beta_i^2 \bar{\tau}^2 d_\pi(\mathcal{D}_{S_i}, \mathcal{D}_T)^2.$$

This gives the result shown in (2).

Although many approximations are made, we observe in our experiments that the auto-weighted scheme derived upon this is good enough to improve FedGP.

## A.5 Additional Discussion of the Auto-weighting Method and FedGP

In this sub-section, we show how we estimate the optimal $\beta_i$ for both FedDA and FedGP. Moreover, we discuss the intuition behind why FedGP with a fixed $\beta = 0.5$ is fairly good in many cases.

In order to compute the $\beta$ for each methods, as shown in Section 4.2, we need to estimate the following three quantities: $\sigma_\pi^2(\widehat{\mathcal{D}}_T)$, $d_\pi^2(\mathcal{D}_{S_i}, \mathcal{D}_T)$, and $\bar{\tau}^2 d_\pi^2(\mathcal{D}_{S_i}, \mathcal{D}_T)$. In the following, we derive unbiased estimators for each of the quantities, followed by a discussion of the choice of $\pi$.

The essential technique is to divide the target domain dataset $\widehat{\mathcal{D}}_T$ into many pieces, serving as samples. Concretely, say we randomly divide $\widehat{\mathcal{D}}_T$ into $B$ parts of equal size, denoting $\widehat{\mathcal{D}}_T = \cup_{j=1}^B \widehat{\mathcal{D}}_T^j$ (without loss of generality we may assume $|\widehat{\mathcal{D}}_T|$ can be divided by $B$). This means

$$g_{\widehat{\mathcal{D}}_T} = \frac{1}{B} \sum_{j=1}^{B} g_{\widehat{\mathcal{D}}_T^j}. \tag{21}$$

Note that each $g_{\widehat{\mathcal{D}}_T^j}$ is a sample of dataset formed by $|\widehat{\mathcal{D}}_T|/B$ data points. We denote $\widehat{\mathcal{D}}_{T,B}$ as the corresponding random variable (i.e., a dataset of $|\widehat{\mathcal{D}}_T|/B$ sample points sampled i.i.d. from $\mathcal{D}_T$). Since we assume each data points in $\widehat{\mathcal{D}}_T$ is sampled i.i.d. from $\mathcal{D}_T$, we have

$$\mathbb{E}[g_{\widehat{\mathcal{D}}_{T,B}}] = \mathbb{E}[g_{\widehat{\mathcal{D}}_T^j}] = \mathbb{E}[g_{\widehat{\mathcal{D}}_T}] = g_{\mathcal{D}_T}. \tag{22}$$

Therefore, we may view $\widehat{\mathcal{D}}_T^j$ as i.i.d. samples of $\widehat{\mathcal{D}}_{T,B}$, which we can used for our estimation.

**Estimator of $\sigma_\pi^2(\widehat{\mathcal{D}}_T)$.**

First, for $\sigma_\pi^2(\widehat{\mathcal{D}}_T)$, we can derive that

$$\sigma_\pi^2(\widehat{\mathcal{D}}_T) := \mathbb{E}\|g_{\mathcal{D}_T} - g_{\widehat{\mathcal{D}}_T}\|_\pi^2 = \mathbb{E}\|g_{\mathcal{D}_T} - \frac{1}{B} \sum_{j=1}^{B} g_{\widehat{\mathcal{D}}_T^j}\|_\pi^2$$

$$= \frac{1}{B^2} \mathbb{E}\| \sum_{j=1}^{B} (g_{\mathcal{D}_T} - g_{\widehat{\mathcal{D}}_T^j})\|_\pi^2$$

$$= \frac{1}{B^2} \sum_{j=1}^{B} \mathbb{E}\|g_{\mathcal{D}_T} - g_{\widehat{\mathcal{D}}_T^j}\|_\pi^2 \qquad \text{(by (22))}$$

$$= \frac{1}{B} \sigma_\pi^2(\widehat{\mathcal{D}}_{T,B}) \qquad \text{(by (22))}$$

Since we have $B$ i.i.d. samples of $\widehat{\mathcal{D}}_{T,B}$, we can use their sample variance, denoted as $\widehat{\sigma}_\pi^2(\widehat{\mathcal{D}}_{T,B})$, as an unbiased estimator for its variance $\sigma_\pi^2(\widehat{\mathcal{D}}_{T,B})$. Concretely, the sample variance is

$$\widehat{\sigma}_\pi^2(\widehat{\mathcal{D}}_{T,B}) = \frac{1}{B-1} \sum_{j=1}^{B} \|g_{\widehat{\mathcal{D}}_T^j} - g_{\widehat{\mathcal{D}}_T}\|_\pi^2.$$

Note that, as shown in (21), $g_{\widehat{\mathcal{D}}_T}$ is the sample mean. It is a known statistical fact that sample variance is an unbiased estimator of the variance, i.e.,

$$\mathbb{E}[\widehat{\sigma}_\pi^2(\widehat{\mathcal{D}}_{T,B})] = \sigma_\pi^2(\widehat{\mathcal{D}}_{T,B}).$$

Combining the above, our estimator for $\sigma_\pi^2(\widehat{\mathcal{D}}_T)$ is

$$\widehat{\sigma}_\pi^2(\widehat{\mathcal{D}}_T) = \frac{1}{B}\widehat{\sigma}_\pi^2(\widehat{\mathcal{D}}_{T,B}) = \frac{1}{(B-1)B} \sum_{j=1}^{B} \|g_{\widehat{\mathcal{D}}_T^j} - g_{\widehat{\mathcal{D}}_T}\|_\pi^2.$$

**Estimator of $d_\pi^2(\mathcal{D}_{S_i}, \mathcal{D}_T)$.**

For $d_\pi^2(\mathcal{D}_{S_i}, \mathcal{D}_T)$, we adopt a similar approach. We first apply the following trick:

$$d_\pi^2(\mathcal{D}_{S_i}, \mathcal{D}_T) = \|g_{\mathcal{D}_{S_i}} - g_{\mathcal{D}_T}\|_\pi^2 = \|g_{\mathcal{D}_{S_i}} - g_{\mathcal{D}_T}\|_\pi^2 + \mathbb{E}\|g_{\mathcal{D}_T} - g_{\widehat{\mathcal{D}}_{T,B}}\|_\pi^2 - \mathbb{E}\|g_{\mathcal{D}_T} - g_{\widehat{\mathcal{D}}_{T,B}}\|_\pi^2$$

$$= \mathbb{E}\|g_{\mathcal{D}_{S_i}} - g_{\mathcal{D}_T} + g_{\mathcal{D}_T} - g_{\widehat{\mathcal{D}}_{T,B}}\|_\pi^2 - \mathbb{E}\|g_{\mathcal{D}_T} - g_{\widehat{\mathcal{D}}_{T,B}}\|_\pi^2 \tag{23}$$

$$= \mathbb{E}\|g_{\mathcal{D}_{S_i}} - g_{\widehat{\mathcal{D}}_{T,B}}\|_\pi^2 - \sigma_\pi^2(\widehat{\mathcal{D}}_{T,B}), \tag{24}$$

where (23) is due to that $\mathbb{E}[g_{\widehat{\mathcal{D}}_{T,B}}] = g_{\mathcal{D}_T}$ and therefore the inner product term

$$\mathbb{E}[\langle g_{\mathcal{D}_{S_i}} - g_{\mathcal{D}_T}, g_{\mathcal{D}_T} - g_{\widehat{\mathcal{D}}_{T,B}}\rangle_\pi] = \langle g_{\mathcal{D}_{S_i}} - g_{\mathcal{D}_T}, \mathbb{E}[g_{\mathcal{D}_T} - g_{\widehat{\mathcal{D}}_{T,B}}]\rangle_\pi = 0.$$

This means

$$\mathbb{E}\|g_{\mathcal{D}_{S_i}} - g_{\mathcal{D}_T} + g_{\mathcal{D}_T} - g_{\widehat{\mathcal{D}}_{T,B}}\|_\pi^2 = \|g_{\mathcal{D}_{S_i}} - g_{\mathcal{D}_T}\|_\pi^2 + \mathbb{E}\|g_{\mathcal{D}_T} - g_{\widehat{\mathcal{D}}_{T,B}}\|_\pi^2.$$

We can see that (24) has an unbiased estimator as the following

$$\widehat{d}_\pi^2(\mathcal{D}_{S_i}, \mathcal{D}_T) = \left(\frac{1}{B} \sum_{j=1}^{B} \|g_{\mathcal{D}_{S_i}} - g_{\widehat{\mathcal{D}}_T^j}\|_\pi^2\right) - \widehat{\sigma}_\pi^2(\widehat{\mathcal{D}}_{T,B})$$

**Estimator of $\bar{\tau}^2 d_\pi^2(\mathcal{D}_{S_i}, \mathcal{D}_T)$.**

Finally, it left to find an estimator for $\bar{\tau}^2 d_\pi^2(\mathcal{D}_{S_i}, \mathcal{D}_T)$. Note that we estimate $\bar{\tau}^2 d_\pi^2(\mathcal{D}_{S_i}, \mathcal{D}_T)$ directly, but not $\bar{\tau}^2$ and $d_\pi^2(\mathcal{D}_{S_i}, \mathcal{D}_T)$ separately, is because our aim in finding an unbiased estimator. Concretely, from Theorem 4.4 we can see its original form (with $\delta_i = 1$) is

$$\mathbb{E}_{\theta\sim\pi}[\tau_i^2(\theta)\|g_{\mathcal{D}_T}(\theta) - g_{\mathcal{D}_{S_i}}(\theta)\|^2] = \mathbb{E}_{\theta\sim\pi}\|g_{\mathcal{D}_T}(\theta) - \langle g_{\mathcal{D}_T}(\theta), u_{\mathcal{D}_{S_i}}(\theta)\rangle u_{\mathcal{D}_{S_i}}(\theta)\|^2, \tag{25}$$

where $u_{\mathcal{D}_{S_i}}(\theta) = g_{\mathcal{D}_{S_i}}(\theta)/\|g_{\mathcal{D}_{S_i}}(\theta)\|$.

Denote $g_{TS}, \widehat{g}_{TS}, \widehat{g}_{TS}^j : \Theta \to \Theta$ as the following:

$$g_{TS}(\theta) := g_{\mathcal{D}_T}(\theta) - \langle g_{\mathcal{D}_T}(\theta), u_{\mathcal{D}_{S_i}}(\theta)\rangle u_{\mathcal{D}_{S_i}}(\theta)$$

$$\widehat{g}_{TS}(\theta) := g_{\widehat{\mathcal{D}}_{T,B}}(\theta) - \langle g_{\widehat{\mathcal{D}}_{T,B}}(\theta), u_{\mathcal{D}_{S_i}}(\theta)\rangle u_{\mathcal{D}_{S_i}}(\theta)$$

$$\widehat{g}_{TS}^j(\theta) := g_{\widehat{\mathcal{D}}_T^j}(\theta) - \langle g_{\widehat{\mathcal{D}}_T^j}(\theta), u_{\mathcal{D}_{S_i}}(\theta)\rangle u_{\mathcal{D}_{S_i}}(\theta).$$

Therefore, we have the following two equations:

$$(25) = \|g_{TS}\|_\pi^2$$

$$\mathbb{E}[\widehat{g}_{TS}] = \mathbb{E}[\widehat{g}_{TS}^j] = g_{TS}.$$

This means we can use our samples $\widehat{\mathcal{D}}_T^j$ to compute samples of $\widehat{g}_{TS}^j$, and then estimate (25) accordingly.

Applying the same trick as before, we have

$$
\begin{aligned}
(25) = \|g_{TS}\|_\pi^2 &= \|g_{TS}\|_\pi^2 + \mathbb{E}\|\widehat{g}_{TS} - g_{TS}\|_\pi^2 - \mathbb{E}\|\widehat{g}_{TS} - g_{TS}\|_\pi^2 \\
&= \mathbb{E}\|g_{TS} + \widehat{g}_{TS} - g_{TS}\|_\pi^2 - \mathbb{E}\|\widehat{g}_{TS} - g_{TS}\|_\pi^2 \\
&= \mathbb{E}\|\widehat{g}_{TS}\|_\pi^2 - \mathbb{E}\|\widehat{g}_{TS} - g_{TS}\|_\pi^2.
\end{aligned}
$$

Note that $\mathbb{E}\|\widehat{g}_{TS}\|_\pi^2$ can be estimated unbiasedly by $\frac{1}{B}\sum_{j=1}^B \|\widehat{g}_{TS}^j\|_\pi^2$. Moreover, $\mathbb{E}\|\widehat{g}_{TS} - g_{TS}\|_\pi^2$ is the variance of $\widehat{g}_{TS}$ and thus can be estimated unbiasedly by the sample variance.

Putting all together, the estimator of $\bar{\tau}^2 d_\pi^2(\mathcal{D}_{S_i}, \mathcal{D}_T)$ (rigorously speaking, the unbiased estimator of (25)) is

$$
\left( \frac{1}{B} \sum_{j=1}^B \|\widehat{g}_{TS}^j\|_\pi^2 \right) - \left( \frac{1}{B-1} \sum_{j=1}^B \|\widehat{g}_{TS}^j - \frac{1}{B}\sum_{k=1}^B \widehat{g}_{TS}^k\|_\pi^2 \right)
$$

Therefore, we have the unbiased estimators of $\sigma_\pi^2(\widehat{\mathcal{D}}_T)$, $d_\pi^2(\mathcal{D}_{S_i}, \mathcal{D}_T)$, and $\bar{\tau}^2 d_\pi^2(\mathcal{D}_{S_i}, \mathcal{D}_T)$. Then, we can computed estimated optimal $\beta_i^{\texttt{FedDA}}$ and $\beta_i^{\texttt{FedGP}}$ according to section 4.2.

**The choice of $\pi$.**

The distribution $\pi$ characterizes where in the parameter space $\Theta$ we want to measure $\sigma_\pi^2(\widehat{\mathcal{D}}_T)$, $d_\pi^2(\mathcal{D}_{S_i}, \mathcal{D}_T)$, and the Delta errors.

In practice, we have different ways to choose $\pi$. For example, in our synthetic experiment, we simply choose $\pi$ to be the point mass of the initialization model parameter. It turns out the Delta errors computed at initialization are pretty accurate in predicting the final test results, as shown in Figure 5. For the more realistic cases, we choose $\pi$ to be the empirical distribution of parameters along the optimization path. This means that we can simply take the local updates, computed by batches of data, as $\widehat{\mathcal{D}}_T^j$ and estimate $\beta$ accordingly. Detailed implementation is shown in Section C.4.

**Intuition of why `FedGP` is more robust to the choice of $\beta$.**

To have an intuition about why FedGP is more robust to the choice of $\beta$ compared FedDA, we examine how varying $\beta$ affects their Delta errors.

Recall that

$$
\begin{aligned}
\Delta_{\texttt{FedDA},S_i}^2 &= (1-\beta_i)^2 \sigma_\pi^2(\widehat{\mathcal{D}}_T) + \beta_i^2 d_\pi^2(\mathcal{D}_{S_i}, \mathcal{D}_T) \\
\Delta_{\texttt{FedGP},S_i}^2 &\approx (1-\beta_i)^2 \sigma_\pi^2(\widehat{\mathcal{D}}_T) + \beta_i^2 \bar{\tau}^2 d_\pi^2(\mathcal{D}_{S_i}, \mathcal{D}_T).
\end{aligned}
$$

Now suppose we change $\beta_i \to \beta_i'$, and since $0 \le \bar{\tau}^2 \le 1$, we can see that the change in $\Delta_{\texttt{FedGP},S_i}^2$ is less than the change in $\Delta_{\texttt{FedDA},S_i}^2$. In other words, `FedGP` should be more robust to varying $\beta$ than `FedDA`.

## B   SYNTHETIC DATA EXPERIMENTS IN DETAILS

The synthetic data experiment aims to bridge the gap between theory and practice by verifying our theoretical insights. Specifically, we generate various source and target datasets and compute the corresponding source-target domain distance $d_\pi(\mathcal{D}_S, \mathcal{D}_T)$ and target domain variance $\sigma_\pi(\widehat{\mathcal{D}}_T)$. We aim to verify if our theory is predictive of practice.

In this experiment, we use one-hidden-layer neural networks with sigmoid activation. We generate 9 datasets $D_1, \ldots, D_9$ each consisting of 5000 data points as the following. We first generate 5000 samples $x \in \mathbb{R}^{50}$ from a mixture of 10 Gaussians. The ground truth target is set to be the sum of 100 radial basis functions, the target has 10 samples. We control the randomness and deviation of the basis function to generate datasets with domain shift. As a result, $D_2$ to $D_9$ have an increasing domain shift compared to $D_1$. We take $D_1$ as the target domain. We subsample (uniformly) 9 datasets

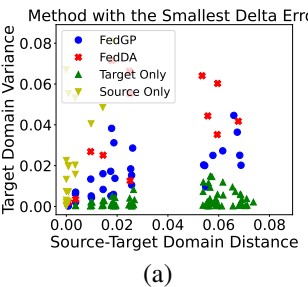 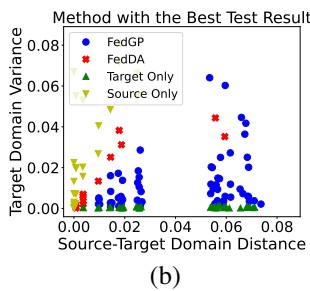 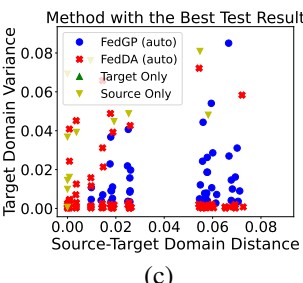

(a)          (b)          (c)

Figure 5: Given specific source-target domain distance and target domain variance: (a) shows which aggregation method has the smallest Delta error; (b)&(c) present which aggregation method actually achieves the best test result. In (a)&(b), `FedDA` and `FedGP` use a fixed $\beta = 0.5$. In (c), `FedDA` and `FedGP` adopt the auto-weighted scheme. **Observations:** Comparing (a) and (b), we can see that the prediction from the Delta errors, computed *at initialization*, mostly track the actual test performance *after training*. Comparing (b) and (c), we can see that `FedDA` is greatly improved with the auto-weighted scheme. Moreover, we can see that `FedGP` with a fixed $\beta = 0.5$ is good enough for most of the cases. These observations demonstrate the practical utility of our theoretical framework.

$\widehat{D}_1^1, \ldots, \widehat{D}_1^9$ from $D_1$ with decreasing number of subsamples. As a result, $\widehat{D}_1^1$ has the smallest target domain variance, and $\widehat{D}_1^9$ has the largest.

**Dataset.** Denote a radial basis function as $\phi_i(\boldsymbol{x}) = e^{-\|\boldsymbol{x} - \boldsymbol{\mu}_i\|_2^2/(2\sigma_i)^2}$, and we set the target ground truth to be the sum of $M = 100$ basis functions as $y = \sum_{i=1}^{M} \phi_i$, where each entry of the parameters are sampled once from $U(-0.5, 0.5)$. We set the dimension of $\boldsymbol{x}$ to be 50, and the dimension of the target to be 10. We generate $N = 5000$ samples of $\boldsymbol{x}$ from a Gaussian mixture formed by 10 Gaussians with different centers but the same covariance matrix $\boldsymbol{\Sigma} = \boldsymbol{I}$. The centers are sampled randomly from $U(-0.5, 0.5)^n$. We use the ground truth target function $y(\cdot)$ to derive the corresponding data $\boldsymbol{y}$ for each $\boldsymbol{x}$. That is, we want our neural networks to approximate $y(\cdot)$ on the Gaussian mixture.

**Methods.** For each pair of $(D_i, \widehat{D}_1^j)$ where $i, j = 1, \ldots, 9$ from the 81 pairs of datasets. We compute the source-target domain distance $d_\pi(D_i, \widehat{D}_1^j)$ and target domain variance $\sigma_\pi(\widehat{D}_1^j)$ with $\pi$ being the point mass on only the initialization parameter. We then train the 2-layer neural network with different aggregation strategies on $(D_i, \widehat{D}_1^j)$. Given the pair of datasets, we identify which strategies have the smallest Delta error and the best test performance on the target domain. We report the average results of three random trials.

As interpreted in Figure 5, the results verify that (i) the Delta error indicates the actual test result, and (ii) the auto-weighted strategy, which minimizes the estimated Delta error, is effective in practice.

## C   Supplementary Experiment Information

In this section, we provide the algorithms, computational and communication cost analysis, additional experiment details, and additional results on semi-synthetic and real-world datasets.

### C.1   Detailed Algorithm Outlines for Federated Domain Adaptation

As illustrated in Algorithm 1, for one communication round, each source client $\mathcal{C}_{S_i}$ performs supervised training on its data distribution $\mathcal{D}_{S_i}$ and uploads the weights to the server. Then the server computes and shuffles the source gradients, sending them to the target client $\mathcal{C}_T$. On $\mathcal{C}_T$, it updates its parameter using the available target data. After that, the target client updates the global model using aggregation rules (e.g. `FedDA`, `FedGP`, and their auto-weighted versions) and sends the model to the server. The server then broadcasts the new weight to all source clients, which completes one round.

## C.2 REAL-WORLD EXPERIMENT IMPLEMENTATION DETAILS AND RESULTS

**Implementation details**   We conduct experiments on three datasets: Colored-MNIST (Arjovsky et al., 2019) (a dataset derived from MNIST (Deng, 2012) but with spurious features of colors), VLCS (Fang et al., 2013) (four datasets with five categories of bird, car, chair, dog, and person), and TerraIncognita (Beery et al., 2018) (consists of 57,868 images across 20 locations, each labeled with one of 15 classes) datasets. The source learning rate is $1e^{-3}$ for Colored-MNIST and $5e^{-5}$ for VLCS and TerraIncognita datasets. The target learning rate is set to $\frac{1}{5}$ of the source learning rate. For source domains, we split the training/testing data with a 20% and 80% split. For the target domain, we use a fraction (0.1% for Colored-MNIST, 5% for VLCS, 5% for TerraIncognita) of the 80% split of training data to compute the target gradient. We report the average of the last 5 epochs of the target accuracy on the test split of the data across 5 trials. For Colored-MNIST, we use a CNN model with four convolutional and batch-norm layers. For the other two datasets, we use pre-trained ResNet-18 (He et al., 2016) models for training. Apart from that, we use the cross-entropy loss as the criterion and apply the Adam (Kingma & Ba, 2014) optimizer. For initialization, we train 2 epochs for the Colored-MNIST, VLCS datasets, as well as 5 epochs for the TerraIncognita dataset. We run the experiment with 5 random seeds and report the average accuracies over five trials with variances. Also, we set the total round $R = 50$ and the local update epoch to 1.

**Personalized FL benchmark**   We adapt the code from Marfoq et al. (2022) to test the performances of different personalization baselines. We report the personalization performance on the target domain with limited data. We train for 150 epochs for each personalization method with the same learning rate as our proposed methods. For `APFL`, the mixing parameter is set to 0.5. `Ditto`'s penalization parameter is set to 1. For `knnper`, the number of neighbors is set to 10. We reported the highest accuracy across grids of weights and capacities by evaluating pre-trained `FedAvg` for `knnper`. Besides, we found out the full training of DomainNet is too time-consuming for personalized baselines.

**Full results of all real-world datasets with error bars**   As shown in Table 2 (Colored-MNIST), Table 3 (VLCS), and Table 4 (TerraIncognita), auto-weighted methods mostly achieve the best performance compared with personalized benchmarks and other FDA baselines, across various target domains and on average. Also, `FedGP` with a fixed weight $\beta = 0.5$ has a comparable performance compared with the auto-weighted versions. `FedDA` with auto weights greatly improves the performance of the fixed-weight version.

| | **Colored-MNIST** | | | |
|---|---|---|---|---|
| Domains | +90% | +80% | -90% | Avg |
| Source Only | 56.82 (0.80) | 62.37 (1.75) | 27.77 (0.82) | 48.99 |
| Finetune_Offline | 66.58 (4.93) | 69.09 (2.62) | 53.86 (6.89) | 63.18 |
| `FedDA` | 60.49 (2.54) | 65.07 (1.26) | 33.04 (3.15) | 52.87 |
| `FedGP` | 83.68 (9.94) | 74.41 (4.40) | **89.76 (0.48)** | 82.42 |
| `FedDA_Auto` | 85.29 (5.71) | 73.13 (7.32) | 88.83(1.06) | 82.72 |
| `FedGP_Auto` | **86.18 (4.54)** | **76.49 (7.29)** | 89.62(0.43) | **84.10** |
| Target Only | 85.60 (4.78) | 73.54 (2.98) | 87.05 (3.41) | 82.06 |
| Oracle | 89.94 (0.38) | 80.32 (0.44) | 89.99 (0.54) | 86.75 |

Table 2: **Target domain test accuracy** (%) on Colored-MNIST with varying target domains.

## C.3 ADDITIONAL RESULTS ON DOMAINBED DATASETS

In this sub-section, we show (1) the performances of our methods compared with the SOTA unsupervised FDA (UFDA) methods on DomainNet and (2) additional results on PACS (Li et al., 2017) and Office-Home (Venkateswara et al., 2017) datasets.

**Implementation**   We randomly sampled 15% target domain samples of PACS, Office-Home, and DomainNet for our methods, while FADA and KD3A use 100% unlabeled data on the target domain. To have a fair comparison, we use the same model architecture ResNet-50 for all methods across three datasets. Additionally, for DomainNet, we train `FedDA`, `FedGP` and auto-weighted methods

| | VLCS | | | | |
|---|---|---|---|---|---|
| Domains | C | L | V | S | Avg |
| Source Only | 90.49 (5.34) | 60.65 (1.83) | 70.24 (1.97) | 69.10 (1.99) | 72.62 |
| Finetune_Offline | 96.65 (3.68) | 68.22 (2.26) | 74.34 (0.83) | 74.66 (2.58) | 78.47 |
| FedDA | 97.72 (0.46) | 68.17 (1.42) | **75.27 (1.45)** | 76.68 (0.91) | 79.46 |
| FedGP | 99.43 (0.30) | 71.09 (1.24) | 73.65 (3.10) | 78.70 (1.31) | 80.72 |
| FedDA_Auto | 99.78 (0.19) | 73.08 (1.31) | **78.41 (1.51)** | **83.73 (2.27)** | **83.75** |
| FedGP_Auto | **99.93 (0.16)** | 73.22 (1.81) | **78.62 (1.54)** | 83.47 (2.46) | **83.81** |
| Target Only | 97.77 (1.45) | 68.88 (1.86) | 72.29 (1.73) | 76.00 (1.89) | 78.74 |
| Oracle | 100.00 (0.00) | 72.72 (2.53) | 78.65 (1.38) | 82.71 (1.07) | 83.52 |

Table 3: **Target domain test accuracy** (%) on VLCS with varying target domains.

| | TerraIncognita | | | | |
|---|---|---|---|---|---|
| | L100 | L38 | L43 | L46 | Avg |
| Source Only | 54.62 (4.45) | 31.39 (3.13) | 36.85 (2.80) | 27.15 (1.21) | 37.50 |
| Finetune_Offline | 77.45 (3.88) | 75.22 (5.46) | 61.16 (4.07) | 58.89 (7.95) | 68.18 |
| FedDA | 77.24 (2.22) | 69.21 (1.83) | 58.55 (3.37) | 53.78 (1.74) | 64.70 |
| FedGP | 81.46 (1.28) | 77.75 (1.55) | 64.18 (2.86) | 61.56 (1.94) | 71.24 |
| FedDA_Auto | **82.35 (1.91)** | **80.77 (1.43)** | **68.42 (2.66)** | **66.87 (2.03)** | **74.60** |
| FedGP_Auto | 81.85 (2.30) | 80.50 (1.60) | **68.43 (2.08)** | **66.64 (1.52)** | 74.36 |
| Target Only | 78.85 (1.86) | 74.25 (2.52) | 58.96 (3.50) | 56.90 (3.09) | 67.24 |
| FedAvg | 38.46 | 21.24 | 39.76 | 20.54 | 30.00 |
| Ditto (Li et al., 2021) | 44.30 | 12.57 | 40.16 | 17.98 | 28.75 |
| FedRep (Collins et al., 2021) | 45.27 | 6.15 | 21.13 | 10.89 | 20.86 |
| APFL (Deng et al., 2020) | 65.16 | 64.42 | 38.97 | 42.33 | 52.72 |
| KNN-per (Marfoq et al., 2022) | 42.20 | 39.86 | 50.00 | 40.21 | 43.07 |
| Oracle | 96.41 (0.18) | 95.01 (0.28) | 91.98 (1.17) | 89.04 (0.93) | 93.11 |

Table 4: **Target domain test accuracy** (%) on TerraIncognita with varying target domains.

with 20 global epochs and 1 local epochs; for the other two datasets, we train our methods for 50 epochs. For the auto-weighted methods, we notice that it converges faster than their fixed-weight counterpart. Therefore, we set the learning rate ratio to be 0.25 to prevent overfitting for DomainNet and PACS. For Office-Home, we set the rate ratio to be 0.1 on A domain and 0.5 on other domains. For DomainNet, the target batch size is set to 64 for auto-weighted methods and 16 for FedDA and FedGP; the source batch size is set to 128. For PACS, the target batch size is 4 for auto-weighted methods and 16 for FedDA and FedGP; the source batch size is 32. For Office-Home, the target batch size is 64 for auto-weighted methods and 16 for FedDA and FedGP; the source batch size is 32. Also, we initialize 2 epochs for DomainNet, PACS, and domain A of Office-Home; for other domains in Office-Home, we perform the initialization for 5 epochs. Also, we use a learning rate of $5e^{-5}$ for source clients. The learning rate of the target client is set to $\frac{1}{5}$ of the source learning rate.

**Comparison with UFDA methods** We note that our methods work differently than UFDA methods. I.e., we consider the case where the target client possesses only limited labeled data, while the UFDA case assumes the existence of abundant unlabeled data on the target domain. As shown in Table 5, FedGP outperforms UFDA baselines using ResNet-101 in most domains and on average with a significant margin. Especially for *quick* domain, when the source-target domain shift is large (Source Only has a much lower performance compared with Oracle), our methods significantly improve the accuracy, which suggests our methods can achieve a better bias-variance trade-off under the condition when the shift is large. However, we observe that the estimated betas from the auto-weighted scheme seem to become less accurate and more expensive to compute with a larger dataset, which potentially leads to its slightly worsened performance compared with FedGP. In future work, we will keep exploring how to speed up and better estimate the bias and variances terms for larger datasets.

**PACS results** As shown in Table 6, we compare our methods with the SOTA Domain Generalization (DG) methods on the PACS dataset. The results show that our FedDA, FedGP and their auto-weighted versions are able to outperform the DG method with significant margins.

| Domains | clip | info | paint | quick | real | sketch | Avg |
|---|---|---|---|---|---|---|---|
| Source Only | 52.1 | 23.1 | 47.7 | 13.3 | 60.7 | 46.5 | 40.6 |
| FADA (100%) | 59.1 | 21.7 | 47.9 | 8.8 | 60.8 | 50.4 | 41.5 |
| KD3A (100%) (ResNet-50) | 63.7 | 15.4 | 53.5 | 11.5 | 65.4 | 53.4 | 43.8 |
| FedDA (ResNet-50) | **67.1** | 26.7 | 56.1 | 33.8 | 67.1 | 55.7 | 51.1 |
| FedGP (ResNet-50) | 64.0 | 26.6 | **56.8** | **51.1** | **71.3** | 52.3 | **53.7** |
| FedDA_auto (ResNet-50) | 62.0 | **27.8** | 56.7 | 50.9 | 68.1 | 53.3 | 53.1 |
| FedGP_auto (ResNet-50) | 62.2 | 27.7 | 56.8 | 50.7 | 68.4 | **53.5** | 53.2 |
| Best DG | 59.2 | 19.9 | 47.4 | 14.0 | 59.8 | 50.4 | 41.8 |
| Oracle | 69.3 | 34.5 | 66.3 | 66.8 | 80.1 | 60.7 | 63.0 |

Table 5: **Target domain test accuracy** (%) on DomainNet. FedGP and auto-weighted methods generally outperform UFDA methods with significant margins by using 15% of the data.

| Domains | A | C | P | S | Avg |
|---|---|---|---|---|---|
| FedDA | 92.6 | 89.1 | 97.4 | 89.2 | 92.0 |
| FedGP | **94.4** | 92.2 | **97.6** | **88.9** | 93.3 |
| FedDA_Auto | 94.2 | 90.9 | 96.6 | 89.6 | 92.8 |
| FedGP_Auto | 94.2 | **93.7** | 97.3 | 88.3 | **93.4** |
| Best DG | 87.8 | 81.8 | 97.4 | 82.1 | 87.2 |

Table 6: **Target domain test accuracy** (%) on PACS by using 15% data samples. We see FedGP and auto-weighted methods outperformed DG methods with significant margins.

**Office-Home results** As shown in Table 7, we compare our methods with the SOTA DG methods on the Office-Home dataset. The results show that our FedDA, FedGP and their auto-weighted versions are able to outperform the DG method with significant margins. We found that the auto-weighting versions of FedDA and FedGP generally outperform their fixed-weight counterparts. We notice that FedDA where the fixed weight choice of $\beta = 0.5$ is surprisingly good on Office-Home. We observe that on Office-Home, the source-only baseline sometimes surpasses the Oracle performance on the target domain. Thus, we conjecture that the fixed weight $\beta = 0.5$ happens to be a good choice for FedDA on Office-Home, while the noisy target domain data interferes with the auto-weighting mechanism. Nevertheless, the auto-weighted FedGP still shows improvement over its fixed-weight version.

| Domains | A | C | P | R | Avg |
|---|---|---|---|---|---|
| Source Only | 50.9 | 66.1 | 74.5 | 76.2 | 66.9 |
| FedDA | **67.9** | **68.2** | **82.6** | **78.6** | **74.3** |
| FedGP | 63.8 | 65.7 | 81.0 | 74.4 | 71.2 |
| FedDA_Auto | 67.4 | 65.6 | 82.2 | 75.8 | 72.7 |
| FedGP_Auto | 66.1 | 64.5 | 82.1 | 74.9 | 71.9 |
| Oracle | 70.9 | 58.5 | 87.4 | 75.0 | 73.0 |
| Best DG | 64.5 | 54.8 | 76.6 | 78.1 | 68.5 |

Table 7: **Target domain test accuracy** (%) on Office-Home by using 15% data samples. We see when source-target shifts are small, FedDA works surprisingly well; FedGP_Auto manages to improve FedGP.

## C.4 Auto-Weighted Methods: Implementation Details, Time and Space Complexity

**Implementation** As outlined in Algorithm 1, we compute the source gradients and target gradients locally when performing local updates. For the target client, during local training, it optimizes its model with $B$ batches of data, and hence it computes $B$ gradients (effectively local updates) during this round. After this round of local optimization, the target client receives the source gradients (source local updates) from the server. Note that the source clients do *not* need to remember nor send its updates for every batch (as illustrated in Algorithm 1), but simply one model update per source

client and we can compute the average model update divided by the number of batches. Also, because of the different learning rates for the source and target domains, we align the magnitudes of the gradients using the learning rate ratio. Then, on the target client, it computes $\{d_\pi(\mathcal{D}_{S_i}, \mathcal{D}_T)\}_{i=1}^N$ and $\sigma_\pi(\widehat{\mathcal{D}}_T)$ using $\{g_{S_i}\}_{i=1}^N$ and $\{g_{\widehat{\mathcal{D}}_T}^j\}_{j=1}^B$. Our theory suggests we can find the best weights $\{\beta_i\}_{i=1}^N$ values for all source domains per round, as shown in Section 4.2, which we use for aggregation.

We discovered the auto-weighted versions (`FedDA_Auto` and `FedGP_Auto`) usually have a quicker convergence rate compared with static weights, we decide to use a smaller learning rate to train, in order to prevent overfitting easily. In practice, we generally decrease the learning rate by some factors after the initialization stage. For Colored-MNIST, we use factors of $(0.1, 0.5, 0.25)$ for domains $(+90\%, +80\%, -90\%)$. For domains of the other two datasets, we use the same factor of $0.25$. Apart from that, we set the target domain batch size to $2, 4, 8$ for Colored-MNIST, VLCS, and TerraIncognita, respectively.

The following paragraphs discuss the extra time, space, and communication costs needed for running the auto-weighted methods.

**Extra time cost**   To compute a more accurate estimation of target variance, we need to use a smaller batch size for the training on the target domain (more batches lead to more accurate estimation), which will increase the training time for the target client during each round. Moreover, we need to compute the auto weights each time when we perform the aggregation. The extra time cost in this computation is linear in the number of batches.

**Extra space cost**   In each round, the target client needs to store the batches of gradients on the target client. I.e., the target client needs to store extra $B$ model updates.

**Extra communication cost**   Since the aggregation is performed on the server, the extra communication would be sending the $B$ model updates from the target client to the central server.

**Discussion**   We choose to designate the target client to estimate the optimal $\beta$ value as we assume the cross-silo setting where the number of source clients is relatively small. In other scenarios where the number of source clients is more than the number of batches used on the target client, one may choose to let the global server compute the auto weights as well as do the aggregation, i.e., the target client sends its batches of local updates to the global server, which reduces the communication cost and shifts the computation task to the global server. If one intends to avoid the extra cost induced by the auto-weighting method, we note that the `FedGP` with a fixed $\beta = 0.5$ can be good enough, especially when the source-target domain shift is big. To reduce the extra communication cost of the target client sending extra model updates to the server, we leave it to be an interesting future work (Rothchild et al., 2020).

## C.5 VISUALIZATION OF AUTO-WEIGHTED BETAS VALUES ON REAL-WORLD DISTRIBUTION SHIFTS

Figure 6 and Figure 7 show the curves of auto weights $\beta$ for each source domain with varying target domains on the Colored-MNIST dataset, using `FedDA` and `FedGP` aggregation rules respectively. Similarly, Figure 8 and Figure 9 are on the VLCS dataset; Figure 10 and Figure 11 are on the TerraIncognita dataset. From the results, we observe `FedDA_Auto` usually has smaller $\beta$ values compared with `FedGP_Auto`. `FedDA_Auto` has drastically different ranges of weight choice depending on the specific target domain (from $1e^{-3}$ to $1e^{-1}$), while `FedGP_Auto` usually has weights around $1e^{-1}$. Also, `FedDA_Auto` has different weights for each source domain while interestingly, `FedGP_Auto` has the almost same weights for each source domain for various experiments. Additionally, the patterns of weight change are unclear - in most cases, they have an increasing or increasing-then-decreasing pattern.

## C.6 ADDITIONAL EXPERIMENT RESULTS ON VARYING STATIC WEIGHTS ($\beta$)

From the Table 8, 9, 10, and 11, we see `FedGP` is less sensitive to the choice of the weight parameter $\beta$, enjoying a wider choice range of values, compared with `FedDA`. Additionally, under fixed weight conditions, `FedGP` generally outperforms `FedDA` in most cases.

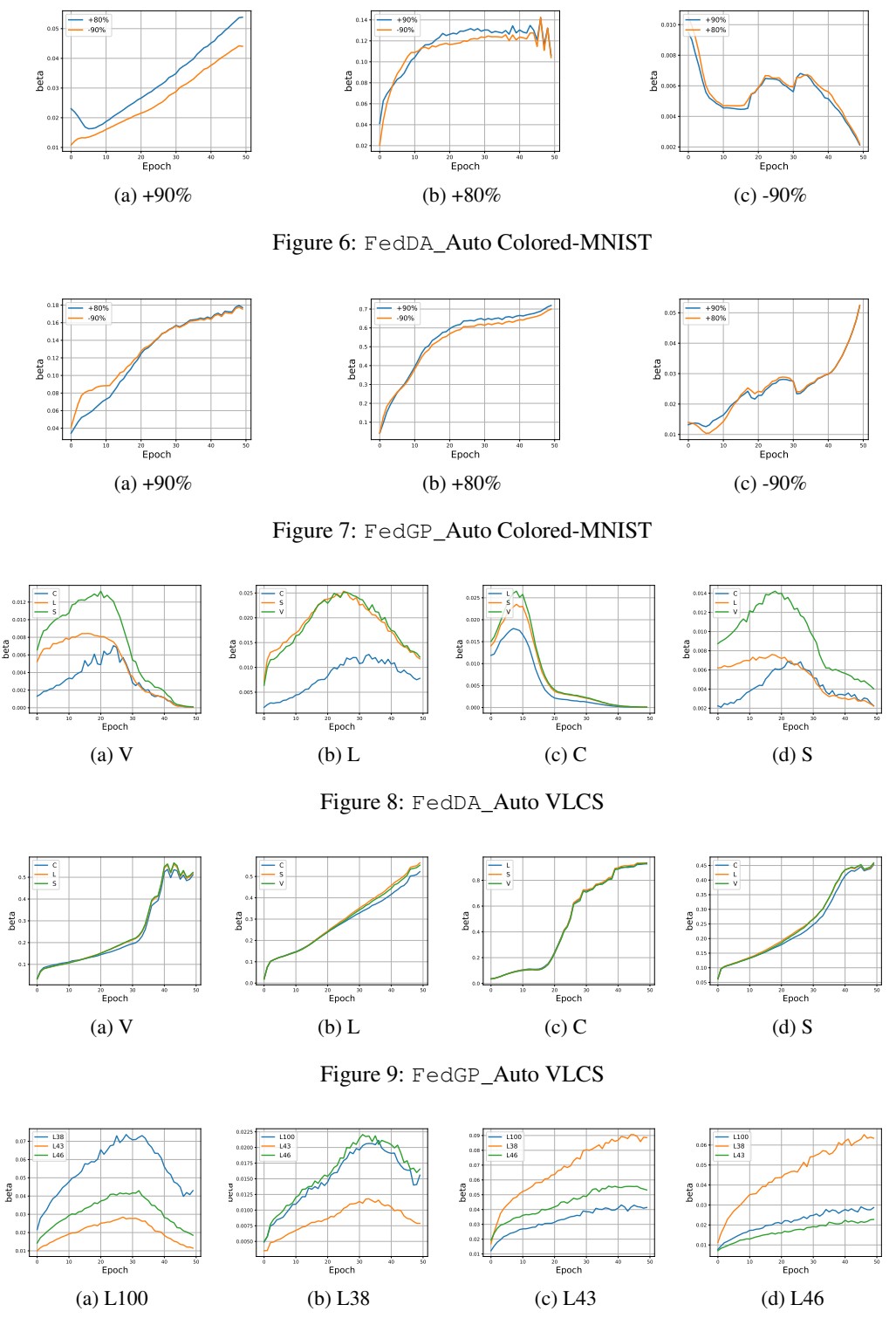

Figure 6: `FedDA_Auto` Colored-MNIST

Figure 7: `FedGP_Auto` Colored-MNIST

Figure 8: `FedDA_Auto` VLCS

Figure 9: `FedGP_Auto` VLCS

Figure 10: `FedDA_Auto` TerraIncognita

## C.7 SEMI-SYNTHETIC EXPERIMENT SETTINGS, IMPLEMENTATION, AND RESULTS

In this sub-section, we empirically explore the impact of different extents of domain shifts on `FedDA`, `FedGP`, and their auto-weighted version. To achieve this, we conduct a semi-synthetic experiment,

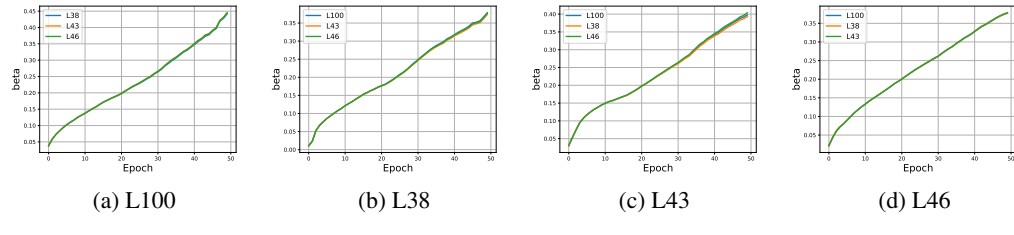

| (a) L100 | (b) L38 | (c) L43 | (d) L46 |

Figure 11: `FedGP_Auto` TerraIncognita

|        | 0     | 0.2       | 0.4       | 0.6       | 0.8       | 1.0       |
|--------|-------|-----------|-----------|-----------|-----------|-----------|
| FedDA  | 61.21 | 61.10     | 59.15     | 49.90     | 29.80     | 17.65     |
| FedGP  | 61.21 | **63.82** | **64.30** | **65.31** | **64.80** | **38.48** |

Table 8: The effect of $\beta$ on `FedDA` and `FedGP` on CIFAR-10 dataset with 0.4 noise level.

| -90%   | 0     | 0.2       | 0.4       | 0.6       | 0.8       | 1.0       |
|--------|-------|-----------|-----------|-----------|-----------|-----------|
| FedDA  | 84.41 | 73.55     | 54.19     | 35.16     | 31.16     | **27.59** |
| FedGP  | 84.41 | **88.96** | **89.50** | **89.95** | **90.03** | 9.85      |

Table 9: The effect of $\beta$ on `FedDA` and `FedGP` on Colored-MNIST -90% domain.

| +90%   | 0     | 0.2       | 0.4       | 0.6       | 0.8       | 1.0       |
|--------|-------|-----------|-----------|-----------|-----------|-----------|
| FedDA  | 88.96 | 82.37     | 71.32     | 62.03     | 58.77     | 55.23     |
| FedGP  | 88.96 | **89.98** | **89.76** | **89.84** | **90.18** | **69.47** |

Table 10: The effect of $\beta$ on `FedDA` and `FedGP` on Colored-MNIST +90% domain.

| +80%   | 0     | 0.2       | 0.4       | 0.6       | 0.8       | 1.0       |
|--------|-------|-----------|-----------|-----------|-----------|-----------|
| FedDA  | 73.66 | **75.12** | 70.21     | 65.14     | 61.84     | 61.32     |
| FedGP  | 73.66 | 73.61     | **74.39** | **79.32** | **80.11** | **76.67** |

Table 11: The effect of $\beta$ on `FedDA` and `FedGP` on Colored-MNIST +80% domain.

where we manipulate the extent of domain shifts by adding different levels of Gaussian noise (noisy features) and degrees of class imbalance (label shifts). We show that the main impact comes from the shifts between target and source domains instead of the shifts between source domains themselves.

**Datasets and models** We create the semi-synthetic distribution shifts by adding different levels of feature noise and label shifts to Fashion-MNIST (Xiao et al., 2017) and CIFAR-10 (Krizhevsky et al., 2009) datasets, adapting from the Non-IID benchmark (Li et al., 2022). For the model, we use a CNN model architecture consisting of two convolutional layers and three fully-connected layers. We set the communication round $R = 50$ and the local update epoch to 1, with 10 clients (1 target, 9 source clients) in the system.

**Baselines** We compare the following methods: **Source Only**: we only use the source gradients by averaging. **Finetune_Offline**: we perform the same number of 50 epochs of fine-tuning after FedAvg. **FedDA** ($\beta = 0.5$): a convex combination with a middle trade-off point of source and target gradients. **FedGP** ($\beta = 0.5$): a middle trade-off point between source and target gradients with gradient projection. **Target Only**: we only use the target gradient ($\beta = 0$). **Oracle**: a fully supervised training on the labeled target domain serving as the upper bound.

**Implementation** For the experiments on the Fashion-MNIST dataset, we set the source learning rate to be 0.01 and the target learning rate to 0.05. For CIFAR-10, we use a 0.005 source learning rate and a 0.0025 learning rate. The source batch size is set to 64 and the target batch size is 16. We

partition the data to clients using the same procedure described in the benchmark (Li et al., 2022). We use the cross-entropy loss as the criterion and apply the Adam (Kingma & Ba, 2014) optimizer.

**Setting 1: Noisy features**  We add different levels of Gaussian noise to the target domain to control the source-target domain differences. For the Fashion-MNIST dataset, we add Gaussian noise levels of $std = (0.2, 0.4, 0.6, 0.8)$ to input images of the target client, to create various degrees of shifts between source and target domains. The task is to predict 10 classes on both source and target clients. For the CIFAR-10 dataset, we use the same noise levels, and the task is to predict 4 classes on both source and target clients. We use 100 labeled target samples for Fashion-MNIST and 10% of the labeled target data for the CIFAR-10 dataset.

**Setting 2: Label shifts**  We split the Fashion-MNIST into two sets with 3 and 7 classes, respectively, denoted as $D_1$ and $D_2$. A variable $\eta \in [0, 0.5]$ is used to control the difference between source and target clients by defining $D_S = \eta$ portion from $D_1$ and $(1 - \eta)$ portion from $D_2$, $D_T = (1 - \eta)$ portion from $D_1$ and $\eta$ portion from $D_2$. When $\eta = 0.5$, there is no distribution shift, and when $\eta \to 0$, the shifts caused by label shifts become more severe. We use 15% labeled target samples for the target client. We test on cases with $\eta = [0.45, 0.30, 0.15, 0.10, 0.05, 0.00]$.

**Auto-weighted methods and `FedGP` maintain a better trade-off between bias and variance**  Table 12 and Table 13 display the performance trends of compared methods versus the change of source-target domain shifts. In general, when the source-target domain difference grows bigger, `FedDA`, Finetune_Offline, and FedAvg degrade more severely compared with auto-weighted methods, `FedGP` and Target Only. We find that auto-weighted methods and `FedGP` outperform other baselines in most cases, showing a good ability to balance bias and variances under various conditions and being less sensitive to changing shifts. For the label shift cases, the target variance decreases as the domain shift grows bigger (easier to predict with fewer classes). Therefore, auto-weighted methods, `FedGP` as well as Target Only surprisingly achieve higher performance with significant shift cases. In addition, auto-weighted `FedDA` manages to achieve a significant improvement compared with the fixed weight `FedDA`, with a competitive performance compared with `FedGP_Auto`, while `FedGP_Auto` generally has the best accuracy compared with other methods, which coincides with the synthetic experiment results.

**Connection with our theoretical insights**  Interestingly, we see that when the shift is relatively small ($\eta = 0.45$ and 0 noise level for Fashion-MNIST), FedAvg and `FedDA` both outperform `FedGP`. Compared with what we have observed from our theory (Figure 5), adding increasing levels of noise can be regarded as going from left to right on the x-axis and when the shifts are small, we probably will get into an area where `FedDA` is better. When increasing the label shifts, we are increasing the shifts and decreasing the variances simultaneously, we go diagonally from the top-left to the lower-right in Figure 5, where we expect FedAvg is the best when we start from a small domain difference.

**Tables of noisy features and label shifts experiments**  Table 12 and Table 13 contain the full results. We see that `FedGP`, `FedDA_Auto` and `FedGP_Auto` methods obtain the best accuracy under various conditions; `FedGP_Auto` outperforms the other two in most cases, which confirms the effectiveness of our weight selection methods suggested by the theory.

**Impact of extent of shifts between source clients**  In addition to the source-target domain differences, we also experiment with different degrees of shifts within source clients. To control the extent of shifts between source clients, we use a target noise = 0.4 with $[3, 5, 7, 9]$ labels available. From the results, we discover the shifts between source clients themselves have less impact on the target domain performance. For example, the 3-label case (a bigger shift) generally outperforms the 5-label one with a smaller shift. Therefore, we argue that the source-target shift serves as the main influencing factor for the FDA problem.

## C.8   ADDITIONAL ABLATION STUDY RESULTS

**The effect of target gradient variances**  We conduct experiments with increasing numbers of target samples (decreased target variances) with varying noise levels $[0.2, 0.4, 0.6]$ on Fashion-MNIST and

| Target noise | Fashion-MNIST | | | | CIFAR-10 | | | |
| | 0 | 0.2 | 0.4 | 0.6 | 0.8 | 0.2 | 0.4 | 0.6 | 0.8 |
|---|---|---|---|---|---|---|---|---|---|
| Source Only | 83.94 | 25.49 | 18.55 | 16.71 | 14.99 | 20.48 | 17.61 | 16.44 | 16.27 |
| Finetune_Offline | 81.39 | 48.26 | 40.15 | 36.64 | 33.71 | 66.31 | 56.80 | 52.10 | 49.37 |
| FedDA_0.5 | **86.41** | 69.73 | 58.6 | 50.13 | 45.51 | 62.25 | 54.67 | 49.77 | 47.08 |
| FedGP_0.5 | 76.33 | 75.09 | 71.09 | **68.01** | 62.22 | 66.40 | **65.28** | **63.29** | **61.59** |
| FedGP_1 | 79.40 | 77.03 | 71.67 | 63.71 | 54.18 | 21.46 | 20.75 | 19.31 | 18.26 |
| FedDA_Auto | 78.25 | **77.03** | **72.68** | 67.69 | 62.85 | 65.83 | 64.09 | 62.29 | 60.39 |
| FedGP_Auto | 76.19 | 75.09 | 71.46 | 67.53 | **62.93** | **67.02** | **65.26** | 63.12 | 61.30 |
| Target Only | 74.00 | 70.59 | 66.03 | 61.26 | 57.82 | 60.69 | 60.25 | 59.38 | 59.03 |
| Oracle | 82.00 | 82.53 | 81.20 | 75.60 | 72.60 | 73.61 | 70.12 | 69.22 | 68.50 |

Table 12: **Target domain test accuracy** (%) by adding feature noise to Fashion-MNIST and CIFAR-10 datasets using different aggregation rules.

| $\eta$ | 0.45 | 0.3 | 0.15 | 0.1 | 0.05 | 0 |
|---|---|---|---|---|---|---|
| Source Only | 83.97 | 79.71 | 69.15 | 59.90 | 52.51 | 0.00 |
| Finetune_Offline | 79.84 | 80.21 | 83.13 | 85.43 | 89.63 | 33.25 |
| FedDA_0.5 | 82.44 | 80.85 | 77.50 | 76.51 | 68.26 | 59.56 |
| FedGP_0.5 | 82.97 | 83.24 | 85.97 | 88.72 | 91.89 | **98.71** |
| FedGP_1 | 77.41 | 73.12 | 62.54 | 53.56 | 27.62 | 0.00 |
| FedDA_Auto | 83.94 | 83.91 | 86.50 | 89.45 | 91.87 | 98.51 |
| FedGP_Auto | **84.68** | **84.14** | **86.72** | **89.58** | **92.03** | 98.53 |
| Target only | 81.05 | 82.44 | 84.00 | 88.02 | 89.80 | 98.32 |
| Oracle | 87.68 | 88.06 | 90.56 | 91.9 | 93.46 | 98.73 |

Table 13: **Target domain test accuracy** (%) by adding feature noise to the Fashion-MNIST dataset using different aggregation rules.

CIFAR-10 datasets with 10 clients in the system. We compare two aggregation rules `FedGP` and `FedDA`, as well as their auto-weighted versions. The results are shown in Table 15 (fixed `FedDA` and `FedGP`) and Table 16 (auto-weighted `FedDA` and `FedGP`). When the number of available target samples increases, the target performance also improves. For the static weights, we discover that `FedGP` can predict quite well even with a small number of target samples, especially when the target variance is comparatively small. For auto-weighted `FedDA` and `FedGP`, we find they usually have higher accuracy compared with `FedGP`, which further confirms our auto-weighted scheme is effective in practice. Also, we observe that sometimes `FedDA_Auto` performs better than `FedGP_Auto` (e.g. on the Fashion-MNIST dataset) and sometimes vice versa (e.g. on the CIFAR-10 dataset). We hypothesize that since the estimation of variances for `FedGP` is an approximation instead of the equal sign, it is possible that `FedDA_Auto` can outperform `FedGP_Auto` in some cases because of more accurate estimations of the auto weights $\beta$. Also, we notice auto-weighted scheme seems to improve the performance more when the target variance is smaller with more available samples and the source-target shifts are relatively small. In addition, we compare our methods with FedAvg, using different levels of data scarcity. We show our methods consistently outperform FedAvg across all cases, which further confirms the effectiveness of our proposed methods.

| Number of labels | 9 | 7 | 3 | 5 |
|---|---|---|---|---|
| Source Only | 11.27 | 18.81 | 25.54 | 34.11 |
| Finetune_Offline | 37.79 | 64.75 | 68.69 | 65.41 |
| FedDA_0.5 | 55.01 | 57.9 | 54.16 | 61.62 |
| FedGP_0.5 | **68.50** | **68.64** | **64.43** | **66.59** |
| FedGP_1 | 65.27 | 59.07 | 26.37 | 39.86 |
| Target_Only | 63.06 | 65.76 | 61.8 | 63.65 |
| Oracle | 81.2 | 81.2 | 81.2 | 81.2 |

Table 14: **Target domain test accuracy** (%) on label shifts with [3,5,7,9] labels available on Fashion-MNIST dataset with target noise = 0.4 and 100 target labeled samples.

| Noise level | | 0.2 | | | 0.4 | | | 0.6 | | |
|---|---|---|---|---|---|---|---|---|---|---|
| | | FedAvg | FedDA | FedGP | FedAvg | FedDA | FedGP | FedAvg | FedDA | FedGP |
| **Fashion-MNIST** | 100 | 75.98 | 69.73 | 75.09 | 59.36 | 58.60 | **71.09** | 49.94 | 50.13 | **68.01** |
| | 200 | 76.50 | 72.07 | 74.21 | 60.20 | 58.59 | **70.93** | 48.56 | 52.67 | **70.31** |
| | 500 | 75.55 | 76.59 | **78.41** | 58.74 | 65.34 | **74.07** | 47.90 | 54.97 | **70.52** |
| | 1000 | 76.12 | 77.92 | **78.68** | 62.33 | 68.26 | **75.17** | 50.81 | 59.16 | **71.63** |
| **CIFAR-10** | 5% | 24.50 | 62.24 | **64.21** | 21.25 | 46.89 | **63.57** | 19.42 | 47.56 | **61.39** |
| | 10% | 22.86 | 62.25 | **65.92** | 22.35 | 54.67 | **65.39** | 18.60 | 49.77 | **63.67** |
| | 15% | 23.20 | 59.16 | **65.97** | 23.25 | 56.93 | **65.11** | 17.88 | 51.83 | **63.73** |

Table 15: **Target domain test accuracy** (%) by adding feature noise=0.2, 0.4, 0.6 on the Fashion-MNIST and CIFAR-10 datasets with different numbers of available target samples using *fixed* weights, in comparison with FedAvg. We see our methods generally are more robust than FedAvg with significant improvements.

| Noise level | | 0.2 | | 0.4 | | 0.6 | |
|---|---|---|---|---|---|---|---|
| | | FedDA_Auto | FedGP_Auto | FedDA_Auto | FedGP_Auto | FedDA_Auto | FedGP_Auto |
| **Fashion-MNIST** | 100 | **79.04** | 75.45 | **72.21** | 71.93 | 66.16 | **67.47** |
| | 200 | **79.74** | 76.74 | **74.30** | 72.96 | **69.27** | 69.04 |
| | 500 | **79.48** | 78.65 | **75.21** | 74.55 | **71.40** | 70.72 |
| | 1000 | **80.23** | 79.91 | **76.75** | 76.35 | **73.16** | 73.16 |
| **CIFAR-10** | 5% | 63.04 | **65.62** | 60.79 | **62.84** | 60.02 | **60.47** |
| | 10% | 65.72 | **67.41** | 64.43 | **65.17** | 62.25 | **62.94** |
| | 15% | 66.57 | **67.56** | 65.4 | **65.92** | 63.36 | **63.14** |

Table 16: **Target domain test accuracy** (%) by adding feature noise=0.2, 0.4, 0.6 on the Fashion-MNIST and CIFAR-10 datasets with different numbers of available target samples using *auto* weights.

## C.9 IMPLEMENTATION DETAILS OF FEDGP

To implement fine-grained projection for the real model architecture, we compute the cosine similarity between one source client gradient $g_i$ and the target gradient $g_T$ for *each layer* of the model with a threshold of 0. In addition, we align the magnitude of the gradients according to the number of target/source samples, batch sizes, local updates, and learning rates. In this way, we implement FedGP by projecting the target gradient towards source directions. We show the details of implementing static and auto-weighted versions of FedGP in the following two paragraphs.

**Static-weighted FedGP implementation** Specifically, we compute the model updates from source and target clients as $G_{S_i}^{(r)} \simeq h_{S_i}^{(r)} - h_{global}^{(r-1)}$ and $G_T^{(r)} \simeq h_T^{(r)} - h_T^{(r-1)}$, respectively. In our real training process, because we use different learning rates, and training samples for source and target clients, we need to **align the magnitude of model updates**. We first align the model updates from source clients to the target client and combine the projection results with the target updates. We use $lr_T$ and $lr_S$ to denote the target and source learning rates; $batchsize_T$ and $batchsize_S$ are the batch sizes for target and source domains, respectively; $n_l$ is the labeled sample size on target client and $n_i$ is the sample size for source client $\mathcal{C}_{S_i}$; $r_S$ is the rounds of local updates on source clients. The total gradient projection $P_{GP}$ from all source clients $\{G_{S_i}^{(r)}\}_{i=1}^N$ projected on the target direction $G_T$ could be computed as follows. We use $\mathcal{L}$ to denote all layers of current model updates. $n_i$ denotes the number of samples trained on source client $\mathcal{C}_{S_i}$, which is adapted from FedAvg (McMahan et al., 2017) to redeem data imbalance issue. Hence, we normalize the gradient projections according to the number of samples. Also, $\bigcup_{l\in\mathcal{L}}^{\mathcal{L}}$ concatenates the projected gradients of all layers.

$$P_{GP} = \bigcup_{l\in\mathcal{L}} \sum_{i=1}^N \left( \mathbf{GP}\left( \left(h_{S_i}^{(r)} - h_{global}^{(r-1)}\right)^l, \left(h_T^{(r)} - h_T^{(r-1)}\right)^l \right) \cdot \frac{n_i}{\sum_i^N n_i} \cdot \frac{\frac{n_l}{batchsize_T}}{\frac{n_i}{batchsize_S}} \cdot \frac{lr_T}{lr_S} \cdot \frac{1}{r_S} \cdot \left(h_{S_i}^{(r)} - h_{global}^{(r-1)}\right) \right)$$

Lastly, a hyper-parameter $\beta$ is used to incorporate target update $G_T$ into $P_{GP}$ to have a more stable performance. The final target model weight $h_T^{(r)}$ at round $r$ is thus expressed as:

$$h_T^{(r)} = h_T^{(r-1)} + (1-\beta) \cdot P_{GP} + \beta \cdot G_T$$

**Auto-weighted `FedGP` Implementation**  For auto-weighted scheme for `FedGP`, we compute a dynamic weight $\beta_i$ for each source domain $\mathcal{D}_{S_i}$ per communication round. With a set of pre-computed $\{\beta_i\}_{i=1}^N$ weight values, the weighted projected gradients for a certain epoch can be expressed as follows:

$$P_{GP} = \bigcup_{l \in \mathcal{L}} \sum_{i=1}^N \left( \mathbf{GP}\left( \left(h_{S_i}^{(r)} - h_{global}^{(r-1)}\right)^l, \left(h_T^{(r)} - h_T^{(r-1)}\right)^l \right) \cdot \frac{n_i \cdot (1-\beta_i)}{\sum_i^N n_i} \cdot \frac{\overline{\frac{n_l}{batchsize_T}}}{\overline{\frac{n_i}{batchsize_S}}} \cdot \frac{lr_T}{lr_S} \cdot \frac{1}{r_S} \cdot \left(h_{S_i}^{(r)} - h_{global}^{(r-1)}\right) \right)$$

Similarly, we need to incorporate target update $G_T$ into $P_{GP}$. The final target model weight $h_T^{(r)}$ at round $r$ is thus expressed as:

$$h_T^{(r)} = h_T^{(r-1)} + P_{GP} + \sum_{i=1}^N \frac{n_i \cdot \beta_i}{\sum_i^N n_i} \cdot G_T$$

## C.10  Gradient Projection Method's Time and Space Complexity

**Time complexity**: Assume the total parameter is $m$ and we have $l$ layers. To make it simpler, assume each layer has an average of $\frac{m}{l}$ parameters. Computing cosine similarity for all layers of one source client is $O((\frac{m}{l})^2 \cdot l) = O(m^2/l)$. We have $N$ source clients so the total time cost for GP is $O(N \cdot m^2/l)$.

**Space complexity**: The extra memory cost for GP (computing cosine similarity) is $O(1)$ per client for storing the current cosine similarity value. In a real implementation, the whole process of projection is fast, with around 0.023 seconds per call needed for $N = 10$ clients of Fashion-MNIST experiments on the NVIDIA TITAN Xp hardware with GPU available.

## C.11  Additional Experiment Results on Fed-Heart

As a showcase of a more realistic healthcare setting, we show the performances of our methods compared with personalized baselines on the Fed-Heart dataset from FLamby (Du Terrail et al., 2022). We randomly sample 20% data for the 0, 1, 3 centers and 100% data for the 2 center since there are only 30 samples on the target domain. As shown in Table 17, our methods generally outperform other baselines with large margins. KNN-per (Marfoq et al., 2022) may not fit this scenario since the neural network we used only consists of one layer.

| center | 0 (20%) | 1 (20%) | 2 (100%) | 3 (20%) | Avg |
|---|---|---|---|---|---|
| `FedDA` | 79.81 | 78.65 | 67.50 | 62.22 | 72.05 |
| `FedGP` | 78.85 | 80.45 | 68.75 | 65.33 | 73.35 |
| `FedDA_Auto` | **80.77** | **80.90** | 68.75 | **71.11** | **75.38** |
| `FedGP_Auto` | 80.77 | 79.78 | 68.75 | 69.78 | 74.77 |
| Source only | 76.92 | 75.96 | 62.50 | 55.56 | 67.74 |
| FedAvg | 75.96 | 76.40 | 62.50 | 55.56 | 67.61 |
| Ditto | 76.92 | 73.03 | 62.50 | 55.56 | 67.00 |
| FedRep | 78.84 | 65.17 | **75.00** | 60.00 | 69.75 |
| APFL | 51.92 | 57.30 | 31.25 | 42.22 | 45.67 |
| KNN-Per | 56.00 | 56.00 | 56.00 | 56.00 | 56.00 |

Table 17: **Target domain test accuracy** (%) on Fed-Heart. `FedGP` and auto-weighted methods generally outperform personalized FL methods with significant margins.

## C.12  Comparison with the Semi-Supervised Domain Adaptation (SSDA) Method

In this sub-section, we show the performances of our methods compared with SSDA methods. However, we note that the suggested SSDA methods cannot be directly adapted to the federated setting without major modification. Kim & Kim (2020) uses feature alignments and requires access to the source and target data at the same time, which is usually difficult to achieve in federated learning. As for Saito et al. (2019), the overall adversarial learning objective functions consist of a

loss objective on both source and target labeled data and the entropy coming from the unlabeled target data, which also cannot be directly adapted to federated learning. On the contrary, auto-weighted methods and `FedGP` have the flexibility to do the single source-target domain adaptation, which can be compared with the SSDA method, though we notice that our setting is different from SSDA since we do not have unlabeled data on the target domain and do not leverage the information coming from this set of data. Here, we perform experiments on real-world datasets: our results suggest that auto-weighted methods and `FedGP` outperform MME (Saito et al., 2019) when the shifts are large even without using unlabeled data (overall our proposed methods have a comparable performance with MME). Also, we observe in a single source-target domain adaptation setting, auto-weighted `FedGP` usually has a better performance than auto-weighted `FedDA` and `FedGP`.

|  | 0 ->1 / 1 ->0 | 0 ->2 / 2 ->0 | 1 ->2 / 2 ->1 |
|---|---|---|---|
| MME (Saito et al., 2019) | **79.55 / 89.28** | 25.98 / 12.99 | 14.66 / 24.43 |
| `FedGP` | 79.34 / 88.34 | **90.23** / 64.47 | **90.23** / 78.70 |
| `FedDA_Auto` | 79.31 / 88.38 | 87.00 / 56.07 | 89.13 / 66.59 |
| `FedGP_Auto` | 79.34 / 71.59 | **90.23 / 80.95** | **90.23 / 79.34** |

Table 18: Colored-MNIST (0: +90%, 1: +80%, 2: -90%)

|  | 0->1 | 1->2 | 2->3 |
|---|---|---|---|
| MME (Saito et al., 2019) | **68.68** | 72.86 | **81.90** |
| `FedGP` | 67.92 | 75.30 | 77.45 |
| `FedDA_Auto` | 67.92 | **75.73** | 76.97 |
| `FedGP_Auto` | **68.45** | 74.21 | 77.69 |

Table 19: VLCS (0: C, 1: L, 2: V, 3: S)

|  | 0->1 | 1->2 | 2->3 |
|---|---|---|---|
| MME (Saito et al., 2019) | **74.51** | 54.91 | 58.50 |
| `FedGP` | 72.35 | **58.19** | 60.03 |
| `FedDA_Auto` | 72.31 | 54.26 | 61.33 |
| `FedGP_Auto` | 72.42 | 56.27 | **61.90** |

Table 20: TerraIncognita (0: L100, 1: L38, 2: L43, 3: L46)

