`. We find that while `FedGP` is less sensitive to re-weighting with relatively good performance for simply averaging the source and target gradients, the under-performing `FedDA` is significantly improved by using auto-weighting – enough to be competitive with `FedGP`, demonstrating the value of our theory. Across extensive datasets, we demonstrate that `FedGP`, as well as the auto-weighted `FedDA` and `FedGP`, outperforms personalized FL and UFDA baselines.

**Summary of Contributions.** Our contributions are both theoretical and practical, addressing the FDA problem through federated aggregation in a principled way.

- We introduce a theoretical framework understanding and analyzing the performance of FDA aggregation rules, inspired by two challenges existing in FDA. Our theories provide a principled response to the question: *How do we define a "good" FDA aggregation rule?*

- We propose `FedGP` as an effective solution to FDA challenges of substantial domain shifts and limited target data.

- Our theory determines the optimal weight parameter for aggregation rules, `FedDA` and `FedGP`. This *auto-weighting* scheme leads to further performance improvements.

- Extensive experiments illustrate that our theory is predictive of practice. The proposed methods outperform personalized FL and UFDA baselines on real-world ColoredMNIST, VLCS, TerraIncognita, and DomainNet datasets.

## 2 THE PROBLEM OF FEDERATED DOMAIN ADAPTATION

We begin with a general definition of the problem of Federated Domain Adaptation and, subsequently a review of related literature in the field.

**Notation.** Let $\mathcal{D}$ be a data domain[1] on a ground set $\mathcal{Z}$. In our supervised setting, a data point $z \in \mathcal{Z}$ is the tuple of input and output data[2]. We denote the loss function as $\ell : \Theta \times \mathcal{Z} \to \mathbb{R}_+$ where the parameter space is $\Theta = \mathbb{R}^m$; an $m$-dimensional Euclidean space. The population loss is $\ell_{\mathcal{D}}(\theta) := \mathbb{E}_{z \sim \mathcal{D}} \ell(\theta, z)$, where $\mathbb{E}_{z \sim \mathcal{D}}$ is the expectation w.r.t. $\mathcal{D}$. Let $\widehat{\mathcal{D}}$ be a finite sample dataset

---

[1]In this paper, the terms distribution and domain are used interchangeably.

[2]Let $x$ be the inputs and $y$ be the targets, then $z = (x, y)$. We do not need the prediction function details for our analysis, so we will not use it.

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

 with Feng et al. (2021) of ResNet-101 for `FedGP`. However, we notice the computational overhead for auto-weighted methods is high in this case since we need to compute and store multiple model updates for each epoch. Hence, we use ResNet-50 for auto-weighted `FedDA` and `FedGP`. Additionally, we train `FedGP` and auto-weighted methods with 20 global epochs and 1 local epoch. For the auto-weighted methods, we notice that it converges faster than their fixed-weight counterpart. Therefore, we set the learning rate ratio to be $0.25$ to prevent overfitting. The target batch size is set to 64 for auto-weighted methods and 16 for `FedGP`. The source batch size is set to 128. Also, we use a learning rate of $5e^{-5}$ for source clients. The learning rate of the target client is set to $\frac{1}{5}$ of the source learning rate.

**Results** As shown in Table 14, `FedGP` outperforms UFDA baselines using ResNet-101 in most domains and on average with a significant margin. Especially for *quick* domain, when the source-target domain shift is large (Source Only has a much lower performance compared with Oracle), our methods significantly improve the accuracy, which suggests our methods can achieve a better bias-variance trade-off under the condition when the shift is large. Also, since we use ResNet-50 instead of ResNet-101 for auto-weighted methods `FedGP` and `FedDA`, we got a degraded performance compared with `FedGP`. However, they still outperform on average compared with KD3A, although using a much smaller model. Here, we also see `FedGP` can scale more easily to larger models and datasets compared with the auto-weighted methods. Further, we observe that personalized FL baselines are usually too slow to run in this case, because empirically speaking, they require more epochs to converge.

## C.11 COMPARISON WITH THE SEMI-SUPERVISED DOMAIN ADAPTATION (SSDA) METHOD

In this sub-section, we show the performances of our methods compared with SSDA methods. However, we note that the suggested SSDA methods cannot be directly adapted to the federated setting without major modification. Kim & Kim (2020) uses feature alignments and requires access to the source and target data at the same time, which is usually difficult to achieve in federated learning. As for Saito et al. (2019), the overall adversarial learning objective functions consist of a loss objective on both source and target labeled data and the entropy coming from the unlabeled target data, which also cannot be directly adapted to federated learning. On the contrary, auto-weighted methods and `FedGP` have the flexibility to do the single source-target domain adaptation, which can be compared with the SSDA method, though we notice that our setting is different from SSDA since

we do not have unlabeled data on the target domain and do not leverage the information coming from this set of data. Here, we perform experiments on real-world datasets: our results suggest that auto-weighted methods and `FedGP` outperform MME (Saito et al., 2019) when the shifts are large even without using unlabeled data (overall our proposed methods have a comparable performance with MME). Also, we observe in a single source-target domain adaptation setting, auto-weighted `FedGP` usually has a better performance than auto-weighted `FedDA` and `FedGP`.

|  | 0 ->1 / 1 ->0 | 0 ->2 / 2 ->0 | 1 ->2 / 2 ->1 |
|---|---|---|---|
| MME (Saito et al., 2019) | **79.55 / 89.28** | 25.98 / 12.99 | 14.66 / 24.43 |
| `FedGP` | 79.34 / 88.34 | **90.23** / 64.47 | **90.23** / 78.70 |
| `FedDA_Auto` | 79.31 / 88.38 | 87.00 / 56.07 | 89.13 / 66.59 |
| `FedGP_Auto` | 79.34 / 71.59 | **90.23 / 80.95** | **90.23 / 79.34** |

Table 15: Colored-MNIST (0: +90%, 1: +80%, 2: -90%)

|  | 0->1 | 1->2 | 2->3 |
|---|---|---|---|
| MME (Saito et al., 2019) | **68.68** | 72.86 | **81.90** |
| `FedGP` | 67.92 | 75.30 | 77.45 |
| `FedDA_Auto` | 67.92 | **75.73** | 76.97 |
| `FedGP_Auto` | **68.45** | 74.21 | 77.69 |

Table 16: VLCS (0: C, 1: L, 2: V, 3: S)

|  | 0->1 | 1->2 | 2->3 |
|---|---|---|---|
| MME (Saito et al., 2019) | **74.51** | 54.91 | 58.50 |
| `FedGP` | 72.35 | **58.19** | 60.03 |
| `FedDA_Auto` | 72.31 | 54.26 | 61.33 |
| `FedGP_Auto` | 72.42 | 56.27 | **61.90** |

Table 17: TerraIncognita (0: L100, 1: L38, 2: L43, 3: L46)