# OpenReview forum: "Principled Federated Domain Adaptation: Gradient Projection and Auto-Weighting"
_ICLR.cc/2024/Conference — ICLR 2024 poster_

### Official Review · Reviewer_d97G · 2023-10-23

**Soundness:** 3 good
**Presentation:** 3 good
**Contribution:** 3 good
**Rating:** 6
**Confidence:** 3

**Summary:**

The submission studied domain adaptation under the federated setting. Two methods (FedDA (1) and FedGP(2)) to aggregate gradients were proposed based on the analysis of the delta error of an aggregation rule (Theorem 3.6). An auto-weighting rule (3) was proposed (FedDA_Auto and FedGP_Auto), too. Experiments showing the robustness (Figure 2) and target accuracies (Table 1) justified the proposed methods.

**Strengths:**

The submission identifies the factors that affect the performance of an aggregation rule and then proposes solutions based on the findings.
The target domain enjoys the robustness and performance increases brought by the solutions.

**Weaknesses:**

Despite a comfortable reading experience and leading performance results, I would like to raise a concern about the problem formulation.

(a) From the federated learning perspective, the server and the clients are learning together to achieve a better performance measured by the sum of ALL clients. Therefore, federatively speaking, paying the whole attention to ONE target client might not align with the original intention of studying federated learning.

**Questions:**

(b) Given the federated learning nature, multiple target clients seem more practical. How would the proposed method scale with the number of target clients?

(c) The current source clients are given and assumed to be well-trained. What are the potential and challenges to extending the proposed method to a scenario where every client learns and transfers simultaneously?

(d) What if one negates the direction of the projections of negative source gradients (e.g., the projections of g_s3 and g_s4 in Figure 1)?

(e) The behaviors of FedDA in Figure 2(a) and Figure 2(c) seem contracdict to each other. Is it trivial? Or may I have a clarification?

---

> ### Author Response · Authors · 2023-11-22
>
> Thank you for reviewing our manuscript and we appreciate your feedback for refining the final version. We address your concern and question in the following.
>
> *”…, federatively speaking, paying the whole attention to ONE target client might not align with the original intention of studying federated learning’’*
> * Thank you for this comment. Our view is that the setting remains federated in the sense that there is a central server coordinating device toward a shared objective. However, we are most interested in the applied problem setting. An example that motivates our federated domain adaptation perspective is that: as a new client with limited local data (and with domain shift) shows up, the goal for the whole federated system is coming up with a model for the new client using federated data across all the clients.
>
>   In general, the settings considered in the paper are different from the original FL setting as the target client has a limited amount of data - which means if we do normal FL, it will gain fewer benefits from the federation.  As we show in the paper, personalized baselines do not work well in these settings. However, these settings still require federated collaboration. For the target client with limited data, our framework shows how other clients with sufficient data can collaborate and help it train a better model. Also, our work focuses on Domain Adaptation, which usually has one target domain to be adapted to. Further, we acknowledge the possibility of extending the current framework to multiple targets, which serves as a promising yet non-trivial future work.
>
>
> *”Given the federated learning nature, multiple target clients seem more practical. How would the proposed method scale with the number of target clients?”*
>
> * Thanks for the great question! We can potentially have several copies of the global model for each target client in the system, where each target client learns their best-personalized model simultaneously from other source clients in the system. The direct extension of the method scales linearly with the number of target clients (in memory, central computation, and communication). Finding more efficient approaches for multi-target settings is intriguing, and left for future work.
>
> *”What are the potential and challenges to extending the proposed method to a scenario where every client learns and transfers simultaneously?”*
>
> * Generally speaking, we can separate the learning and transferring stages for this more challenging scenario: where clients are first trained using normal FL methods (e.g., FedAvg) without transferring. At some point when its model is comparatively well-trained, we can start to use it as source clients and perform transferring. However, we see the challenges are: (1) when we should stop the personalization and start to perform adaptation, (2) clients may have different best aggregation rules for transferring individually. However, one can potentially leverage our theoretical framework to decide which aggregation rule is the best to apply.

---

> ### Author Response · Authors · 2023-11-22
>
> *”What if one negates the direction of the projections of negative source gradients (e.g., the projections of g_s3 and g_s4 in Figure 1)?”*
>
> * Thanks for the interesting idea! **We performed the following experiments and the results show that negating the direction does not impact the performance much.**
>
>
>  **On Fashion-mnist with label shifts**
>
>
> | Label shifts level         | 0.45  | 0.3   | 0.15  | 0.10  | 0.05  |
> |--------------|-------|-------|-------|-------|-------|
> | fedgp        | 82.11 | 83.45 | 85.74 | 88.67 | 91.79 |
> | fedgp_negate | 82.62 | 83.30 | 86.88 | 88.38 | 91.69 |
>
>   **On Fashion-mnist with noisy features**
>
> | Target noise level | 0     | 0.2   | 0.4   | 0.6   | 0.8   |
> |--------------------|-------|-------|-------|-------|-------|
> | fedgp              | 76.33 | 75.09 | 71.09 | 68.01 | 62.22 |
> | fedgp_negate       | 75.14 | 73.63 | 69.54 | 65.17 | 61.71 |
>
>   **On Cifar-10 with noisy features**
>
> | Target noise level | 0     | 0.2   | 0.4   | 0.6   | 0.8   |
> |--------------------|-------|-------|-------|-------|-------|
> | fedgp              | 64.14 | 65.92 | 65.39 | 63.67 | 61.29 |
> | fedgp_negate       | 65.38 | 66.18 | 64.84 | 63.93 | 61.35 |
>
>
>
> *”The behaviors of FedDA in Figure 2(a) and Figure 2(c) seem contradict to each other. Is it trivial? Or may I have a clarification?”*
>
> * Thanks for the question and we appreciate the chance for clarification. The reason behind the distinct behaviors of FedDA in Figure 2(a) and 2(c) is due to the experiment settings. For 2(a), we have noisy features, so when the shifts are larger FedDA’s performance becomes worse. In this setting, the variances on the target domain do not change much. However, for 2(c), the target variance decreases as the domain shift (eta -> 0) grows bigger (easier to predict with fewer classes), FedGP and auto methods will benefit from this with higher performance when the shifts are large. However, FedDA will have a very low performance with large shifts (eta closer to 0), which makes it have a different behavior than Figure 2(a).

---

> > ### Comment · Reviewer_d97G · 2023-11-22
> > **Thank you for the feedback**
> >
> > Thank you for the materials provided. I will decide on my final score during the reviewer discussion. Due to the limited time, my preliminary understanding of combining the paper and the feedback materials is that the submission leverages the advantages of federated learning (FL) but avoids addressing the challenges of FL, such as community-wide optimization, multiclients, and asynchronous updates.

---

> > > ### Author Response · Authors · 2023-11-23
> > >
> > > Thank you for your reply. We appreciate your insightful feedback on the potential future directions of our work. Moreover, we want to highlight that we work on an extension of the standard FL setting, i.e., Federated Domain Adaptation (FDA).  Due to the coexistence of domain shift and data scarcity on the target domain, FDA on its own is a challenging yet prevalent real-world problem, in addition to the challenges of FL itself.
> > >
> > > We note that we are the first to address this problem in a principled way through a theoretical framework, and the proposed methods are demonstrated to be superior to existing work on personalized FL and unsupervised FDA, which could be the conventional wisdom to tackle the FDA problem.

---

> > > > ### Comment · Reviewer_d97G · 2023-12-05
> > > >
> > > > After reading the materials, I decided to keep my score unchanged. Thank you.

---

### Official Review · Reviewer_tJ7A · 2023-10-29

**Soundness:** 3 good
**Presentation:** 3 good
**Contribution:** 3 good
**Rating:** 6
**Confidence:** 4

**Summary:**

This paper proposes two algorithms to solve federated domain adaptation, a case in which there exists a distributional/domain shift between clients in federated learning. The authors tackle the problems of domain shift and data scarcity in their work. To solve these problems, they propose to design algorithms concerning the __server aggregation rule__, i.e., how the server merges the different gradients of the same model coming from the clients. The two proposed methods are called __FedDA__ which does a convex combination of clients' gradients (including the target), and __FedGP__, which extracts information from source clients' gradients based on the target client gradients.

**Strengths:**

__Originality.__ The authors provide a novel theoretical framework for the analysis of Federated Learning under heterogeneity.

__Quality.__ The paper is globally well-written and clear. Parts of the experimental section could be improved

__Clarity.__ The novel theoretical framework is easy to follow. Assumptions, notation and definitions are clearly stated and the proposed algorithms are intuitive.

__Significance.__ This paper tackles an extremly important problem in federated learning, i.e., how to deal with __client heterogeneity__. In this sense, besides being important for the niche of federated DA, it can also impact federated learning in general.

**Weaknesses:**

__Major__

__W1.__ The description of the real-world experiments in the main paper is not sufficient. While the authors do provide enough information in the appendix, the main paper does not fully describes the methodology the authors employed in consolidating the results of Table 1. For instance, how are the labeled data points chosen for the experiments? How does the performance change w.r.t. to the choice of data points (i.e. standard deviation of the accuracy on target domain)? These questions are answered in the appendix, but they should be clear in the main paper.

__Minor.__ (note, this point __did not__ impacted negatively in my review).

__W2.__ I would like to raise that, while this is a Federated DA paper, the authors assume access to a (small) set of labeled data in the target domain. This somewhat breaks the rules of _Unsupervised_ DA, and may bias performance towards methods that use target labeled data when comparing with UDA algorithms such as KD3A. This remains somewhat true even when supposing a small amount of labeled target data, depending on the degree of distributional shift. Nonetheless, __I do think the authors use labeled data in a clearly motivated and justified way__.

**Questions:**

__Q1.__ Concerning __FedDA__ and __FedGP__ aggregation schemes, in order to have a convex combination, shouldn't $\sum_{i=1}^{n}\beta_{i}=1$? Is this constraint enforced? For instance, in Figure 9 (appendix), the sum of betas exceeds 1.

---

> ### Author Response · Authors · 2023-11-22
>
> Thank you for reviewing our manuscript and we appreciate your feedback for refining the final version. We address your concern and question in the following.
>
> *”While the authors do provide enough information in the appendix, the main paper does not fully describes the methodology the authors employed in consolidating the results of Table 1. For instance, how are the labeled data points chosen for the experiments? How does the performance change w.r.t. to the choice of data points (i.e. standard deviation of the accuracy on target domain)? These questions are answered in the appendix, but they should be clear in the main paper.”*
>
> * Thank you for pointing out the insufficient description of the real-world experiments. The labeled data are sampled uniformly at random, and the performance of the FedGP, FedDA_Auto and FedGP_Autp are robust w.r.t. different levels of target domain data scarcity. We have revised our manuscript and made sure the information was clear in the updated main paper.
>
> *”I would like to raise that, while this is a Federated DA paper, the authors assume access to a (small) set of labeled data in the target domain. This somewhat breaks the rules of Unsupervised DA, and may bias performance towards methods that use target labeled data when comparing with UDA algorithms such as KD3A. This remains somewhat true even when supposing a small amount of labeled target data, depending on the degree of distributional shift. Nonetheless, I do think the authors use labeled data in a clearly motivated and justified way.”*
>
> * Thanks for highlighting the difference between our setting and the UDA setting. We note that we included comparisons with UDA algorithms for the sake of completeness and to provide a broader perspective on the performance of our method. Indeed, there is a distinction between the setting of Federated DA and Unsupervised DA.
>
> *Concerning FedDA and FedGP aggregation schemes, in order to have a convex combination, shouldn't (sum of beta = 1)? Is this constraint enforced? For instance, in Figure 9 (appendix), the sum of betas exceeds 1.*
>
> * Thanks for the question and we appreciate the chance to clarify it. For each $\beta_i$, it is the combination factor between one source domain $S_i$ and the target domain $T$. In other words, we use $\beta_i$ and $(1-\beta_i)$ to convexly combine the target domain model update and one source domain  $S_i$'s model update. For example, for the non-auto-weighting FedDA and FedGP, we set $\beta_i = 0.5$ for all source domains. Therefore, the sum of beta may not be 1. Then, among different source domains, we use the convention of $w_i = \frac{n_i}{n}$ ($n_i$ being the number of data points at source domain $S_i$, and $n$ being the total number of source data points) for weighting each (source, target) pair, which has a sum of 1.

---

### Official Review · Reviewer_28v7 · 2023-10-31

**Soundness:** 3 good
**Presentation:** 3 good
**Contribution:** 3 good
**Rating:** 6
**Confidence:** 3

**Summary:**

This paper studies the optimal design of an aggregator in federated domain generalization (generalizing to a target client/domain in cross-silo). The FedDA method works by aggregating interpolations (i.e. weighted averages) of the target domain gradient and the source domain gradients. The FedGP method further projects the source gradient onto the "positive direction" of the domain gradient before interpolation (i.e. zeroing out conflicting directions). The interpolation factor for each domain is chosen such that the $L^\pi$ distance between the source and target gradients is minimized w.r.t. some prior $\pi$ on the parameters. The authors show that this error can be decomposed as a noise term of the target domain and a distance term between the source and the target domains. The optimal interpolation factor would then balance the source and target gradients based on those terms, which can be estimated during training in a scheme called auto_weight. Extensive experiments show the benefits of this approach.

**Strengths:**

- The paper's analysis and experiments are well-detailed.
- The analysis is interesting and covers many aspects of the design of an optimal aggregator in the federated domain generalization setting.
- The method is intuitive and is easy to implement (save for the auto-weighting scheme).
- The improvement seems to be significant in terms of generalizing to the target domain with respect to personalized federated learning algorithms.

**Weaknesses:**

- It seems like it would be better to compare the algorithms presented in this paper to domain generalization algorithms, such as the ones shown in DomainBed's GitHub repo.
- The methods presented make sense mostly in the cross-silo setting, as mentioned in the paper, which limits its applicability to general federated learning problems with a relatively larger number of clients that can benefit a lot from methods for generalizing to new clients.
- It is mentioned multiple times that data scarcity is the setting of interest, in which FedDA and FedGP are supposed to perform more favorably. However, we do not see experiments showing the effect of data scarcity on the robustness of the performance of FedGP vs. FedAvg, for example.
- Personalized federated learning algorithms are relevant for comparison, but I think that direct comparison of such algorithms with FedGP might put them at a disadvantage since they are not specifically designed for domain generalization. ColoredMNIST, VLCS, and TerraIncognita datasets are more concerned with shifts in p(x) or p(x|y), whereas personalized FL is more concerned with shifts in p(y) and p(y|x), i.e. personalizing the prediction rather than adapting to spurious correlations or invariant attributes. You should either choose federated datasets for comparison, or you should compare your algorithm to domain generalization algorithms, e.g. IRM and others. Or why not use a hospital dataset that fits the setting you described in the paper? For example, you can consider FLamby [2].
- One work [1] from federated continual learning might be of interest (which even shares the same name FedGP). It is motivated from a similar intuition, which is to remove from the gradient its projection onto the negative direction of the reference gradient.
- In algorithm 1, the auto_weight scheme requires intermediate gradients for each domain, which might require a lot of memory.

[1] FedGP: Buffer-based Gradient Projection for Continual Federated Learning. Dai et al. 2023.
[2] FLamby: Datasets and Benchmarks for Cross-Silo Federated Learning in Realistic Healthcare Settings. Terrail et al. 2022.

**Questions:**

Can you train your algorithms and compare them on federated datasets that follows the setting of interest in the paper (cross-silo with data scarcity)?

---

> ### Author Response · Authors · 2023-11-22
>
> Thank you for reviewing our manuscript and we appreciate your feedback for refining the final version. We address your concerns and questions as follows.
>
> *”It seems like it would be better to compare the algorithms presented in this paper to domain generalization algorithms, such as the ones shown in DomainBed's GitHub repo.”*
>
> * We thank the reviewer for this constructive suggestion to make our experimental analysis more comprehensive. **We have conducted the suggested experiment and the results have been detailed in the updated version of the paper.** As summarized below, the proposed method outperforms those reported in the DomainBed's GitHub repo for the Domain Generalization (DG) methods (we report the highest average DG performance vs. our methods).
>
> |         | Colored-MNIST |  VLCS    | PACS     | Office-Home | TerraIncognita | DomainNet |
> |---------|---------------|----------|----------|-------------|----------------|-----------|
> | Ours    | **84.1**     | **83.8** | **93.4** | **74.3**    | **74.6**       | **53.7**  |
> | Best DG |  67.7         | 79.9     | 87.2     | 68.5        | 54.4           | 41.8      |
>
> *”The methods presented make sense mostly in the cross-silo setting, as mentioned in the paper, which limits its applicability to general federated learning problems with a relatively larger number of clients that can benefit a lot from methods for generalizing to new clients.“*
>
> * In our work, we focus on the cross-silo setting, but it is possible to extend our ideas to the cross-device setting. For example, we can potentially first perform clustering before applying our methods, to find a subset of clients to collaborate for better target performance. We believe our theoretical and empirical findings showcase good insights for extending current work to this different setting.
>
> *”It is mentioned multiple times that data scarcity is the setting of interest, in which FedDA and FedGP are supposed to perform more favorably. However, we do not see experiments showing the effect of data scarcity on the robustness of the performance of FedGP vs. FedAvg, for example.”*
>
> * The experiment showing the effect of data scarcity on the robustness is included in Appendix C.8 where it shows the proposed methods are robust to data scarcity. In addition, we appreciate your suggestion of comparing with FedAvg as FedAvg serves as a baseline for the effect of data scarcity. **As shown below (detailed in appendix C.8 of the updated paper), we compare to FedAvg showing the effect of data scarcity. Our FedDA_auto consistently outperforms FedAvg with large margins.**
>
> |   noise level | Number of samples |     0.2    |        |     0.4    |        |     0.6    |        |
> |--------------:|-------------------|:----------:|:------:|:----------:|:------:|:----------:|:------:|
> |               |                   | FedDA_auto | FedAvg | FedDA_auto | FedAvg | FedDA_auto | FedAvg |
> | Fashion-MNIST |               100 |  **79.04** |  75.98 |  **72.21** |  59.36 |  **66.16** |  49.94 |
> |               |               200 |  **79.74** |  76.50 |  **74.30** |  60.28 |  **69.27** |  48.56 |
> |               |               500 |  **79.48** |  75.55 |  **75.21** |  58.74 |  **71.40** |  47.93 |
> |               |              1000 |  **80.23** |  76.12 |  **76.75** |  62.33 |  **73.16** |  50.81 |
> |    CIFAR-10   |                5% |  **63.04** |  24.54 |  **60.79** |  21.25 |  **60.02** |  19.42 |
> |               |               10% |  **65.72** |  22.86 |  **64.43** |  22.35 |  **62.25** |  18.60 |
> |               |               15% |  **66.57** |  23.20 |  **65.40** |  23.25 |  **63.36** |  17.88 |

---

> > ### Author Response · Authors · 2023-11-22
> >
> > *”ColoredMNIST, VLCS, and TerraIncognita datasets are more concerned with shifts in p(x) or p(x|y), whereas personalized FL is more concerned with shifts in p(y) and p(y|x), i.e. personalizing the prediction rather than adapting to spurious correlations or invariant attributes. You should either choose federated datasets for comparison, or you should compare your algorithm to domain generalization algorithms, e.g. IRM and others. Or why not use a hospital dataset that fits the setting you described in the paper? For example, you can consider FLamby [2]. Can you train your algorithms and compare them on federated datasets that follows the setting of interest in the paper (cross-silo with data scarcity)? ”*
> >
> > * Thanks for the great suggestion and question. Our methods can be applied in both two cases: no matter whether it is label shift (p(y) and p(y|x)) or feature shift (p(x) or p(x|y)), and we covered the experiments for both cases in the experiments on Fashion-MNIST and CIFAR-10. **Moreover, we ran the suggested experiment on the federated dataset Fed-Heart in FLamby to confirm our methods outperform the personalized FL baselines, as shown below.**
> >
> > |    center   | 0 (20%) | 1 (20%)  | 2 (100%) too few samples on the target domain | 3 (20%) |  Avg  |
> > |:-----------:|:-------:|:--------:|:---------------------------------------------:|:-------:|:-----:|
> > |    FedDA    |   79.81 |    78.65 |                                         67.50 |   62.22 | 72.05 |
> > |    FedGP    |   78.85 |    80.45 |                                         68.75 |   65.33 | 73.35 |
> > |  FedDA_Auto |   80.77 |    80.90 |                                         68.75 |   71.11 | 75.38 |
> > |  FedGP_Auto |   80.77 |    79.78 |                                         68.75 |   69.78 | 74.77 |
> > | Source only |   76.92 |    75.96 |                                         62.50 |   55.56 | 67.74 |
> > |    FedAvg   |   75.96 |    76.40 |                                          62.5 |   55.56 | 67.61 |
> > |    Ditto    |   76.92 |    73.03 |                                         62.50 |   55.56 | 67.00 |
> > |    FedRep   |   78.84 |    65.17 |                                         75.00 |   60.00 | 69.75 |
> > |     APFL    |   51.92 |    57.30 |                                         31.25 |   42.22 | 45.67 |
> >
> >
> > *”One work [1] from federated continual learning might be of interest (which even shares the same name FedGP). It is motivated from a similar intuition, which is to remove from the gradient its projection onto the negative direction of the reference gradient.”*
> >
> > * Thanks for mentioning the related work.This paper focuses on the catastrophic forgetting problem in federated continual learning, which is different from our setting. Also, their way of applying gradient projection is different: they leverage the global buffer gradient, which is the average buffer gradient across all clients, as a reference to project the current gradient. However, in our work, we project the target gradient towards the source directions.
> >
> > *”In algorithm 1, the auto_weight scheme requires intermediate gradients for each domain, which might require a lot of memory.”*
> >
> > * Thanks for bringing this up. We would like to point out that the auto-weighted scheme does not require intermediate gradients for each domain, but only for the target domain. For the source domain, we can easily calculate the average gradient using model updates for 1 epoch divided by the number of batches. Also, on the target domain, since we usually only have a limited number of samples, it is not expensive to save intermediate gradients to estimate the variance term.

---

> > > ### Comment · Reviewer_28v7 · 2023-11-22
> > >
> > > Thanks for the great rebuttal and for running the extra experiments. The experiments show that your method is consistently better than some baselines. Thus, I have raised my score accordingly.

---

### Official Review · Reviewer_nVHX · 2023-10-31

**Soundness:** 3 good
**Presentation:** 3 good
**Contribution:** 2 fair
**Rating:** 5
**Confidence:** 5

**Summary:**

This paper considers two important issues in federated learning: small data at client sites and domain shift across clients. Simple, intuitive strategies such as gradient projection and auto-weighting for mitigating these issues are proposed. Several interesting theorems are proved regarding federated aggregation, gradient projectin. Results on three datasets are provided.

**Strengths:**

Nice treatment of federated learning in the presence of domain shift and small data. A good mixture of theoretical and experimental work.

**Weaknesses:**

I loved the paper till I came to the experiments section. In this day and age, should we still be doing experiments with ColoredMNIST, VLCS, CIFAR-10 and TerraIncognita? ColoredMNIST, VLCS and TerraIngocnita are from 2019, 2013 and 2018 respectively! This raises the questions whether the proposed solutions will scale to larger and difficult datasets.

**Questions:**

Try on DomainNet, Office Home and PACS datasets. Although these datasets are from 2019 and 2017, atleast they are more challenging datasets.

---

> ### Author Response · Authors · 2023-11-22
>
> Thank you for reviewing our manuscript and we appreciate your feedback for refining the final version. We address your concerns as follows.
>
> *”I loved the paper till I came to the experiments section. In this day and age, should we still be doing experiments with ColoredMNIST, VLCS, CIFAR-10 and TerraIncognita? ColoredMNIST, VLCS and TerraIngocnita are from 2019, 2013 and 2018 respectively! This raises the questions whether the proposed solutions will scale to larger and difficult datasets.”*
>
> *  We thank the reviewer for this constructive suggestion to make our experimental analysis more comprehensive. **We conducted experiments on the PACS, Office-Home, and DomainNet datasets. The results are detailed in the updated version of our paper (appendix C.3); and are summarized as follows.**
>
>     * We see **the proposed methods outperform other Domain Generalization (DG) methods with significant margins** on PACS, Office-Home and DomainNet.
>     * **Despite its simplicity and efficiency, FedGP consistently performs well across most of the cases.**
>     * We found that **the auto-weighting versions of FedDA and FedGP generally outperform their fixed-weight counterparts.** An interesting exception is observed with FedDA on the OfficeHome dataset, where the fixed weight choice of $\beta=0.5$ is surprisingly good. We observe that on Office-Home potentially the source-only baseline sometimes surpasses the Oracle performance on the target domain. Thus, we conjecture that the fixed weight $\beta=0.5$ happens to be a good choice for FedDA on Office-Home, while the noisy target domain data interferes the auto-weighting mechanism. Nevertheless, the auto-weighted FedGP still shows improvement over its fixed-weight version.
>
>
> * **PACS (with 15% of target domain data)**
>
> | Domains    | A        | C        | P        | S        | Avg      |
> |------------|----------|----------|----------|----------|----------|
> | FedDA      |     92.6 |     89.1 |     97.4 |     89.2 | 92.0     |
> | FedGP      | **94.4** |     92.2 | **97.6** | **88.9** | 93.3     |
> | FedDA_Auto |     94.2 |     90.9 |     96.6 |     89.6 | 92.8     |
> | FedGP_Auto |     94.2 | **93.7** |     97.3 |     88.3 | **93.4** |
> | Best DG    |   87.8   |   81.8   |   97.4   |   82.1   |   87.2   |
>
>
> * **Office-Home (with 15% of target domain data)**
>
> | Domains     | A        | C        | P        | R        | Avg      |
> |-------------|----------|----------|----------|----------|----------|
> | Source Only | 50.9     | 66.1     | 74.5     | 76.2     | 66.9     |
> | FedDA       | **67.9** | **68.2** | **82.6** | **78.6** | **74.3** |
> | FedGP       |     63.8 |     65.7 |     81.0 |     74.4 | 71.2     |
> | FedDA_Auto  |     67.4 |     65.6 |     82.2 |     75.8 | 72.7     |
> | FedGP_Auto  |     66.1 |     64.5 |     82.1 |     74.9 | 71.9     |
> | Oracle      |     70.9 |     58.5 |     87.4 |     75.0 | 73.0     |
> | Best DG     |     64.5 |     54.8 |     76.6 |     78.1 | 68.5     |
>
>
> * **DomainNet (with 15% of target domain data)**
>
> |                   | clip     | info     | paint    | quick    | real     | sketch   | avg      |
> |-------------------|----------|----------|----------|----------|----------|----------|----------|
> | KD3A (ResNet -50) | 63.7     | 15.4     | 53.5     | 11.5     | 65.4     | 53.4     | 43.8     |
> | FedDA             | **67.1** | 26.7     | 56.1     | 33.8     | 67.1     | **55.7** | 51.1     |
> | FedGP             | 64.0     | 26.6     | **56.8** | **51.1** | **71.3** | 52.3     | **53.7** |
> | FedDA_auto        | 62.0     | **27.8** | 56.7     | 50.9     | 68.1     | 53.3     | 53.1     |
> | FedGP_auto        | 62.2     | 27.7     | **56.8** | 50.7     | 68.4     | 53.5     | 53.2     |
> |  Best DG          | 59.2     | 19.9     | 47.4     | 14.0     | 59.8     | 50.4     |  41.8    |

---

### Author Response · Authors · 2023-11-22
**Summary of Revisions**

**We thank the reviewers for their thorough review and the constructive feedback that has significantly contributed to enhancing the quality of our paper.** Below is a summary of the key revisions we have made (changes are highlighted in blue in the revised manuscript):

1. We have detailed the descriptions in the real-world experiment section for clarification.
2. The domain generalization results have been included in the real-world experiment section, which improves the comprehensiveness of our experimental evaluation.
3. In the appendix, **we included the Fed-Heart results of FLamBy, the data scarcity ablation study with FedAvg, and the results of other DomainBed datasets (PACS, OfficeHome, and DomainNet).** These additional experiments further demonstrate the effectiveness of the proposed methods over related baselines.

**We hope we address your concerns and would greatly appreciate your re-evaluations based on the new evidence with our revisions.**

---

### Meta-Review · Area_Chair_G3ir · 2023-12-06

**Metareview:**

This paper introduces two algorithms designed to address federated domain adaptation, a scenario characterized by a distributional or domain shift among clients in federated learning. The authors specifically target the challenges posed by domain shift and data scarcity. Their approach involves the design of algorithms focusing on the server aggregation rule—determining how the server consolidates distinct gradients of the same model from various clients. Although there were initial concerns regarding the experiments and baseline comparison, the reviewer expressed satisfaction with the updates provided during the rebuttal. Therefore, I recommend acceptance of the paper.

**Justification For Why Not Higher Score:**

There is no strong support for this paper.

**Justification For Why Not Lower Score:**

N/A

---

### Decision · Program_Chairs · 2024-01-16

Accept (poster)